# HIF-1α and HIF-2α differently regulate tumour development and inflammation of clear cell renal cell carcinoma in mice

Rouven Hoefflin [1,15], Sabine Harlander [2,3,15], Silvia Schäfer[1,4,5], Patrick Metzger [5,6], Fengshen Kuo [7], Désirée Schönenberger[2,3], Mojca Adlesic[1,4], Asin Peighambari [1,4,5], Philipp Seidel[1,4], Chia-yi Chen[8], Miguel Consenza-Contreras[8], Andreas Jud[9], Bernd Lahrmann[10], Niels Grabe[10], Danijela Heide[11], Franziska M. Uhl [1,5], Timothy A. Chan [7], Justus Duyster[1,12], Robert Zeiser [1,4,12], Christoph Schell[8], Mathias Heikenwalder[11], Oliver Schilling[8,12,13], A. Ari Hakimi [7,14], Melanie Boerries[6,12,13] & Ian J. Frew [1,2,3,4,5,12 ✉]

Mutational inactivation of *VHL* is the earliest genetic event in the majority of clear cell renal cell carcinomas (ccRCC), leading to accumulation of the HIF-1α and HIF-2α transcription factors. While correlative studies of human ccRCC and functional studies using human ccRCC cell lines have implicated HIF-1α as an inhibitor and HIF-2α as a promoter of aggressive tumour behaviours, their roles in tumour onset have not been functionally addressed. Herein we show using an autochthonous ccRCC model that *Hif1a* is essential for tumour formation whereas *Hif2a* deletion has only minor effects on tumour initiation and growth. Both HIF-1α and HIF-2α are required for the clear cell phenotype. Transcriptomic and proteomic analyses reveal that HIF-1α regulates glycolysis while HIF-2α regulates genes associated with lipoprotein metabolism, ribosome biogenesis and E2F and MYC transcriptional activities. HIF-2α-deficient tumours are characterised by increased antigen presentation, interferon signalling and CD8[+] T cell infiltration and activation. Single copy loss of *HIF1A* or high levels of *HIF2A* mRNA expression correlate with altered immune microenvironments in human ccRCC. These studies reveal an oncogenic role of HIF-1α in ccRCC initiation and suggest that alterations in the balance of HIF-1α and HIF-2α activities can affect different aspects of ccRCC biology and disease aggressiveness.

[1] Department of Medicine I, Medical Center – University of Freiburg, Faculty of Medicine, University of Freiburg, Freiburg, Germany. [2] Institute of Physiology, University of Zurich, Zurich, Switzerland. [3] Zurich Center for Integrative Human Physiology, University of Zurich, Zurich, Switzerland. [4] Signalling Research Centres BIOSS and CIBSS, University of Freiburg, Freiburg, Germany. [5] Faculty of Biology, University of Freiburg, Freiburg, Germany. [6] Institute of Medical Bioinformatics and Systems Medicine, Medical Centre—University of Freiburg, Faculty of Medicine, University of Freiburg, Freiburg, Germany. [7] Immunogenomics & Precision Oncology Platform (IPOP), Memorial Sloan Kettering Cancer Center, New York, NY, USA. [8] Institute for Surgical Pathology, Medical Center – University of Freiburg, Faculty of Medicine, University of Freiburg, Freiburg, Germany. [9] Department of General and Visceral Surgery, Medical Center – University of Freiburg, Faculty of Medicine, University of Freiburg, Freiburg, Germany. [10] Hamamatsu Tissue Imaging and Analysis (TIGA) Center, BioQuant, University of Heidelberg, Heidelberg, Germany. [11] Division of Chronic Inflammation and Cancer, German Cancer Research Center (DKFZ), Heidelberg, Germany. [12] Comprehensive Cancer Center Freiburg (CCCF), Medical Center – University of Freiburg, Faculty of Medicine, University of Freiburg, Freiburg, Germany. [13] German Cancer Consortium (DKTK), Partner Site Freiburg, and German Cancer Research Center (DKFZ), Heidelberg, Germany. [14] Urology Service, Department of Surgery, Memorial Sloan Kettering Cancer Center, New York, NY, USA. [15]These authors contributed equally: Rouven Hoefflin and Sabine Harlander. ✉email: ian.frew@uniklinik-freiburg.de

More than 400,000 new cases of kidney cancer arose worldwide in 2018[1]. Clear cell renal cell carcinoma (ccRCC) represents 70–80% of all cancers of the kidneys[2]. Biallelic inactivation of the von Hippel–Lindau (VHL) tumour suppressor gene is a truncal genetic event that arises in the majority of cases of ccRCC[3–6], demonstrating that loss of one or more of the various tumour suppressor functions of the pVHL protein isoforms[2,7] is central to the earliest steps in the initiation of tumour formation. Subsequent mutations or chromosomal copy number alterations in epigenetic regulatory genes (including PBRM1, BAP1, SETD2, and KDM5C), cell-cycle regulatory genes (including TP53, CDKN2A, and MYC) or PI3K pathway genes (including PIK3CA, PTEN, MTOR, and TSC1) arise recurrently in ccRCC and are believed to cooperate with VHL inactivation to promote the development and evolution of ccRCC tumours[8,9]. Numerous mouse models have supported this notion of genetic cooperation by showing that renal epithelial cell-specific inactivation of different combinations of Vhl together with Pten[10], Tsc1[11], Pbrm1[11–13], Bap1[11,14], Trp53[15], Trp53/Rb1[16], Cdkn2a[17], or with Myc[17] overexpression causes the formation of cystic and solid precursor lesions or ccRCC tumours.

The best characterised tumour suppressor function of pVHL relates to its role in targeting the alpha subunits of the hypoxia-inducible transcription factors (HIF-1α and HIF-2α) for oxygen-dependent, ubiquitin-mediated proteolytic degradation[18]. Genetic inactivation of VHL causes the constitutive stabilisation of HIF-1α and HIF-2α, which induce gene expression programmes that play a central role in the pathogenesis of ccRCC by altering cellular metabolism, inducing angiogenesis, promoting epithelial-to-mesenchymal transition, invasion, and metastatic spread. Numerous lines of evidence argue that HIF-2α plays a major pro-tumourigenic role in established human ccRCCs, whereas HIF-1α appears to function rather to inhibit aggressive tumour behaviour. Loss of the region of chromosome 14q harbouring HIF1A correlates with poor survival[19] and is commonly found in ccRCC metastases[20]. ccRCC tumours that express only HIF-2α have higher proliferation rates than those expressing HIF-1α and HIF-2α[21]. ccRCC tumour cell lines frequently display intragenic deletions of HIF1A but express wild-type (WT) HIF-2α[22]. HIF-2α is necessary for the formation of ccRCC xenografts[23,24] while knockdown of HIF-1α enhances xenograft tumour formation in cell lines that express both HIF-1α and HIF-2α[22]. These observations have given rise to the concept that HIF-2α functions as a ccRCC oncogene and HIF-1α as a tumour suppressor. This prompted the development of HIF-2α-specific inhibitors which show excellent on-target efficacy in ccRCC xenograft models, efficacy in a subset of patient-derived xenograft models and clinical responses in some patients in phase I clinical trials[25–27]. These pharmacological studies in patient-derived xenograft models however also indicate that HIF-2α specific inhibition is not sufficient to inhibit the growth of all ccRCCs[25], suggesting that other oncogenic drivers may be important in some or all tumours. It should be noted that all of the functional and genetic data described above largely relates to either studies of established, later stage ccRCC human tumours or to the somewhat artificial setting of xenograft tumour formation by cultured ccRCC cell lines or patient-derived xenograft models. These studies have necessarily been unable to adequately assess the involvement of HIF-1α and HIF-2α throughout the entire process of tumour evolution beginning with VHL mutant cells in the context of a normal renal tubular epithelium.

To address the roles of HIF-1α and HIF-2α in the development of ccRCC we take advantage of an accurate mouse model of ccRCC based on tamoxifen-inducible renal epithelial cell-specific deletion (Ksp-CreER^T2) of Vhl, Trp53, and Rb1[16]. This mouse model at least partly reflects the complex patterns of chromosomal copy number gains and losses of cell-cycle regulatory genes in human ccRCC and reproduces many aspects of the evolution of human ccRCC by first developing cystic and solid precursor lesions that progress to tumours over the course of 5–12 months following gene deletion in adult mice[16]. Tumours arising in this model exhibit histological, immunohistochemical, transcriptional, and mutational similarities to human ccRCC[16]. We introduce floxed alleles of Hif1a and Hif2a (also known as Epas1) into this genetic background and show that HIF-1α is essential for tumour formation whereas deletion of HIF-2α has only moderate effects on tumour onset and growth rate but leads to increased intra-tumoural immune activation. This study defines differing roles of HIF-1α and HIF-2α in ccRCC formation and progression and suggests a model in which alterations in their relative activities affect different aspects of tumour biology and immunology.

## Results

**ccRCC formation is strongly dependent on Hif1a.** We fed 6-week-old mice tamoxifen-containing food for 2 weeks to induce gene deletion in cohorts of Ksp-CreER^T2; Vhl^fl/fl; Trp53^fl/fl; Rb1^fl/fl (hereafter termed Vhl^Δ/ΔTrp53^Δ/ΔRb1^Δ/Δ in the text and VpR in figures), Ksp-CreER^T2; Vhl^fl/fl; Trp53^fl/fl; Rb1^fl/fl; Hif1a^fl/fl (hereafter termed Vhl^Δ/ΔTrp53^Δ/ΔRb1^Δ/ΔHif1a^Δ/Δ in the text and VpRH1 in figures) and Ksp-CreER^T2; Vhl^fl/fl; Trp53^fl/fl; Rb1^fl/fl; Hif2a^fl/fl (hereafter termed Vhl^Δ/ΔTrp53^Δ/ΔRb1^Δ/ΔHif2a^Δ/Δ in the text and VpRH2 in figures) mice. Tumour onset, volume and numbers were monitored over time using contrast-assisted μCT imaging and mice were sacrificed at individual time points based on the presence of rapid tumour growth. These data were added to, or compared to, our previously published[16] analyses of separate Vhl^Δ/ΔTrp53^Δ/ΔRb1^Δ/Δ and Trp53^Δ/ΔRb1^Δ/Δ (termed pR in figures) cohorts, respectively. All animals from both cohorts were housed in the same animal facility. We first determined that tumour growth curves (Supplementary Fig. 1a) showed an excellent goodness of fit (Supplementary Fig. 1b) to the exponential linear regression $e^{\alpha t}$ where $\alpha$ describes the coefficient of exponential growth, a mathematical description of the tumour growth rate, and $t$ represents time in days after gene deletion. These analyses showed that Vhl deletion accelerates tumour onset (Fig. 1a), increases tumour number (Fig. 1b) and increases tumour growth rate (Fig. 1c) in the Trp53^Δ/ΔRb1^Δ/Δ background. Hif1a co-deletion completely abolished these tumour-promoting effects of Vhl deletion (Fig. 1a) and these mice developed very few tumours (Fig. 1b), which grew slowly when they did develop (Fig. 1c). In contrast, Hif2a deletion caused more moderate, yet statistically significant effects, partly delaying tumour onset (Fig. 1a), partly reducing the number of tumours per mouse (Fig. 1b) and average tumour growth rates (Fig. 1c). Metastases were not observed in any of the genotypes. These data indicate that HIF-1α is very important for the efficient evolution and growth of Vhl mutant ccRCCs, while HIF-2α is only partly required and many tumours still develop in the Vhl/Trp53/Rb1/Hif2a quadruple mutant background.

Since Hif1a deletion provided such a strong phenotypic rescue we next investigated whether the Hif1a and Hif2a genes are indeed deleted in the relevant tumours to exclude that the tumours might be escapers in which Cre activity failed to correctly recombine the floxed Hif1a or Hif2a alleles. PCRs specific for the recombined Hif1a and Hif2a alleles revealed that tumour DNA exhibited Cre-induced recombined alleles of these genes in tumours from the relevant mice (Supplementary Fig. 2a). To compare the extent of Cre-mediated deletion of Vhl, Trp53, Rb1, Hif1a, and Hif2a we conducted quantitative real-time PCR using primers to specifically amplify floxed exons of each gene

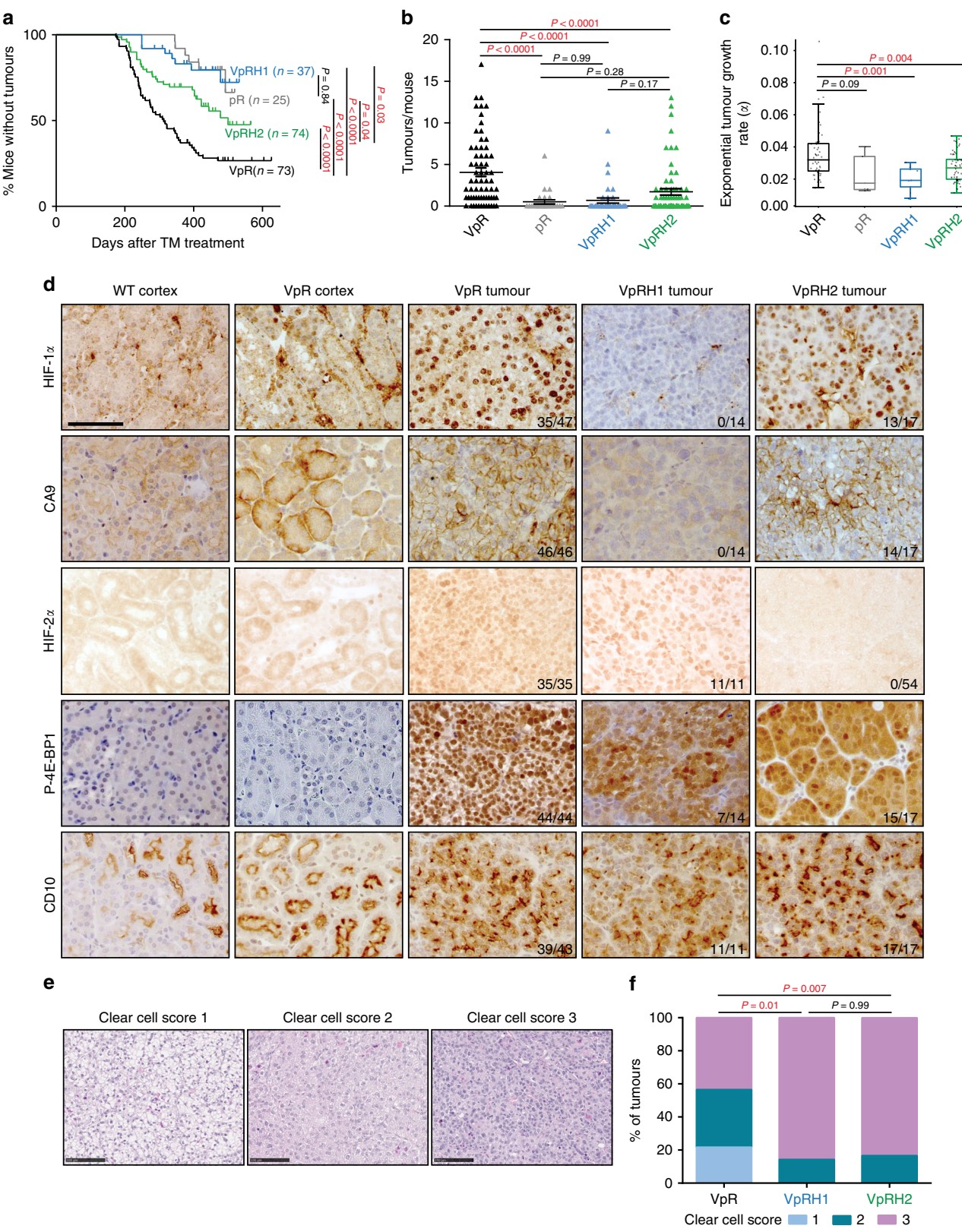

as well as non-floxed exons of the *Vhl*, *Trp53*, and *Hif2a* genes (which served as normalisation controls) from genomic DNA from cortex samples from non-Cre mice (WT cortex), as well as tumours from Vhl$^{\Delta/\Delta}$Trp53$^{\Delta/\Delta}$Rb1$^{\Delta/\Delta}$, Vhl$^{\Delta/\Delta}$Trp53$^{\Delta/\Delta}$Rb1$^{\Delta/\Delta}$Hif1a$^{\Delta/\Delta}$, and Vhl$^{\Delta/\Delta}$Trp53$^{\Delta/\Delta}$Rb1$^{\Delta/\Delta}$Hif2a$^{\Delta/\Delta}$ mice. These analyses showed that the allelic ratios of floxed exons

normalised to the average of the three non-floxed exons were reduced for *Vhl*, *Trp53*, and *Rb1* compared to WT cortex and for *Hif1a* and *Hif2a* when compared to WT cortex or to mouse genotypes that did not harbour the floxed allele (Supplementary Fig. 2b). Residual floxed exons (~10–15% allelic burden) in the genomic DNA of tumours likely reflect DNA

**Fig. 1 ccRCC formation is strongly dependent on *Hif1a* and only moderately affected by *Hif2a* deletion. a** Tumour onset in cohorts of pR, VpR, VpRH1, and VpRH2 mice. *P* values were calculated by two-sided log-rank Mantel–Cox test. **b** Number of tumours per mouse at the time of sacrifice based on μ-CT imaging (VpR $n = 65$, pR $n = 25$, VpRH1 $n = 36$ and VpRH2 $n = 65$ mice). Mean ± SEM are shown, *P* values were calculated by Dunn's multiple comparisons test. **c** Tumour growth rates based on μ-CT imaging (VpR $n = 48$, pR $n = 5$, VpRH1 $n = 7$ and VpRH2 $n = 56$ tumours). Box–whisker plots depict median, bounded by Q1 (25% lower quartile) and Q3 (75% upper quartile) and whiskers depict 1.5 times the Q3-Q1 interquartile range. *P* values were calculated by two-sided Student's *t* test without adjustment for multiple comparisons. **d** Representative immunohistochemical stainings for the indicated antibodies in samples from WT cortex, a non-tumour region of VpR cortex, VpR, VpRH1, and VpRH2 tumours. All panels are the same magnification, scale bar = 50 μm. The number of positive tumours/number of tumours examined are indicated. **e** Representative examples of the histological appearance of tumours assigned clear cell scores of 1, 2, or 3. Scale bars = 100 μm. **f** Distribution of clear cell scores between VpR ($n = 10$ mice, 23 tumours), VpRH1 ($n = 8$ mice, 14 tumours) and VpRH2 ($n = 9$ mice, 18 tumours) tumour cohorts. *P* values were calculated using the two-sided Mann–Whitney *U* test without adjustments for multiple comparisons.

derived from non-Cre-expressing cells of the tumour stroma. These analyses demonstrated that all genes are deleted by Cre and importantly that *Hif1a* and *Hif2a* are deleted to similar extents to *Vhl*, *Trp53*, and *Rb1* in the relevant tumour samples. We additionally analysed RNA-sequencing data (see experiments described below) which showed that $Vhl^{\Delta/\Delta}Trp53^{\Delta/\Delta}Rb1^{\Delta/\Delta}$ tumours displayed lower mRNA levels of *Hif1a* and *Hif2a* than WT cortex but that there was no compensatory upregulation of *Hif2a* in $Vhl^{\Delta/\Delta}Trp53^{\Delta/\Delta}Rb1^{\Delta/\Delta}Hif1a^{\Delta/\Delta}$ tumours, nor of *Hif1a* in $Vhl^{\Delta/\Delta}Trp53^{\Delta/\Delta}Rb1^{\Delta/\Delta}Hif2a^{\Delta/\Delta}$ tumours (Supplementary Fig. 3a, b). This data also revealed the specific reduction in the relative numbers of sequencing reads in the floxed exons compared to adjacent non-floxed exons. This was true for all floxed genes in the mouse genotypes that contain the floxed alleles but not in those that do not (Supplementary Fig. 2c), consistent with specific Cre-mediated recombination occurring equivalently for all genes in all genotypes. In the case of *Vhl*, the deletion of the first exon including the first intronic mRNA splice site leads to sequencing read-through into the intron. Since the intronic sequencing reads do not start at the same position in different tumour samples, it is difficult to assess the effect of this read-through on potential translation of the resulting mRNA transcript, however western blotting of primary cells derived from $Vhl^{fl/fl}$ mice demonstrated that Cre-mediated recombination results in complete loss of the pVHL protein isoforms[15] and Supplementary Fig. 5b. The slightly varying degrees of residual sequencing reads in the floxed exons of the different genes likely reflects gene expression in various types of tumour stromal cells, which likely differentially express the different genes.

Tumours in $Vhl^{\Delta/\Delta}Trp53^{\Delta/\Delta}Rb1^{\Delta/\Delta}Hif1a^{\Delta/\Delta}$ mice lacked the clear nuclear HIF-1α signal that was present in tumours from $Vhl^{\Delta/\Delta}Trp53^{\Delta/\Delta}Rb1^{\Delta/\Delta}$ and $Vhl^{\Delta/\Delta}Trp53^{\Delta/\Delta}Rb1^{\Delta/\Delta}Hif2a^{\Delta/\Delta}$ mice when staining with an anti-HIF-1α antibody (Fig. 1d). RNA-sequencing analyses identified *Car9* as a HIF-1α specific target gene (Supplementary Fig. 3c) and the protein product of this gene, CA9, showed membrane staining in tumours from $Vhl^{\Delta/\Delta}Trp53^{\Delta/\Delta}Rb1^{\Delta/\Delta}$ and $Vhl^{\Delta/\Delta}Trp53^{\Delta/\Delta}Rb1^{\Delta/\Delta}Hif2a^{\Delta/\Delta}$ mice but not in tumours from $Vhl^{\Delta/\Delta}Trp53^{\Delta/\Delta}Rb1^{\Delta/\Delta}Hif1a^{\Delta/\Delta}$ mice (Fig. 1d). Nuclear HIF-2α staining was present in tumours from $Vhl^{\Delta/\Delta}Trp53^{\Delta/\Delta}Rb1^{\Delta/\Delta}$ and $Vhl^{\Delta/\Delta}Trp53^{\Delta/\Delta}Rb1^{\Delta/\Delta}Hif1a^{\Delta/\Delta}$ mice but absent in all tumours from $Vhl^{\Delta/\Delta}Trp53^{\Delta/\Delta}Rb1^{\Delta/\Delta}Hif2a^{\Delta/\Delta}$ mice. Collectively these analyses demonstrate that Cre activity occurs equivalently for all floxed genes and that *Hif1a* and *Hif2a* are deleted in the tumours arising in the relevant mouse backgrounds.

**Characterisation of *Hif1a*- and *Hif2a*-deficient mouse ccRCC.** Histomorphological analyses and comparisons were performed for the different genetic backgrounds on a total of 26 ($Vhl^{\Delta/\Delta}Trp53^{\Delta/\Delta}Rb1^{\Delta/\Delta}$), 16 ($Vhl^{\Delta/\Delta}Trp53^{\Delta/\Delta}Rb1^{\Delta/\Delta}Hif1a^{\Delta/\Delta}$),

and 21 ($Vhl^{\Delta/\Delta}Trp53^{\Delta/\Delta}Rb1^{\Delta/\Delta}Hif2a^{\Delta/\Delta}$) H&E stained tumours. In line with our previous report[16], all $Vhl^{\Delta/\Delta}Trp53^{\Delta/\Delta}Rb1^{\Delta/\Delta}$-tumours were classified as mid-to-high grade (46% grade 2; 54% grade 3) tumours. The malignant lesions in the $Vhl^{\Delta/\Delta}Trp53^{\Delta/\Delta}Rb1^{\Delta/\Delta}Hif1a^{\Delta/\Delta}$ (19% grade 2; 75% grade 3; 6% grade 4) or $Vhl^{\Delta/\Delta}Trp53^{\Delta/\Delta}Rb1^{\Delta/\Delta}Hif2a^{\Delta/\Delta}$-background (14% grade 2; 81% grade 3; 5% grade 4) displayed on average higher grades. Since tumour grade up to grade 3 is classified mostly based on nucleolus size, these data hint that loss of HIF-1α or HIF-2α may modify processes such as transcription of ribosomal DNA genes that affect the nucleolus[28]. Potentially relevant mechanisms that have been previously linked to HIF-α activities and that might contribute to nucleolar alterations include metabolic generation of ATP and deoxynucleotides to fuel transcription, epigenetic regulatory mechanisms, and DNA repair[28]. In order to analyse similarities to the classical human ccRCC clear cell phenotype, we established a scoring system (Fig. 1e) based on a three-tiered classification of tumours with completely clear cytoplasm (score 1), partly clear or weakly stained cytoplasm (score 2), or stronger cytoplasmic eosin staining (score 3). Fifty-seven percent of the $Vhl^{\Delta/\Delta}Trp53^{\Delta/\Delta}Rb1^{\Delta/\Delta}$ tumours were classified with a clear cell score of one or two, whereas the vast majority of tumours in the other genetic backgrounds showed a score of three (86% of $Vhl^{\Delta/\Delta}Trp53^{\Delta/\Delta}Rb1^{\Delta/\Delta}Hif1a^{\Delta/\Delta}$ and 83% of $Vhl^{\Delta/\Delta}Trp53^{\Delta/\Delta}Rb1^{\Delta/\Delta}Hif2a^{\Delta/\Delta}$) (Fig. 1f). This observation is consistent with our previous findings that HIF-1α is necessary for the clear cell phenotype of normal renal epithelial cells following *Vhl* deletion[29] but also implicates HIF-2α in the clear cell phenotype in this tumour model. Intra-tumoural histomorphological heterogeneity was observed mainly in $Vhl^{\Delta/\Delta}Trp53^{\Delta/\Delta}Rb1^{\Delta/\Delta}$ (23%) and $Vhl^{\Delta/\Delta}Trp53^{\Delta/\Delta}Rb1^{\Delta/\Delta}Hif2a^{\Delta/\Delta}$ tumours (28%) (Supplementary Fig. 4a) and is a well-known characteristic of human ccRCC[30]. Other typical histopathological features of ccRCC like necrosis (Supplementary Fig. 4b) or intra-tumoural haemorrhage (Supplementary Fig. 4c) were equally distributed throughout the different genotypes and were mainly detected in larger tumours. The vast majority of tumours showed a solid and spherical growth pattern with pushing rather than infiltrating borders. Hemangioinvasion with direct tumour infiltration of blood vessels or extra-parenchymal invasion of the perirenal fat tissue was not observed in any of the cases. A subset of $Vhl^{\Delta/\Delta}Trp53^{\Delta/\Delta}Rb1^{\Delta/\Delta}$ (15%), $Vhl^{\Delta/\Delta}Trp53^{\Delta/\Delta}Rb1^{\Delta/\Delta}Hif1a^{\Delta/\Delta}$ (25%), and $Vhl^{\Delta/\Delta}Trp53^{\Delta/\Delta}Rb1^{\Delta/\Delta}Hif2a^{\Delta/\Delta}$ tumours (19%) exhibited cystic features (Supplementary Fig. 4d). While all ccRCC in the $Vhl^{\Delta/\Delta}Trp53^{\Delta/\Delta}Rb1^{\Delta/\Delta}$ background showed strong phospho-4E-BP1-staining, indicative of PI3K/mTOR-pathway activation, only 50% of the $Vhl^{\Delta/\Delta}Trp53^{\Delta/\Delta}Rb1^{\Delta/\Delta}Hif1a^{\Delta/\Delta}$ and 88% of the $Vhl^{\Delta/\Delta}Trp53^{\Delta/\Delta}Rb1^{\Delta/\Delta}Hif2a^{\Delta/\Delta}$ tumours were positive (Fig. 1d). Irrespective of the genetic background, all malignant lesions stained positively for the proximal tubule marker CD10 (Fig. 1d).

**Cancer assays do not reflect HIF-1α's in vivo oncogenic role**. Since our genetic experiments demonstrated that HIF-1α is necessary for the efficient initiation of ccRCC formation we wondered firstly if HIF-1α is generally required for cellular proliferation following loss of *Vhl* and secondly whether established Vhl$^{\Delta/\Delta}$Trp53$^{\Delta/\Delta}$Rb1$^{\Delta/\Delta}$ tumours remain dependent on HIF-1α or whether they might lose this dependency during tumour evolution. To mimic the earliest events in ccRCC formation in a genetically tractable cellular system that allows long-term proliferation assays, we derived mouse embryo fibroblasts (MEFs) from wild type, *Vhl$^{fl/fl}$*, *Vhl$^{fl/fl}$Hif1a$^{fl/fl}$*, *Vhl$^{fl/fl}$Hif2a$^{fl/fl}$*, and *Vhl$^{fl/fl}$ Hif1a$^{fl/fl}$Hif2a$^{fl/fl}$* embryos, infected them with Adeno-GFP as control or Adeno-Cre-GFP and confirmed the deletion of the floxed genes by real-time PCR and western blotting (Supplementary Fig. 5a, b). Long-term proliferation assays confirmed previous findings that loss of *Vhl* in MEFs induces an early loss of proliferative capacity, however, in contrast to the initial claim that this senescence phenotype is independent of HIF-α activity[31], our results clearly demonstrate the dependency on *Hif1a* but not on *Hif2a* (Fig. 2a). This proliferative rescue due to *Hif1a* deletion contrasts with the suppression of ccRCC initiation by *Hif1a* deletion in vivo. To remove the potential confounding factor of senescence we took advantage of the fact that deletion of *Trp53* overcomes the phenotype of loss of proliferative capacity associated with loss of *Vhl*[15,32]. To investigate the effect of loss of HIF-1α function in immortalised cells we infected *Vhl$^{fl/fl}$Trp53$^{fl/fl}$* MEFs with lentiviruses expressing either non-silencing control shRNA or expressing two different shRNAs against *Hif1a* and infected the cells with Adeno-GFP or Adeno-Cre-GFP. Western blotting confirmed the reduced abundance of pVHL, p53, and HIF-1α (Supplementary Fig. 5c). Knockdown of HIF-1α further increased the proliferation rate of immortalised *Vhl/Trp53* null MEFs (Fig. 2b), furthering illustrating that HIF-1α generally acts to inhibit proliferation in the context of *Vhl* deletion.

We next used a cell line derived from a mouse Vhl$^{\Delta/\Delta}$Trp53$^{\Delta/\Delta}$ Rb1$^{\Delta/\Delta}$ ccRCC (termed 2020 cells) (Supplementary Fig. 5d) and introduced human pVHL30 to rescue *Vhl* function (Supplementary Fig. 5e) as well as knocked down *Hif1a* with two independent shRNAs (Supplementary Fig. 5f). We confirmed the efficient functional re-introduction of pVHL30 and knockdown of *Hif1a* mRNA in reducing HIF-1α protein (Supplementary Fig. 5g) and showed that the knockdowns reduced the abundance of the PDK1 and LDH-A proteins, that are encoded by the HIF-1α transcriptional target genes *Pdk1* and *Ldha*, equivalently to pVHL30 re-introduction (Supplementary Fig. 5h). Proliferation assays revealed that neither pVHL30 re-introduction (Fig. 2c), nor *Hif1a* (Fig. 2d) knockdown affected cellular proliferation of 2020 cells growing in renal epithelial medium on cell culture plastic but that either of these manipulations were sufficient to increase the growth of 2020 cells as spheroids in non-adherent cell culture conditions (Fig. 2e–g), a common readout of cellular transformation. Since re-introduction of pVHL into human ccRCC cell lines does not affect proliferation rates in culture, but does inhibit tumour formation in the xenograft setting[33], we conducted allograft studies in SCID-Beige mice. pVHL re-introduction into 2020 cells significantly delayed tumour growth (Fig. 2h) but HIF-1α knockdown did not (Fig. 2i).

Collectively, these studies show that HIF-1α in fact antagonises cellular proliferation of normal mouse cells lacking *Vhl* and is dispensable for proliferation and allograft tumour formation of a mouse ccRCC cell line, highlighting the specificity of the requirement for HIF-1α for tumour onset in the autochthonous setting. This argues that the oncogenic role of HIF-1α is evident only in the context of the physiological environment of the renal epithelium.

**Impact of HIF-1α and HIF-2α on mouse ccRCC transcriptome**. Our previous analyses demonstrated that the mouse ccRCC model exhibits an excellent overlap with human ccRCC at the global transcriptional level[16]. To gain further insight into in vivo relevant functions of HIF-1α and HIF-2α we compared the molecular features of tumours that developed in the presence of both HIF-1α and HIF-2α to those that were genetically restricted to develop in the absence of either HIF-1α or HIF-2α. We conducted RNA sequencing of six WT cortex samples, six Vhl$^{\Delta/\Delta}$ Trp53$^{\Delta/\Delta}$Rb1$^{\Delta/\Delta}$ tumour samples, eight Vhl$^{\Delta/\Delta}$Trp53$^{\Delta/\Delta}$ Rb1$^{\Delta/\Delta}$Hif1a$^{\Delta/\Delta}$ tumour samples, and ten Vhl$^{\Delta/\Delta}$Trp53$^{\Delta/\Delta}$Rb1$^{\Delta/\Delta}$ Hif2a$^{\Delta/\Delta}$ tumour samples, and combined these data with our previously obtained RNA-sequencing data from three WT cortex samples and six Vhl$^{\Delta/\Delta}$Trp53$^{\Delta/\Delta}$Rb1$^{\Delta/\Delta}$ tumour samples. After read trimming and mapping (Supplementary Fig. 6a, b) the mRNA abundance of 19,723 genes was determined in each sample. All normalised gene expression values are provided in Supplementary Data 1. Transcriptomic profile principal component analysis and unsupervised hierarchical clustering by sample Euclidean distance matrix (Supplementary Fig. 6c) suggested minimal batch effect amongst different sequencing runs. Principal component analysis (Fig. 3a) also revealed clear separation of WT cortex from all tumour samples on the PC1 axis and this accounted for 36% of the overall variability. Vhl$^{\Delta/\Delta}$Trp53$^{\Delta/\Delta}$ Rb1$^{\Delta/\Delta}$ tumours and Vhl$^{\Delta/\Delta}$Trp53$^{\Delta/\Delta}$Rb1$^{\Delta/\Delta}$Hif2a$^{\Delta/\Delta}$ tumours tended to segregate from one another on the PC2 axis, which represented 9% of total variability, suggesting that they are the most distinct in terms of gene expression patterns, whereas Vhl$^{\Delta/\Delta}$Trp53$^{\Delta/\Delta}$Rb1$^{\Delta/\Delta}$Hif1a$^{\Delta/\Delta}$ tumours were more widely distributed along the entire axis. These analyses are consistent with the deletion of *Vhl*, *Trp53*, and *Rb1* in all three tumour genotypes inducing large transcriptional changes, with more limited and specific contributions of HIF-1α and HIF-2α to the regulation of specific sets of genes.

We focused analyses on genes that were differentially expressed between Vhl$^{\Delta/\Delta}$Trp53$^{\Delta/\Delta}$Rb1$^{\Delta/\Delta}$ tumours and the Vhl$^{\Delta/\Delta}$ Trp53$^{\Delta/\Delta}$Rb1$^{\Delta/\Delta}$Hif1a$^{\Delta/\Delta}$ and Vhl$^{\Delta/\Delta}$Trp53$^{\Delta/\Delta}$Rb1$^{\Delta/\Delta}$Hif2a$^{\Delta/\Delta}$ tumour genotypes and identified 396 differentially expressed genes that are dependent on HIF-1α (Supplementary Fig. 7a), 804 differentially expressed genes that are dependent on HIF-2α (Supplementary Fig. 7b) and 131 differentially expressed genes that are dependent on both HIF-1α and HIF-2α (Supplementary Fig. 7c). To begin to identify biological processes that these sets of genes are likely to reflect or regulate, we conducted generally applicable gene-set enrichment (GAGE) analyses using the pathway databases from ConsensusPathDB, the Biological Processes from Gene Ontology and MSigDB terms for Chemical and Genetic Perturbations and Transcription Factor Targets. The full list of statistically significantly altered (*P adj.* <0.05) gene sets of these GAGE analyses are provided in Supplementary Data 2. Vhl$^{\Delta/\Delta}$Trp53$^{\Delta/\Delta}$Rb1$^{\Delta/\Delta}$Hif1a$^{\Delta/\Delta}$ tumours in comparison to Vhl$^{\Delta/\Delta}$Trp53$^{\Delta/\Delta}$Rb1$^{\Delta/\Delta}$ and Vhl$^{\Delta/\Delta}$Trp53$^{\Delta/\Delta}$Rb1$^{\Delta/\Delta}$Hif2a$^{\Delta/\Delta}$ tumours show low expression of glycolytic genes (Fig. 3b), as well as signatures associated with hypoxia and known HIF-1α targets. This is consistent with a large body of previous work in human ccRCC cells[34] and in mouse models[29] that shows that HIF-1α is the primary transcription factor that promotes Warburg-like metabolism of high rates of glycolysis and low oxidative phosphorylation. Additional HIF-1α-dependent signatures include reduced expression of genes involved in cell adhesion (Fig. 3c) and focal adhesion and receptor signalling (Fig. 3d). Genes that were expressed at low levels in Vhl$^{\Delta/\Delta}$Trp53$^{\Delta/\Delta}$Rb1$^{\Delta/\Delta}$ Hif2a$^{\Delta/\Delta}$ tumours compared to the other two tumour genotypes included those involved in different DNA repair processes (e.g. *Fancf*—DNA interstrand cross link repair, *Rad52*—homologous

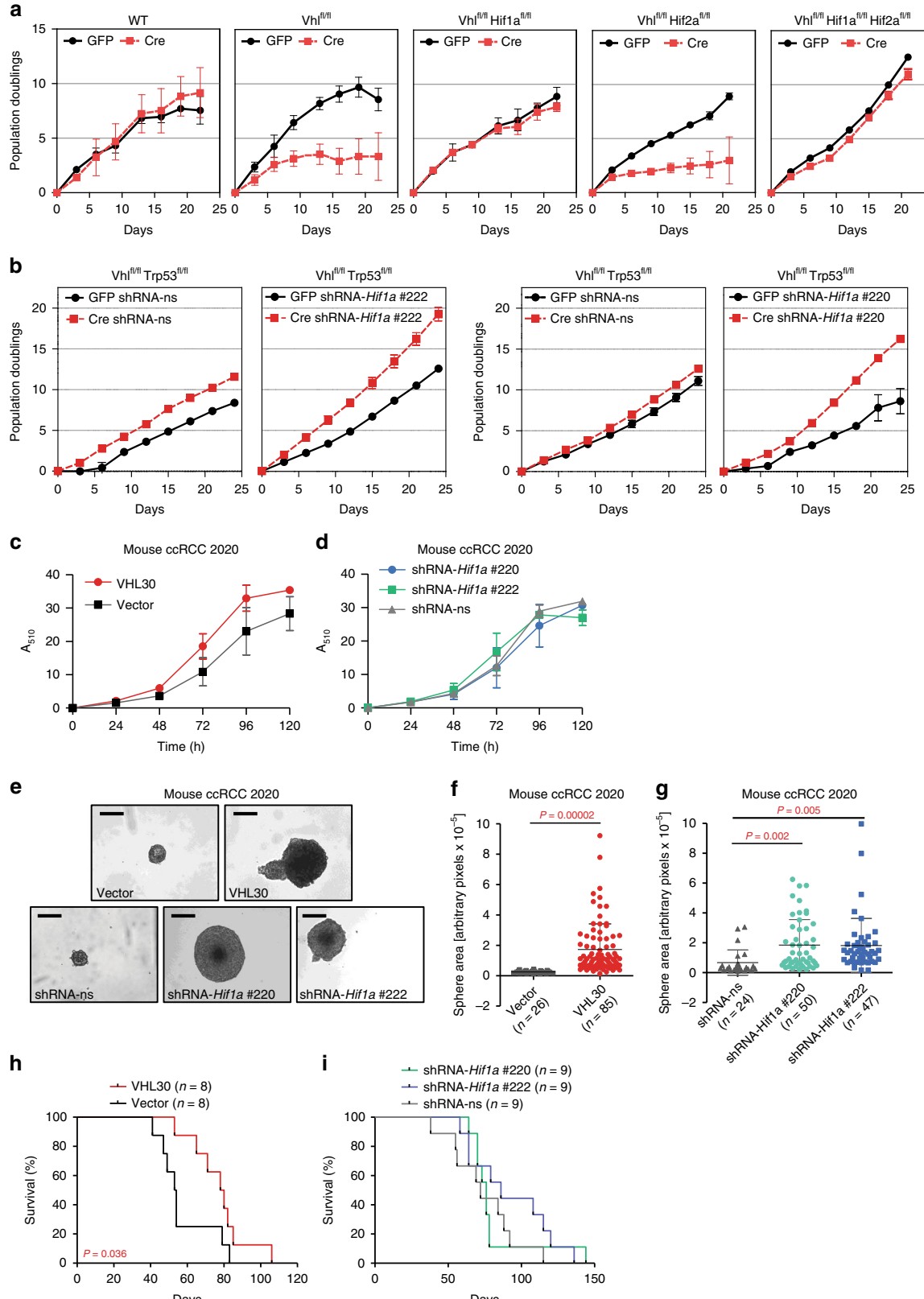

recombination repair, *Ogg1*—oxidative stress induced base excision repair, *Ercc2*—transcription coupled nucleotide excision repair) (Fig. 3e), cholesterol uptake and lipoprotein metabolism (Fig. 3f), which may relate to the observed dependency of the clear cell phenotype on HIF-2α, and ribosome biogenesis (Fig. 3g),

potentially consistent with the slower rate of growth of Vhl$^{Δ/Δ}$ Trp53$^{Δ/Δ}$Rb1$^{Δ/Δ}$Hif2a$^{Δ/Δ}$ tumours. Other HIF-2α-dependent GAGE terms that might be relevant to the evolution and proliferation of Vhl$^{Δ/Δ}$Trp53$^{Δ/Δ}$Rb1$^{Δ/Δ}$Hif2a$^{Δ/Δ}$ tumours included genes that are targets of the MYC and E2F transcription

**Fig. 2 HIF-1α is dispensable for cellular proliferation and for allograft tumour formation. a** 3T3 proliferation assays of MEFs derived from mice of the indicated genotypes infected with adenoviruses expressing GFP or Cre. Mean ± std. dev. are derived from three independent cultures. **b** 3T3 proliferation assays of MEFs derived from *Vhl*fl/fl*Trp53*fl/fl mice infected with non-silencing shRNA (shRNA-ns) or shRNA against *Hif1a* (shRNA-*Hif1a* #1 and shRNA-*Hif1a* #2), followed by infection with adenoviruses expressing GFP or Cre. Mean ± std. dev. are derived from three independent cultures. **c, d** Proliferation assays of mouse ccRCC cell line 2020 expressing empty vector control or human pVHL30 **c** or non-silencing shRNA (shRNA-ns) or shRNA against *Hif1a* (shRNA-*Hif1a* #1 and shRNA-*Hif1a* #2) **d**. Mean ± std. dev. are derived from two independent experiments each with replicates of six cultures. **e–g** Representative images (scale bars depict 200 μm) **e** and size distributions **f, g** of spheres formed by the cells described in **c, d** when grown in non-adherent cell culture plates. Mean ± std. dev. of the total number of colonies pooled from three independent experiments are shown, *P* values were calculated by two-sided Student's *t* test. **h, i** Survival of mice following subcutaneous allograft tumour assays of the cells described in c into SCID-Beige mice. *P* values were calculated by two-sided log-rank Mantel–Cox test.

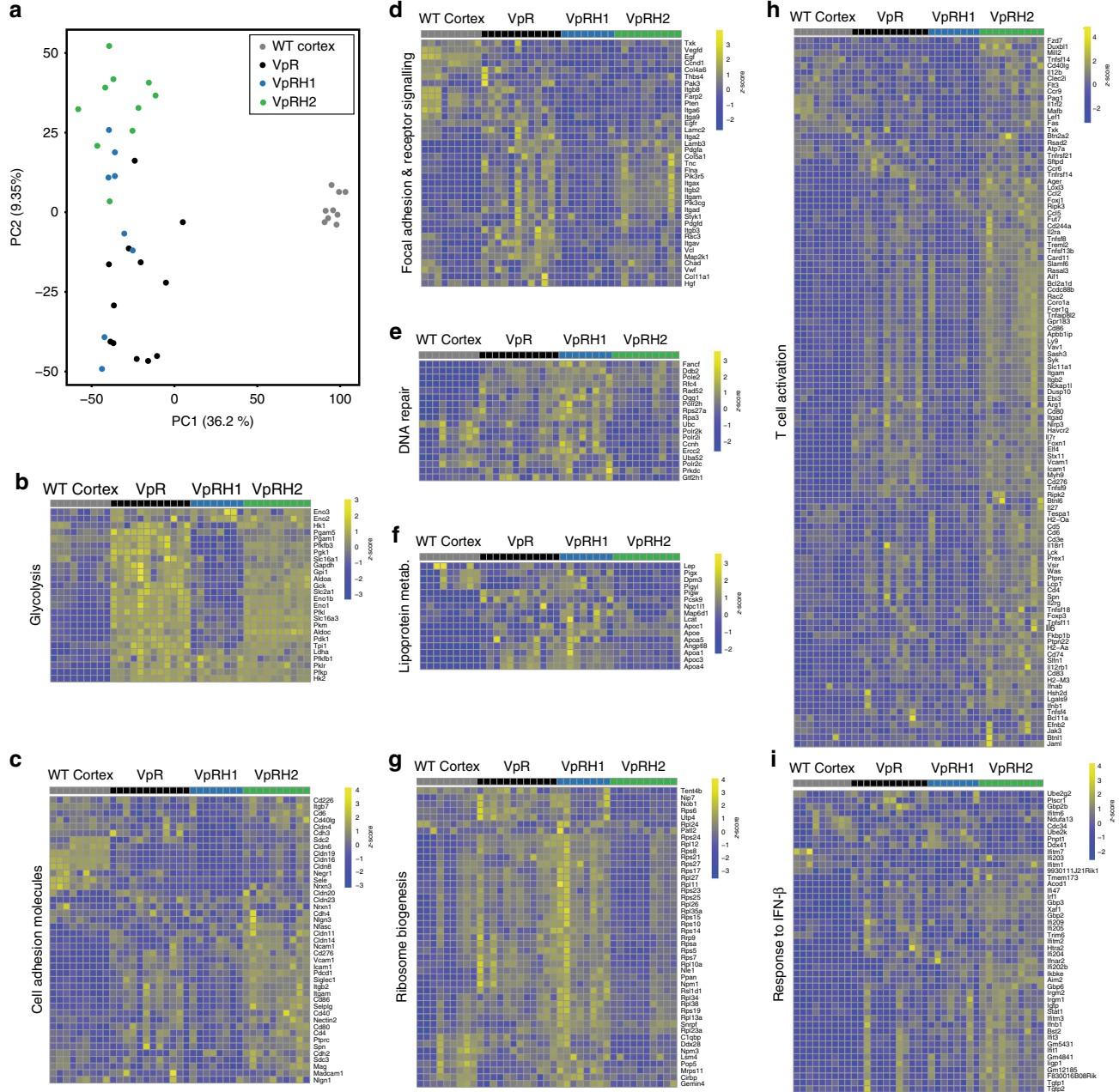

**Fig. 3 HIF-1α and HIF-2α deletion affect different transcriptional programmes and inflammatory responses. a** Principal component analysis of RNA sequencing of WT Cortex and VpR, VpRH1, and VpRH2 tumours. Gene expression heatmaps for selected differentially regulated genes from the indicated GSEA terms glycolysis (**b**), cell adhesion molecules (**c**), focal adhesion and receptor signalling (**d**), DNA repair (**e**), lipoprotein metabolism (**f**), ribosome biogenesis (**g**), T-cell activation (**h**), and response to IFN-β (**i**). Rows represent row-normalised *z*-scores of mRNA abundance, each column represents an individual sample from WT cortex or VpR, VpRH1, and VpRH2 tumours. Source data is provided in Supplementary Data 1 and 2.

factors, consistent with previous studies showing that HIF-2α promotes the activity of the MYC transcription factor[21]. GAGE analyses also identified a significant downregulation of the gene set HIF-2α Transcription Network, including *Epo*, *Egln1*, *Egln3*, *Igfbp1*, and *Pfkfb3*. To further investigate potential overlap with recently defined HIF-2α-dependent genes in human ccRCC, we used a set of 277 genes that were identified as being inhibited specifically in tumour cells in ccRCC tumorgrafts in mice treated with the HIF-2α inhibitor PT2399[25,35]. Analyses of the expression levels of the mouse orthologues of this set of HIF-2α target genes revealed that many of these genes are highly upregulated in $Vhl^{\Delta/\Delta}Trp53^{\Delta/\Delta}Rb1^{\Delta/\Delta}$ tumours compared to WT cortex, but that the loss of either HIF-1α or HIF-2α did not broadly affect the upregulation of these genes (Supplementary Fig. 7d). Nonetheless, 11 genes, marked in red in Supplementary Fig. 7d and shown in Supplementary Fig. 7e, were expressed at significantly lower levels (fold change $< -1.7$, $P < 0.05$) in $Vhl^{\Delta/\Delta}Trp53^{\Delta/\Delta}Rb1^{\Delta/\Delta}Hif2a^{\Delta/\Delta}$ tumours than in $Vhl^{\Delta/\Delta}Trp53^{\Delta/\Delta}Rb1^{\Delta/\Delta}$ tumours. The relatively small overlap between mouse and human HIF-2α-dependent genes may be due to inherent differences between mice and humans, to the very different experimental settings of acute pharmacological inhibition versus tumour evolution in the genetic absence of *Hif2a*, or to specific features of this particular model of ccRCC. In this latter context, it is noteworthy that many human HIF-2α-dependent ccRCC genes are related to the cell cycle and to DNA damage responses. These signatures are highly represented in the comparison $Vhl^{\Delta/\Delta}Trp53^{\Delta/\Delta}Rb1^{\Delta/\Delta}$ versus WT cortex (see GAGE signatures in Supplementary Data 2). We speculate that it is likely that these genes are not dependent on HIF-2α in the mouse model due to the fact that the deletion of *Rb1* and *Trp53* already strongly affects these classes of genes.

Interestingly, genes that were upregulated in HIF-2α-deficient tumours include those enriched in diverse GAGE terms for interferon signalling, T-cell activation, innate immunity, adaptive immunity, antigen processing and presentation, and NF-κB as well as IRF transcription factor targets, suggestive of an altered immune environment in these tumours. Supplementary Fig. 8a shows a selection of these enriched immune signatures and highlights that the signatures are upregulated in $Vhl^{\Delta/\Delta}Trp53^{\Delta/\Delta}Rb1^{\Delta/\Delta}$ tumours compared to WT cortex, that there are very few or no statistically significant differences in these signatures between $Vhl^{\Delta/\Delta}Trp53^{\Delta/\Delta}Rb1^{\Delta/\Delta}Hif1a^{\Delta/\Delta}$ and $Vhl^{\Delta/\Delta}Trp53^{\Delta/\Delta}Rb1^{\Delta/\Delta}$ tumours (i.e. that HIF-1α deficiency does not strongly alter the inflammatory tumour environment in $Vhl^{\Delta/\Delta}Trp53^{\Delta/\Delta}Rb1^{\Delta/\Delta}$ tumours) and that all of these signatures are further highly statistically significantly upregulated in $Vhl^{\Delta/\Delta}Trp53^{\Delta/\Delta}Rb1^{\Delta/\Delta}Hif2a^{\Delta/\Delta}$ tumours in comparison to $Vhl^{\Delta/\Delta}Trp53^{\Delta/\Delta}Rb1^{\Delta/\Delta}$ tumours. Gene expression heatmaps of differentially expressed genes associated with GAGE terms for T-cell activation (Fig. 3h), response to IFN-β (Fig. 3i), and IFN-γ production (Supplementary Fig. 8b) are shown as examples of these inflammatory signatures. We conclude that these analyses suggest that there is a complex inflammatory response in $Vhl^{\Delta/\Delta}Trp53^{\Delta/\Delta}Rb1^{\Delta/\Delta}$ tumours that is further modified by HIF-2α deficiency. These phenotypes were further investigated in experiments described in the following sections.

**Impact of HIF-1α and HIF-2α on mouse ccRCC proteome.** In order to further explore whether the biological alterations predicted by transcriptomic analyses are also reflected at the protein expression level, as well as to attempt to capture differences in the proteomes of the tumours that might not be reflected in their transcriptomes, we used exploratory quantitative proteomic analyses of six samples of WT cortex and six tumours each from $Vhl^{\Delta/\Delta}Trp53^{\Delta/\Delta}Rb1^{\Delta/\Delta}$, $Vhl^{\Delta/\Delta}Trp53^{\Delta/\Delta}Rb1^{\Delta/\Delta}Hif1a^{\Delta/\Delta}$, and

$Vhl^{\Delta/\Delta}Trp53^{\Delta/\Delta}Rb1^{\Delta/\Delta}Hif2a^{\Delta/\Delta}$ mice as an independent discovery tool. These analyses allowed the quantification of 4257 proteins that were present in at least four of six samples of each genotype (Supplementary Data 3). As is commonly observed in comparisons of proteome and transcriptome data, the overall correlations of protein abundance and mRNA abundance were low (Supplementary Fig. 9a). However, there were strong correlations between fold changes in mRNA abundance and fold changes in protein abundances when analysing only those proteins that showed differential expression between genotypes (Supplementary Fig. 9b–d). Through comparison with previously conducted analyses of the proteome of eight human ccRCC tumours[36,37], we identified a strong correlation in the relative abundance of proteins in mouse $Vhl^{\Delta/\Delta}Trp53^{\Delta/\Delta}Rb1^{\Delta/\Delta}$ ccRCC and in human ccRCC (Fig. 4a). Of the differentially expressed proteins identified in the comparison between mouse $Vhl^{\Delta/\Delta}Trp53^{\Delta/\Delta}Rb1^{\Delta/\Delta}$ ccRCC and WT cortex, 82% were also identified as differentially expressed proteins in comparisons of human ccRCC to normal kidney (Fig. 4b), further emphasising that the $Vhl^{\Delta/\Delta}Trp53^{\Delta/\Delta}Rb1^{\Delta/\Delta}$ model accurately reflects the molecular features of human ccRCC. Using a less stringent cut-off for statistical significance ($P < 0.01$), we identified 884 proteins that are upregulated in $Vhl^{\Delta/\Delta}Trp53^{\Delta/\Delta}Rb1^{\Delta/\Delta}$ ccRCCs compared to WT cortex (Fig. 4c). To characterise biological pathways that are altered in tumour compared to normal tissue, we conducted two complementary analyses; ROAST (rotation gene set testing) analysis[38] was used to assess gene set enrichment based on the expression levels of all measured proteins and gene set enrichment analysis using the online platform of MSigDB (https://www.gsea-msigdb.org/gsea/msigdb/index.jsp) was performed using only the lists of statistically differentially upregulated proteins. These analyses revealed many overlaps with one another as well as with GAGE gene set terms that emerged from the analyses of the transcriptome, including glycolysis, hypoxia, DNA repair, mTORC1 signalling, E2F, and MYC targets and IFNγ response (Supplementary Data 3 and Supplementary Fig. 10a).

To compare the effect of the absence of HIF-1α or HIF-2α on the proteome, we first conducted principal components analysis (Supplementary Fig. 10b), which revealed that all tumour samples clustered separately from the WT cortex samples, but that the tumour samples of all of the different genotypes largely overlapped with one another, suggesting a relatively high degree of similarity in the overall protein expression patterns of tumours from the different genetic backgrounds. ROAST analyses as well as gene set enrichment analyses of the lists of proteins that are differentially expressed between $Vhl^{\Delta/\Delta}Trp53^{\Delta/\Delta}Rb1^{\Delta/\Delta}Hif1a^{\Delta/\Delta}$ and $Vhl^{\Delta/\Delta}Trp53^{\Delta/\Delta}Rb1^{\Delta/\Delta}$ tumours (Fig. 4d) and between $Vhl^{\Delta/\Delta}Trp53^{\Delta/\Delta}Rb1^{\Delta/\Delta}Hif2a^{\Delta/\Delta}$ and $Vhl^{\Delta/\Delta}Trp53^{\Delta/\Delta}Rb1^{\Delta/\Delta}$ tumours (Fig. 4e and Supplementary Data 3) both also highlighted numerous similarities to the transcriptomic analyses. HIF-1α deficiency reduced expression of glycolytic enzymes and increased expression of proteins associated with oxidative phosphorylation and respiratory electron transport (Fig. 4f), while HIF-2α deficiency reduced the expression of MYC targets and resulted in increased expression of genes associated with immune responses, interferon signalling, cytokine signalling, and antigen presentation (Fig. 4g). In conclusion, the analyses of the proteomes strongly align with the analyses of the transcriptomes, providing independent validation for the predicted biological differences between the tumour genotypes.

**HIF-2α deficiency alters antigen presentation in ccRCC.** Our transcriptomic and proteomic analyses implicated antigen presentation as being upregulated in $Vhl^{\Delta/\Delta}Trp53^{\Delta/\Delta}Rb1^{\Delta/\Delta}Hif2a^{\Delta/\Delta}$ tumours. Heatmaps of RNA-sequencing data revealed higher

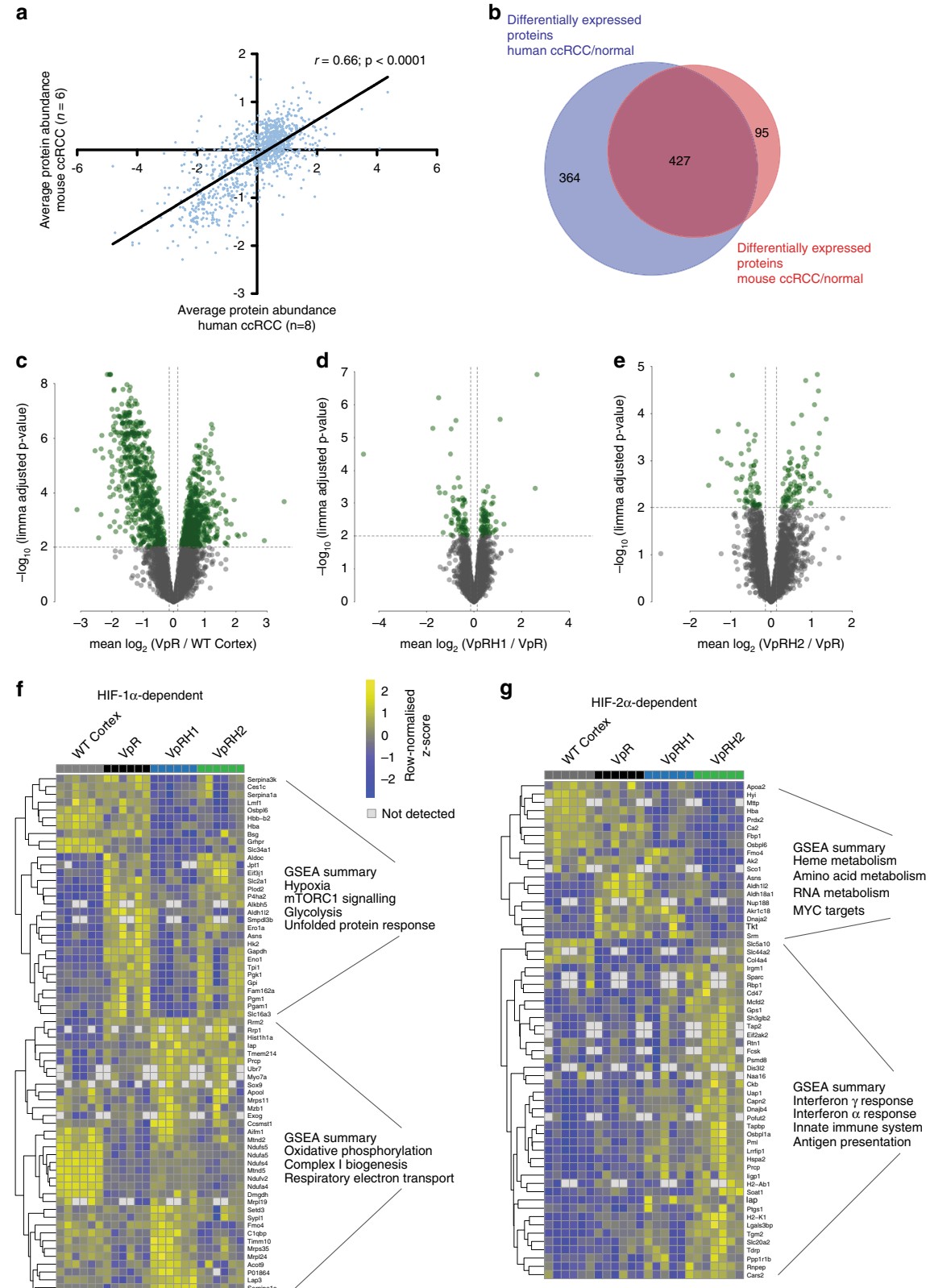

**Fig. 4 Proteomic analyses of the effects of HIF-1α and HIF-2α deletion in mouse ccRCCs. a** Correlation of protein abundance in mouse and human ccRCC samples. Each dot represents a unique pair of orthologous proteins between the two species. Spearmann's correlation coefficient is depicted. **b** Venn diagram showing the overlap of differentially expressed proteins derived from comparison of mouse ccRCC with WT cortex and human ccRCC with normal kidney. Volcano plots showing differentially expressed proteins (green dots) between VpR and WT cortex (**c**), VpRH1 and VpR (**d**), and VpRH2 and VpR (**e**). Protein expression heatmaps showing differentially expressed proteins between VpRH1 and VpR (**f**) and VpRH2 and VpR (**g**) as well as a summary of GSEA terms associated with the down- and upregulated proteins. Rows represent row-normalised z-scores of protein abundance, each column represents an individual sample from WT cortex or VpR, VpRH1, and VpRH2 tumours. Source data is provided in Supplementary Data 3.

levels of expression of many MHC class I (Fig. 5a) and MHC class II (Fig. 5b) genes, as well as other genes involved in antigen processing and presentation (Fig. 5c) in $Vhl^{\Delta/\Delta}Trp53^{\Delta/\Delta}Rb1^{\Delta/\Delta}Hif2a^{\Delta/\Delta}$ tumours than in the tumours of the other genotypes. Immunohistochemical staining with an anti-MHC class II antibody revealed that all tumour genotypes displayed cases in which the tumour cells were either negative, partly positive, or almost entirely positive (Fig. 5d). However, tumour cells in $Vhl^{\Delta/\Delta}Trp53^{\Delta/\Delta}Rb1^{\Delta/\Delta}Hif2a^{\Delta/\Delta}$ tumours were more frequently positive than the other genotypes (Fig. 5e). Since MHC class II expression is upregulated by interferon-γ signalling, these results are consistent with the fact that interferon signalling terms were upregulated in $Vhl^{\Delta/\Delta}Trp53^{\Delta/\Delta}Rb1^{\Delta/\Delta}Hif2a^{\Delta/\Delta}$ tumours in our transcriptomic and proteomic analyses. Tumour cells in at least half of all human ccRCCs are immunohistochemically positive for MHC class II expression[39,40] and ccRCC cells have been shown to present class II ligands[41]. Building upon these observations, we next sought to determine in human ccRCC whether *HIF2A* (also known as *EPAS1*) expression correlates with expression of MHC class I and class II genes, as well as other genes involved in antigen processing and presentation. Analyses of TCGA mRNA expression data revealed that *HIF2A*, but not *HIF1A*, is more highly abundant in ccRCC in comparison to normal kidney and that *HIF2A* shows a wide distribution of expression levels amongst tumours (Fig. 5f, g). This upregulation and wide expression level distribution is not observed in chromophobe RCC or papillary RCC (Fig. 5f, g). The wide expression distribution provided a good basis to investigate potential correlations between *HIF2A* mRNA abundance and the abundance of mRNAs involved in antigen presentation. Examples of correlations of MHC class I (Fig. 5h, i), MHC class II (Fig. 5j, k) and antigen processing and presentation (Fig. 5l, m) genes are shown. The Spearman correlation analyses of the full lists of these classes of genes are provided in Supplementary Data 4. Consistent with our mouse tumour data, Spearman correlation analyses revealed statistically significant negative correlations between *HIF2A* expression and the expression of 3 of 6 MHC class I genes, 12 of 15 MHC class II genes, and 26 of 51 non-MHC genes from the GO antigen processing and presentation gene set. While highly statistically significant, the relatively small magnitudes of some of these correlations suggest that *HIF2A* expression levels may be one of several factors that influence the overall expression of antigen presenting genes in ccRCC. Collectively the mice and human data suggest that HIF-2α suppresses antigen presentation.

**HIF-1α and HIF-2α alter the ccRCC immune microenvironment.** Since diverse and complex inflammatory signatures were observed in the transcriptomes and proteomes of ccRCC tumours, we next applied three different bioinformatic methods to our RNA-sequencing data to attempt to further deconvolve the relative abundance of different types of immune cells or specific gene signatures associated with inflammation in the different tumour genotypes. We applied two methods that were previously used to deconvolve the immune microenvironment of human ccRCC[42,43]. These methods are based on single-sample gene set enrichment analyses (ssGSEA) using the mouse orthologues of an expanded number of immune-specific gene signatures to those initially described by Bindea et al.[44], which we term Bindea et al. (described in ref. [43]), and a set of gene signatures that were identified by analyses of human ccRCC tumourgrafts, termed eTME (described in ref. [42]). The genes in these signatures and their overlap are listed in Supplementary Data 5. The third method is CIBERSORT with the mouse specific ImmuCC gene panel, which uses a matrix-weighted score, based on both high and low expressed genes in each immune subset, to assess the relative abundance of each immune

cell population[45]. For all three methods we generated *z*-scores representing the degree of enrichment of each signature and statistically compared the immune infiltration scores of all tumours of a given genotype in a series of pairwise comparisons to WT cortex and to the other genotypes (Fig. 6a). In general, comparisons of all three tumour genotypes to WT cortex revealed enrichment of gene sets associated with myeloid cell inflammation, including dendritic cells, monocytes, macrophages, and neutrophils, but not mast cells or eosinophils. Terms associated with different types of T cells and B cells revealed inconsistent results, varying depending on the deconvolution method used. $Vhl^{\Delta/\Delta}Trp53^{\Delta/\Delta}Rb1^{\Delta/\Delta}Hif2a^{\Delta/\Delta}$ tumours showed greater enrichment of a number of different immune cell signatures when compared to $Vhl^{\Delta/\Delta}Trp53^{\Delta/\Delta}Rb1^{\Delta/\Delta}$ tumours and to $Vhl^{\Delta/\Delta}Trp53^{\Delta/\Delta}Rb1^{\Delta/\Delta}Hif1a^{\Delta/\Delta}$ tumours, including several types of T cells, monocytes, and macrophages, and notably for interferon-γ signalling (REACTOME.IFNG), consistent with the previous GAGE analyses. $Vhl^{\Delta/\Delta}Trp53^{\Delta/\Delta}Rb1^{\Delta/\Delta}Hif1a^{\Delta/\Delta}$ tumours showed statistically lower average *z*-scores, or trends to lower scores, for myeloid cell signatures than $Vhl^{\Delta/\Delta}Trp53^{\Delta/\Delta}Rb1^{\Delta/\Delta}$ and $Vhl^{\Delta/\Delta}Trp53^{\Delta/\Delta}Rb1^{\Delta/\Delta}Hif2a^{\Delta/\Delta}$ tumours.

To further characterise the immune microenvironments of the three different tumour genotypes, we next conducted immuno-histochemical stainings for a series of markers of different types of immune cells to permit analyses of a larger set of tumours of each genotype ($n = 14$–26 tumours). We stained sections of whole tumour-bearing kidneys with antibodies against CD3 to label T cells, CD4 to label helper T cells, CD8 to label effector T cells, CD69 as an early activation marker of T cells and NK cells, perforin to label activated cytotoxic T cells and NK cells, PD-1 to label antigen-exposed activated or exhausted T cells, B220 to label B cells, CD68 to label monocytes and macrophages, F4/80 to label differentiated macrophages, and Ly-6G to label granulocytes and neutrophils. These markers revealed considerable inter-tumoural heterogeneity in terms of the density of infiltrating cells, even within the same kidney (Supplementary Fig. 11a–j). We quantified the densities of positively stained cells either by manual counting, via automated detection and quantification algorithms, or we calculated the average relative staining intensity for F4/80 where it was not possible to identify individual cells in the network of macrophages, within the tumours as well as in unaffected regions of kidney tissue (normal) within the same mouse (Supplementary Fig. 11k, l). Consistent with HIF-2α deficient tumours showing the highest GAGE mRNA signatures of T-cell inflammation, $Vhl^{\Delta/\Delta}Trp53^{\Delta/\Delta}Rb1^{\Delta/\Delta}Hif2a^{\Delta/\Delta}$ tumours displayed increased densities of CD3 (Fig. 6b), CD4 (Fig. 6c), and CD8 (Fig. 6d) positive T cells compared to normal tissue, whereas only CD8 positive T-cell densities were significantly increased in $Vhl^{\Delta/\Delta}Trp53^{\Delta/\Delta}Rb1^{\Delta/\Delta}$ and $Vhl^{\Delta/\Delta}Trp53^{\Delta/\Delta}Rb1^{\Delta/\Delta}Hif1a^{\Delta/\Delta}$ tumours compared to the respective normal tissues. Notably, both $Vhl^{\Delta/\Delta}Trp53^{\Delta/\Delta}Rb1^{\Delta/\Delta}Hif1a^{\Delta/\Delta}$ and $Vhl^{\Delta/\Delta}Trp53^{\Delta/\Delta}Rb1^{\Delta/\Delta}Hif2a^{\Delta/\Delta}$ tumours exhibited higher densities of CD8 positive T cells than $Vhl^{\Delta/\Delta}Trp53^{\Delta/\Delta}Rb1^{\Delta/\Delta}$ tumours, in line with the Bindea et al. ssGSEA CD8 T-cell signature results. $Vhl^{\Delta/\Delta}Trp53^{\Delta/\Delta}Rb1^{\Delta/\Delta}Hif1a^{\Delta/\Delta}$ but not $Vhl^{\Delta/\Delta}Trp53^{\Delta/\Delta}Rb1^{\Delta/\Delta}Hif2a^{\Delta/\Delta}$ tumours showed increased densities of CD4 positive cells compared to $Vhl^{\Delta/\Delta}Trp53^{\Delta/\Delta}Rb1^{\Delta/\Delta}$ tumours. This result is not reflected by any of the bioinformatic immune deconvolution methods. Analyses of the T-cell activation markers CD69 (Fig. 6e) and perforin (Fig. 6f) revealed that all tumours showed increased T-cell activation compared to normal tissue, and that $Vhl^{\Delta/\Delta}Trp53^{\Delta/\Delta}Rb1^{\Delta/\Delta}Hif2a^{\Delta/\Delta}$ tumours showed higher levels of T-cell activation than $Vhl^{\Delta/\Delta}Trp53^{\Delta/\Delta}Rb1^{\Delta/\Delta}$ or $Vhl^{\Delta/\Delta}Trp53^{\Delta/\Delta}Rb1^{\Delta/\Delta}Hif1a^{\Delta/\Delta}$ tumours, consistent with our conclusions from the GAGE analyses. There was no statistically significant enrichment of PD-1 positive cells, a marker of

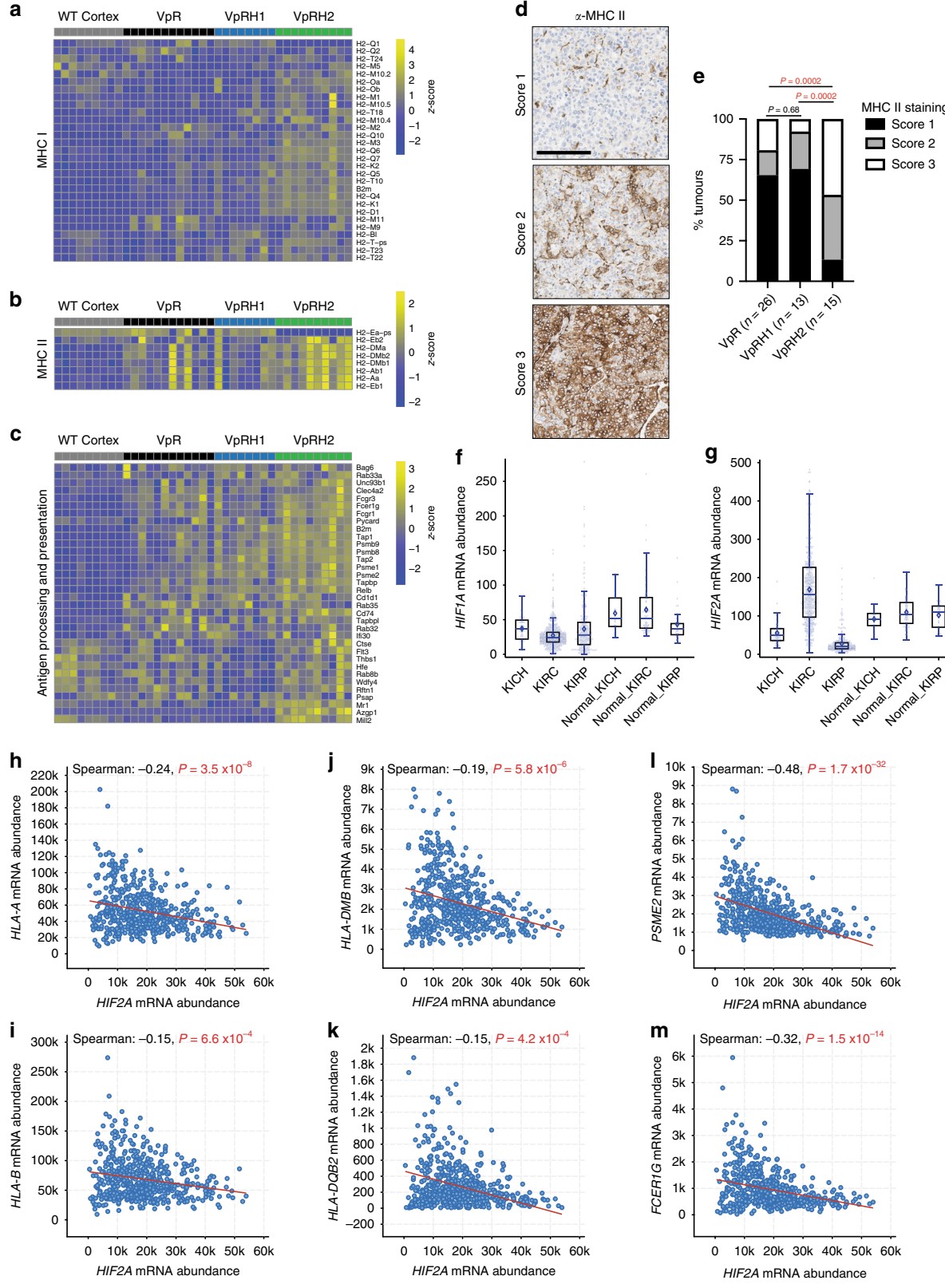

exhausted T cells, in any of the tumour genotypes (Fig. 6g). B220 staining revealed increased B cell density in all tumour genotypes compared to normal tissue, and higher densities in Vhl$^{\Delta/\Delta}$Trp53$^{\Delta/\Delta}$Rb1$^{\Delta/\Delta}$Hif1a$^{\Delta/\Delta}$ tumours than in Vhl$^{\Delta/\Delta}$Trp53$^{\Delta/\Delta}$Rb1$^{\Delta/\Delta}$ or Vhl$^{\Delta/\Delta}$Trp53$^{\Delta/\Delta}$Rb1$^{\Delta/\Delta}$Hif2a$^{\Delta/\Delta}$ tumours (Fig. 6h). Interestingly, these observations do not reflect the

results of the bioinformatic immune cell deconvolutions for B cells. In contrast to the relatively low numbers of tumour-infiltrating lymphocytes, myeloid lineage cells were much more abundant within tumours. CD68 positive monocytes/macrophages (Fig. 6i), F4/80 positive macrophages (Fig. 6j), and Ly-6G-labelled granulocytes/neutrophils (Fig. 6k) were highly abundant

**Fig. 5 HIF-2α influences the expression of MHC class I and II genes.** Gene expression heatmaps for MHC class I (**a**), class II (**b**), and other antigen processing and presenting (**c**) genes. Rows represent row-normalised z-scores of mRNA abundance, each column represents an individual sample from WT cortex or VpR, VpRH1, VpRH2 tumours. Source data is provided in Supplementary Data 1 and 2. **d** Examples of different scores for MHC class II immunohistochemical staining. All panels are the same magnification, scale bar = 100 μm. **e** Distribution of MHC class II staining scores in the indicated (*n*) number of VpR, VpRH1, and VpRH2 tumours. *P* values are derived from the two-sided Mann–Whitney *U* test without adjustments for multiple comparisons. **f, g** Relative *HIF1A* and *HIF2A* mRNA abundance in TCGA datasets of human chromophobe (KICH, *n* = 66), clear cell (KIRC, *n* = 533), and papillary (KIRP, *n* = 290) renal cell carcinomas and associated normal renal tissues (Normal_KICH *n* = 25, Normal_KIRC *n* = 72, Normal_KIRP *n* = 32). Box–whisker plots depict median, bounded by Q1 (25% lower quartile) and Q3 (75% upper quartile) and whiskers depict 1.5 times the Q3–Q1 interquartile range. Spearman's correlation analyses between *HIF2A* mRNA abundance and mRNA abundance of two MHC class I (**h, i**), class II (**j, k**), and antigen processing/presentation (**l, m**) genes in ccRCC (TCGA KIRC dataset). Source data is provided in Supplementary Data 4.

in tumours compared to normal tissue but there were no differences in abundance of these cells between tumour genotypes.

Given that HIF-1α and HIF-2α deficiencies increase CD8 positive T-cell infiltration, we first sought to gain insight into potential mechanisms that might explain these observations by examining our RNA-sequencing data. We previously demonstrated that Vhl$^{\Delta/\Delta}$Trp53$^{\Delta/\Delta}$Rb1$^{\Delta/\Delta}$ tumours show upregulation of numerous cytokines in comparison to WT cortex[16], however analyses of the new dataset did not reveal any cytokines that were specifically altered by the absence of HIF-1α or HIF-2α (Supplementary Fig. 12a) that might be expected to influence T-cell infiltration or activation. To functionally investigate whether signalling molecules or metabolic factors that are released by mouse or human ccRCC cells might directly influence the proliferation of CD8$^+$ T cells, we activated mouse splenic CD8+ T cells and incubated them for 3 days in conditioned medium from mouse 2020 ccRCC cells (including VHL30 rescue and *Hif1a* knockdown cells), human renal proximal tubule epithelial cells (RPTEC), human A498 ccRCC cells or human 786-O (including VHL30 rescue) ccRCC cells, compared to non-conditioned medium. However, none of the conditioned media altered CD8$^+$ T-cell proliferation (Supplementary Fig. 12c, d), arguing against a direct, soluble factor-mediated, VHL- or HIF-α-dependent cross-talk between ccRCC and T cells as being the mechanism that underlies the altered immune microenvironment in the Vhl$^{\Delta/\Delta}$Trp53$^{\Delta/\Delta}$Rb1$^{\Delta/\Delta}$Hif1a$^{\Delta/\Delta}$ and Vhl$^{\Delta/\Delta}$Trp53$^{\Delta/\Delta}$Rb1$^{\Delta/\Delta}$Hif2a$^{\Delta/\Delta}$ tumours.

To investigate whether genetic alterations in *HIF1A* or *HIF2A* might influence immune cell infiltration in human ccRCC, we analysed data from the TCGA KIRC study (Firehose-legacy dataset)[9] using cBioPortal[46,47]. ccRCC tumours frequently lose one copy of *HIF1A* and less frequently gain one copy of *HIF2A* (Fig. 7a). Loss of one copy of either gene correlated with lower mRNA levels but gain of a copy did not correlate with increased mRNA abundance (Supplementary Fig. 13a, b). ccRCC tumours that exhibit mono- or bi-allelic loss of *HIF1A* (collectively *HIF1A* loss) show worse overall survival (Fig. 7b) and progression-free survival (Supplementary Fig. 13c) than unaffected tumours, whereas there are no overall or progression-free survival differences between tumours with a copy number gain of *HIF2A* and unaffected tumours (Fig. 7c and Supplementary Fig 13d). Previous studies have identified that loss of larger regions of chromosome 14q, including the *HIF1A* gene, correlates with poor prognosis[19]. We took advantage of the extensive clinical and whole-exome sequencing data of the TCGA dataset to investigate whether co-variants of *HIF1A* loss may account for the observed survival differences. Tumours with *HIF1A* loss were statistically more likely to have higher grade and stage, and display lymph node positivity and metastases (Supplementary Table 1), consistent with this subgroup representing a more aggressive form of ccRCC. The only mutation that occurred more frequently in the *HIF1A* loss subgroup than the unaltered subgroup was *BAP1*,

which was detected in 9% of all ccRCC tumours (Supplementary Fig. 13e, f). However, *BAP1* mutation status alone did not significantly affect survival (Supplementary Fig. 13g) in this cohort and removal of all *BAP1* mutant tumours from the cohort did not alter the correlation of *HIF1A* loss with poor prognosis (Supplementary Fig. 13h, i). The conclusion that *BAP1* mutation status is not a relevant co-variant that affects survival outcome in the *HIF1A* loss cohort was also demonstrated by COX univariate (HR 1.839, 95% CI 1.332–2.539, *P* = 0.00021) and multivariate proportional hazards analyses (HR 1.776, 95% CI 1.274–2.474, *P* = 0.00069). These findings suggest that loss of one allele of *HIF1A*, which is predicted to lead to diminished HIF-1α abundance, may be selected for during the evolution of some ccRCC tumours and that this correlates with aggressive disease.

To investigate whether *HIF1A* loss correlates with altered inflammation, we first demonstrated that *HIF1A* loss tumours exhibit on average between 1.9- and 2.1-fold higher levels of mRNA of *CD3D, CD3E, CD8A,* and *CD8B* and 1.4-fold higher levels of *CD4* than unaltered tumours (Fig. 7d, f, h, j, l), suggesting that this group of tumours has higher CD8$^+$, and to a lesser extent CD4$^+$, T-cell infiltration. This observation is consistent with the mouse analyses in which *Hif1a*-deficient tumours on average display approximately double the number of CD8$^+$ and CD4$^+$ T cells. In contrast, *HIF2A* gain tumours show no differences in the expression of any of these T-cell marker genes compared to unaltered tumours (Fig. 7e, g, i, k, m). To gain a more in-depth overview of the effects of *HIF* gene copy number or expression level alterations on the immune microenvironment, we performed immune deconvolution analyses of RNA-seq data, again using three independent methods of immune cell deconvolution; ssGSEA using the Bindea et al. and eTME gene signatures and using the CIBERSORT method[48]. We compared *HIF1A* loss and *HIF2A* gain tumours to diploid human ccRCCs (Fig. 7n) and also took advantage of the wide distribution of mRNA expression levels of *HIF2A* to compare tumours in the top (Q4) and bottom (Q1) quartiles of *HIF2A* mRNA abundance. While *HIF2A* gain tumours exhibited very few alterations in immune scores, *HIF1A* loss tumours showed statistically significant increases or decreases in 57 of 83 immune signatures of a variety of lymphoid and myeloid lineage cells. Notable amongst these are consistently upregulated scores for T helper cells and for B cells, mirroring our immunohistochemical findings of the comparison of Vhl$^{\Delta/\Delta}$Trp53$^{\Delta/\Delta}$Rb1$^{\Delta/\Delta}$ and Vhl$^{\Delta/\Delta}$Trp53$^{\Delta/\Delta}$Rb1$^{\Delta/\Delta}$Hif1a$^{\Delta/\Delta}$ tumours. It should however be noted that the magnitude of the z-scores are generally low, suggesting that this group of ccRCCs on average exhibits numerous subtle differences in immune inflammation compared to tumours with normal *HIF1A* copy number. In contrast, high *HIF2A* mRNA expressing tumours displayed downregulation of scores for interferon-γ and for APM2 (measuring MHC class II antigen presentation), consistent with the upregulation of these features in mouse ccRCC tumours lacking HIF-2α. Somewhat paradoxically, high *HIF2A* expressing tumours also displayed general

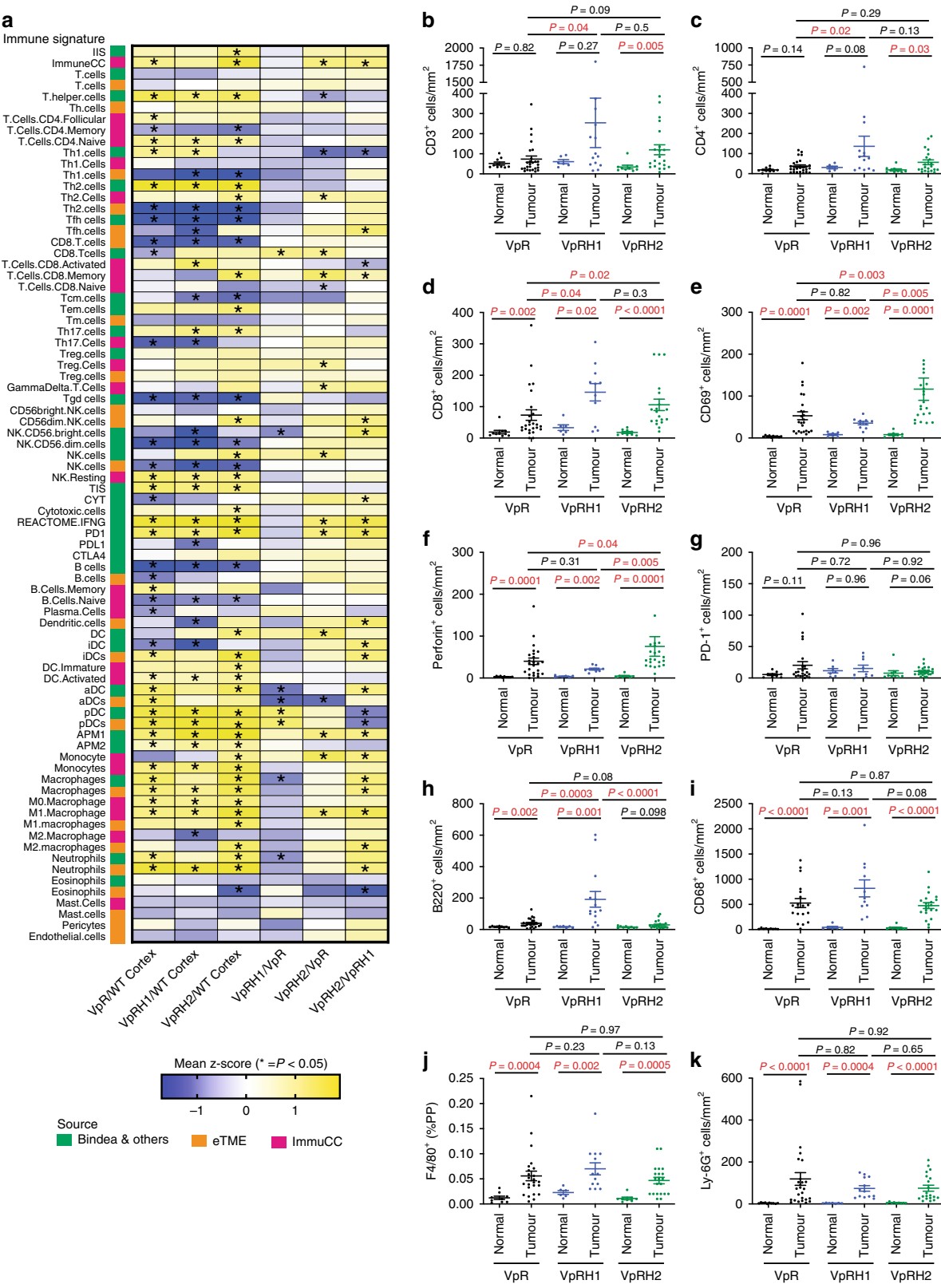

upregulation of CD8 T-cell scores and some NK cell scores and downregulation of all three scores for regulatory T cells and for the immune checkpoint proteins PD-1 and CTLA4. These scores might be predicted to reflect elevated CD8 T-cell activity. However, these tumours also display elevated scores for monocytes, neutrophils, and mast cells, which in some settings contribute to suppression of anti-tumour CD8 T-cell responses. Immunosuppressive mast cells were shown to correlate with *HIF2A* mRNA abundance in human ccRCC[49]. Thus, the extent of T cell mediated anti-tumour immunity is likely to be determined by the balance of the abundance and activities of several different immune cell types in a manner that is partly influenced by *HIF2A*

**Fig. 6 Deconvolution of the immune microenvironment of VpR, VpRH1, and VpRH2 tumours. a** Summary of immune deconvolution results using ssGSEA with the Bindea et al. and eTME gene signatures, as well as the ImmuCC method. Pairwise comparisons of the expression levels of each immune cell-specific gene set between WT cortex, VpR, VpRH1, and VpRH2 tumours are shown. Columns depict the comparison between the genotypes and rows depict the gene set. Heatmap colours represent the mean differences in the $z$-scores. Comparisons marked with an asterisk show $P$ values < 0.05 between each genotype, two-sided Mann–Whitney $U$ test without multiple comparisons. Gene signatures and source data with $z$-scores and $P$ values are provided in Supplementary Data 5 and 6, respectively. **b–k** Quantification of the densities of immunohistochemically positive cells stained with the indicated antibodies in unaffected normal renal tissue and VpR ($n = 26$), VpRH1 ($n = 14$), and VpRH2 ($n = 21$) tumours. Mean ± SEM are shown, $P$ values for pairwise comparisons were calculated by one-way ANOVA followed by two-sided Mann–Whitney $U$ test without adjustments for multiple comparisons.

expression. Finally, it is also noteworthy that *HIF1A* copy loss and *HIF2A* mRNA high tumours showed opposite effects on signatures for pericytes, endothelial cells and angiogenesis, implying that both HIF-1α and HIF-2α may act as positive factors that promote blood vessel formation in ccRCC tumours.

Collectively, the mouse and human analyses demonstrate that the genetic copy number or expression level status of the *HIF* genes in ccRCC tumour cells correlate with the composition and activation state of the innate and adaptive immune systems in the tumour microenvironment.

## Discussion

Herein we show by introducing floxed alleles of *Hif1a* or *Hif2a* into an autochthonous mouse model of ccRCC that HIF-1α is necessary for tumour formation whereas HIF-2α deficiency has only a moderate effect on tumour initiation and growth. While it cannot be excluded that there might be differences between mice and humans, and the findings may be contextual to the *Vhl/Trp53/Rb1* mutant background, this result seemingly contrasts with several independent lines of evidence from the study of human ccRCC tumours and ccRCC cell lines, which have demonstrated that HIF-2α possesses strong oncogenic activity and HIF-1α acts in the manner of a tumour suppressor to suppress aggressive tumour behaviour. How can these apparent discrepancies be reconciled? There are several possible explanations which suggest that these observations may not in fact be discrepancies. In addition, our current study also suggests that there may be caveats to the interpretation of previous studies, arguing against an oversimplified, binary oncogene/tumour suppressor model of the contributions of these proteins to ccRCC development. We argue that HIF-1α and HIF-2α are likely to play different roles at different stages of tumour formation and progression and that these roles might potentially be modulated by the spectrum of mutations present in each individual ccRCC. We further argue that it is possible that the balance of the relative strengths of the activities of the different HIF-α proteins might be dynamically modulated via different mechanisms throughout the lifetime of a ccRCC tumour to tailor their combined transcriptional outputs to provide an evolutionary advantage at the given stage of the tumour.

In addition to genetic mechanisms that can irreversibly affect the balance of HIF-1α and HIF-2α activities, such as loss of one copy of chromosome 14q, encoding *HIF1A*, or intragenic deletions of *HIF1A*[22], several mechanisms exist that potentially provide tumour cells with a more dynamic mode of fine-tuning the relative strengths of HIF-1α and HIF-2α expression and activities. These include mutual epigenetic suppression[50], mutual suppression of protein levels[23], HIF-2α-mediated suppression of HIF-1α translation[51] and HAF-mediated ubiquitination and degradation of HIF-1α to promote the switch from HIF-1α towards HIF-2α activity[52]. Analyses of expression patterns in human ccRCC support the notion that HIF-1α plays an oncogenic role at early and late stages of ccRCC development and progression. In VHL patients, the earliest *VHL*-null multi-cellular renal tubule lesions

tend to strongly express HIF-1α and weakly express HIF-2α, but later lesions such as cysts and tumours express HIF-1α as well as higher levels of HIF-2α[23,53]. While HIF-2α protein is present in the vast majority of all sporadic ccRCC tumours, HIF-1α protein expression is not detected in about 30% of cases and this correlates with increased tumour cell proliferation[21]. Nonetheless, HIF-1α is detected in about 70% of ccRCC tumours and several studies have correlated higher HIF-1α expression levels with poor patient survival (reviewed in ref. [54]). Consistent with our current findings of an obligate oncogenic role of HIF-1α, transgenic overexpression of constitutively stabilised HIF-1α[55], but not HIF-2α[56] in mouse renal tubules causes the formation of small lesions that have some features of precursor lesions of ccRCC. Interestingly, combined overexpression of HIF-1α and HIF-2α did not cause a more severe phenotype than HIF-1α overexpression alone[56], demonstrating that even the combined actions of both HIF-1α or HIF-2α are insufficient to induce tumour formation. In agreement with this conclusion, numerous previous studies showed that deletion of *Vhl* in mouse renal epithelial cells in vivo[10,57–62], resulting in abrogation of the many different tumour suppressor functions of pVHL[2,7,63], was insufficient to cause tumour initiation either when both HIF-1α and HIF-2α were stabilised, or when the balance of HIF-1α and HIF-2α activities were genetically altered by co-deletion of *Hif1a* or *Hif2a*[29]. Nonetheless, deletion of either *Hif1a* or *Hif2a* was sufficient to inhibit the formation of cysts and tumours induced by *Vhl/Trp53* double mutation[29], demonstrating that both HIF-1α and HIF-2α have pro-tumourigenic activities. In this context, it is noteworthy that we now show that *Hif2a* deletion fails to strongly inhibit tumour formation in the *Vhl/Trp53/Rb1* deletion model. One explanation for these different findings may relate to our RNA-sequencing observations which highlighted that HIF-2α increases expression of MYC and E2F target genes, consistent with previous findings that HIF-2α stimulates MYC activity[21]. This predicted cell cycle promoting activity of HIF-2α is likely to be necessary for tumour formation in the *Vhl/Trp53* background, but may become at least partly redundant due to the additional mutation of *Rb1*, which promotes the cell cycle by removing the negative regulation of E2F transcription factors.

The patterns of copy number alterations that arise in the TGCA ccRCC dataset are consistent with the idea that the balance of HIF-1α and HIF-2α activities may be differently selected for, or tolerated, depending on the status of the network of G1-S cell cycle controlling genes. We have previously described a copy number signature in which multiple genes involved in the p53 pathway and G1-S checkpoint are altered in about two thirds of ccRCC tumours[16]. While *HIF1A* copy number losses or mutations (45%) occur in ccRCC more frequently than gains (2.5%), *HIF2A* losses are very rare (4%), and gains more common (15%) (Supplementary Fig. 14a). Importantly, of the rare tumours with *HIF2A* losses or mutations, only 2 of 16 did not exhibit a copy number alteration in one or more genes that control the G1-S checkpoint (Supplementary Fig. 14b). Thus, rare genetic events that might be predicted to result in lowered HIF-2α activity almost always arise in the background of genetic alterations that

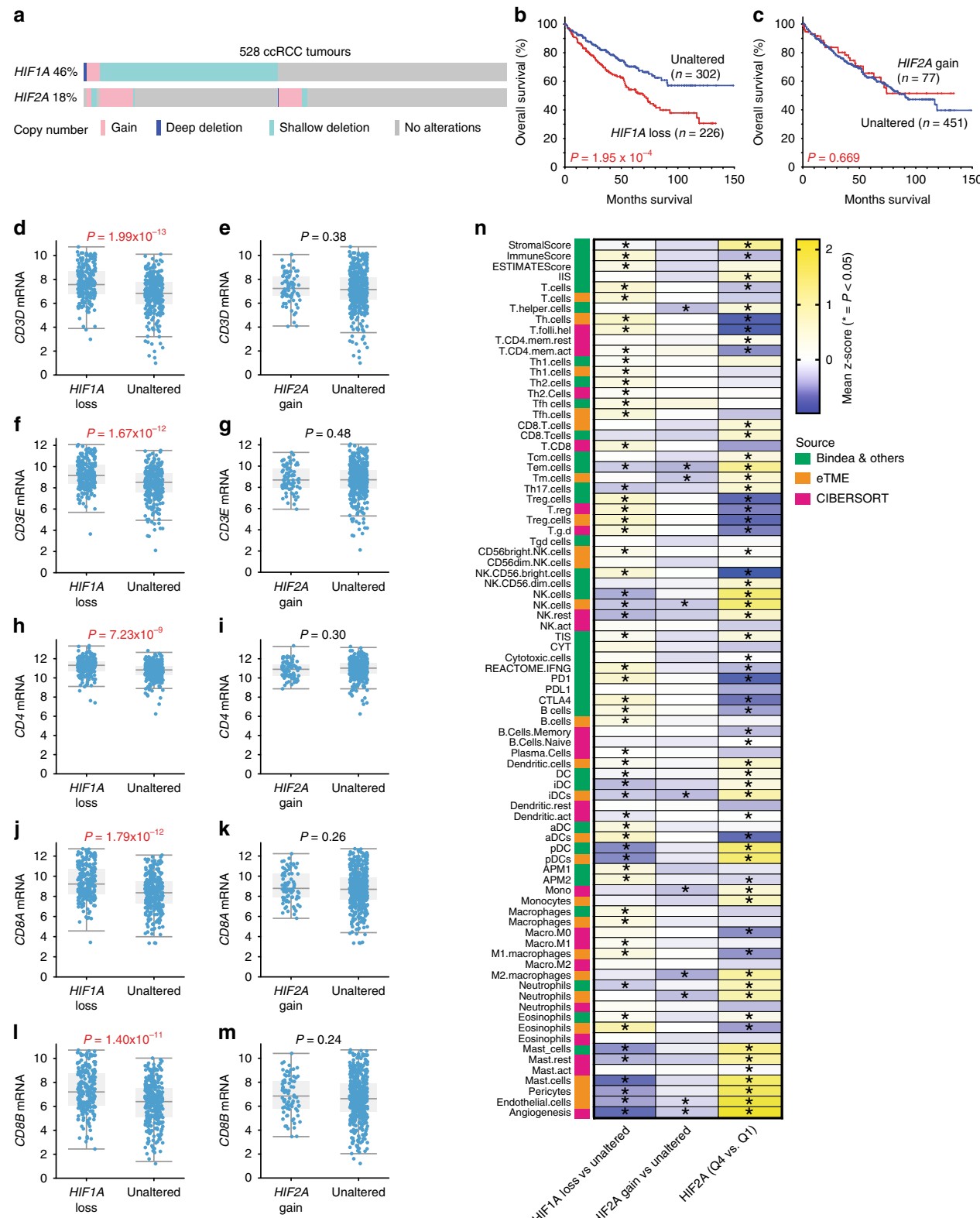

are predicted to abrogate or weaken normal cell-cycle control. We speculate that this may allow the tumours to grow in the absence of a lowered cell cycle promoting activity of HIF-2α. In contrast, the distribution of copy number losses of *HIF1A* does not correlate with the presence or absence of alterations in the G1-S network: 52 tumours harbouring *HIF1A* copy loss showed no alteration in G1-S genes, while 146 tumours with *HIF1A* copy loss did display alterations (Supplementary Fig. 14c).

Since our current and previous mouse genetic studies collectively show that the requirement for HIF-1α and HIF-2α is dependent on the underlying genotype of the tumour (*Vhl/Trp53* mutations versus *Vhl/Trp53/Rb1* mutations), it will be important to test the generality of our findings in other genetic mouse models of ccRCC, particularly as epigenetic modifications resulting from mutations in *Pbrm1* have been shown to alter HIF-α transcriptional outputs[12,64,65]. It is likely that other ccRCC-

**Fig. 7 HIF1A copy number loss and HIF2A mRNA expression levels correlate with altered immune microenvironments in human ccRCC. a** Oncoprint showing the genetic alterations in HIF1A and HIF2A in human ccRCC tumours based on GISTIC. **b, c** Kaplan–Meier curves showing overall survival of ccRCC patients whose tumours exhibit loss of one or two copies of HIF1A (loss) or gain of a copy of HIF2A (gain) versus patients without these copy number alterations (Unaltered). P values are derived from the two-sided log-rank test. mRNA abundance (log$_2$ transformed, normalised, RNA-seq v2 RSEM) of CD3D (**d, e**), CD3E (**f, g**), CD4 (**h, i**), CD8A (**j, k**), CD8B (**l, m**) in HIF1A loss and HIF2A gain ccRCC tumours compared to Unaltered tumours. P values are derived from two-sided Student's t test. **n** Summary of immune deconvolution results in ccRCC (TCGA KIRC dataset) using ssGSEA with the Bindea et al. and eTME gene signatures, as well as the CIBERSORT method. Depicted are pairwise comparisons of the expression levels of each gene set between HIF2A gain (n = 65) and HIF2A diploid (Unaltered, n = 349), between HIF1A loss (n = 188) and HIF1A diploid (Unaltered, n = 220) tumours and between tumours in the top quartile (Q4) and the lowest quartile (Q1) of HIF2A expression. Columns depict the comparison between the genotypes and rows depict the gene set. Heatmap colours represent the mean differences in the z-scores. Comparisons marked with an asterisk show P values < 0.05 between each genotype, two-sided Mann–Whitney U test without multiple comparisons. Source data with z-scores and P values is provided in Supplementary Data 6.

relevant epigenetic tumour suppressors such as BAP1, SETD2, and KDM5C also influence HIF-α transcriptional outputs. Thus, the combinations of mutations present in ccRCC cells might represent another mechanism for altering the balance of the relative activities of HIF-1α and HIF-2α, potentially affecting the genetic dependency on the two HIF-α proteins.

Our present study also highlights a potential general caveat to the interpretation of the results of studies of human ccRCC cell lines, or patient-derived tumour lines, in cell culture and in xenograft assays. In contrast to the clear requirement for Hif1a in tumour formation in the autochthonous setting, we find that HIF-1α rather exhibits putative tumour suppressor activities in cell culture-based assays, including inducing the early loss of proliferative capacity of MEFs following Vhl deletion, inhibiting the proliferation of immortalised Vhl/Trp53 null MEFs and inhibiting anchorage independent growth of a Vhl/Trp53/Rb1 mutant mouse ccRCC cell line. Furthermore, HIF-1α knockdown did not affect the growth of allograft tumours generated with this ccRCC cell line. This latter experiment argues against the idea that removal of HIF-1α activity in an established mouse ccRCC tumour, to mimic the loss of HIF-1α function that arises as a later event in tumour progression in a subset of human ccRCCs, has a potent effect on tumour aggressiveness. Collectively, our studies show that in the same genetic ccRCC tumour model, the cell culture and allograft assays do not reflect the in vivo requirement in the autochthonous setting. Previous studies which have interpreted the oncogenic and tumour suppressive roles of HIF-α factors in ccRCC based on studies of cultured ccRCC cells and xenograft models may therefore not necessarily be reflective of the true in vivo functions of HIF-1α and HIF-2α at different stages of tumour development in the physiological context of the kidney.

Our transcriptomic and proteomic analyses suggest that one potential oncogenic contribution of HIF-1α to tumour initiation is the induction of expression of genes that induce Warburg metabolism of high rates of conversion of glucose to lactate and low rates of pyruvate entry into mitochondrial respiration. These findings of HIF-1α dependency are in accordance with numerous studies in mice and humans[29,66–68]. This mode of Warburg tumour metabolism has recently been proven to occur in vivo in human ccRCC tumours[69] and is in stark contrast to the normal mode of metabolism of RPTECs, the cell of origin of ccRCC, which generate ATP predominantly through mitochondrial oxidation of substrates other than glucose, allowing them to fuel solute transport, including the transport of glucose, from the tubular fluid back to the blood stream[70,71]. It remains to be functionally elucidated if and how this shift to glucose metabolism contributes to ccRCC formation by proximal tubule cells and whether this represents a true driver of tumour formation or acts as an enabler of tumour formation once additional transforming genetic mutations arise. Previously published mouse genetic data suggest that the latter is likely to be the case as induction of

Warburg metabolism in mice through Vhl deletion alone did not cause tumour formation. We identified that both HIF-1α and HIF-2α are necessary for the clear cell phenotype of Vhl/Trp53/Rb1-deficient ccRCCs, implying that they might differently contribute to the accumulation of cytoplasmic glycogen and lipids. HIF-1α-dependent alterations in glucose and glycogen utilisation and reduction of mitochondrial abundance and activity[29] might lead to accumulation of glycogen and failure to efficiently metabolise fatty acids. We also identified a HIF-2α-dependent metabolic signature involving genes involved in cholesterol and lipoprotein uptake and metabolism, which may account for the requirement of HIF-2α for the clear cell phenotype.

A major advantage of the study of autochthonous models of ccRCC is that the tumours develop in the presence of a functioning immune system. In general, the activities of cytotoxic innate and adaptive immune cells can act to inhibit or delay tumour formation and progression, but an inflamed tumour microenvironment can also promote tumour formation through a variety of mechanisms[72]. The immune microenvironment of human ccRCC appears to be unusual in some respects as higher levels of T-cell infiltration have been shown to correlate with disease recurrence and worse survival[43,73,74]. In almost all other tumour types the degree of T-cell inflammation is a good prognostic factor[14]. ccRCC tumours are also frequently inflamed with immature myeloid lineage cells that at least partly reflect different states of differentiation along the path from myeloid progenitor to differentiated dendritic cells, macrophages, and neutrophils. At least some of these myeloid lineage cells have been proposed to promote tumour formation by suppressing T-cell activation[75–77]. Our bioinformatic immune deconvolution and immunohistochemical analyses of the immune microenvironment of mouse ccRCC tumours is broadly consistent with this latter observation, namely that the immune microenvironment is dominated by myeloid lineage cells with a modest degree of T-cell inflammation and activation. An unexpected finding was that tumours with Hif2a deletion showed stronger mRNA signatures associated with tumour immune cell infiltration, antigen presentation, and interferon activity, as well as higher densities of CD8 T cells and cells expressing the T-cell activation markers CD69 and perforin, compared to the other two tumour genotypes. Furthermore, MHC class I and II genes, as well as other genes involved in antigen processing and presentation, were upregulated and tumour cells in Vhl$^{\Delta/\Delta}$Trp53$^{\Delta/\Delta}$Rb1$^{\Delta/\Delta}$Hif2a$^{\Delta/\Delta}$ tumours were more frequently immunohistochemically positive for MHC class II expression, suggestive of increased presentation of intracellular and extracellular epitopes in this tumour genotype. Collectively, these observations reflect a generally higher degree of immune activity directed against Hif2a-deficient tumour cells and it is plausible that this immune control may at least partly contribute to the reduced numbers of tumours and delayed tumour onset seen in this genotype. Interestingly, we demonstrated not only in Vhl$^{\Delta/\Delta}$Trp53$^{\Delta/\Delta}$Rb1$^{\Delta/\Delta}$Hif2a$^{\Delta/\Delta}$ tumours but also in Vhl$^{\Delta/\Delta}$

Trp53$^{\Delta/\Delta}$Rb1$^{\Delta/\Delta}$Hif1a$^{\Delta/\Delta}$ tumours that the density of intra-tumoural CD8$^+$ T cells were approximately double the density in Vhl$^{\Delta/\Delta}$Trp53$^{\Delta/\Delta}$Rb1$^{\Delta/\Delta}$ tumours. The densities of CD8$^+$ T cells observed in the mouse ccRCC tumours (~40–80 cells/mm$^2$) are within the range seen in most cases of human ccRCC (~20–160 cells/mm$^2$)[49], providing support for the relevance of our mouse model. Consistent with our findings of HIF-2α acting as a suppressor of T-cell inflammation in mouse ccRCC, it was shown that HIF-2α expression levels in human ccRCCs anti-correlate with T-cell abundance and markers of T-cell activation[49]. We speculate that this apparent suppression of anti-tumour immune responses by HIF-2α might reflect a mechanism of positive selection that maintains HIF-2α expression in ccRCCs. Moreover, we show that loss of one copy of *HIF1A* correlates with a worse survival outcome, higher mRNA signatures of T-cell abundance and a broadly altered immune microenvironment in human ccRCCs. This observation is consistent with the fact that higher levels of T-cell infiltration have been shown to correlate with disease recurrence and worse survival in ccRCC[43,73,74].

Thus, genetic and likely transcriptional and translational mechanisms that alter the balance of HIF-1α and HIF-2α abundance and activities appear to affect T-cell inflammation. The mechanisms that underlie the increased T-cell infiltration and/or activity in the absence of HIF-1α or HIF-2α will require further study. Analyses of RNA sequencing revealed that the mRNA levels of many cytokine-encoding genes are upregulated in tumours compared to normal cortex, but none of these genes were differentially regulated by HIF-1α or HIF-2α. Our functional tests to investigate if T-cell proliferation might be suppressed by ccRCC cells in general, for example through the secretion of immunosuppressive glycolytic metabolic products such as lactate or H$^+$[78], did not reveal any cross-talk between mouse or human ccRCC cells and mouse CD8$^+$ T cells under cell culture conditions. It is possible that these simplified assays failed to reproduce the metabolic conditions that are present in vivo. It is also likely that many other complex factors such as the presence and activation states of other immune microenvironmental cells (such as macrophages, MDSCs, dendritic cells, and regulatory T cells), antigen presentation and the presence or absence of co-activating or inhibitory ligands by ccRCC cells, as well as the composition of the extracellular matrix might all play a role in the trafficking and activation of T cells. Indeed, immune deconvolution analyses revealed that the group of tumours that exhibit loss of one copy of *HIF1A* or that show high levels of mRNA expression of *HIF2A* show many differences in signatures of different types of cells in the immune microenvironment. Since our own studies comparing multiple bioinformatic immune cell deconvolution methods showed that different methods and different signatures can lead to different results for the same immune cell type, as well as the fact that we observed both consistencies and inconsistencies between bioinformatic predictions and direct immune cell enumeration using immunohistochemistry, it will be important to treat these bioinformatic observations as hypothesis-generating starting points that will need to be orthogonally tested by staining for specific immune cell markers in larger cohorts of human ccRCC tumours. Given the clinical importance of immune checkpoint-based therapies for ccRCC and the fact that not all patients respond to these regimes, we believe that further investigation of the relationship between HIF-1α and HIF-2α status and the immune microenvironment is of potential therapeutic relevance. A further corollary of our findings is that inhibition of HIF-1α and HIF-2α transcription factor activities in ccRCC cells could be investigated therapeutically to inhibit tumour cell proliferation and simultaneously to attempt to increase T-cell infiltration and activation. The potential direct effects of pharmacological inhibition of HIF-α factors on different immune cells would also have to be considered in this strategy. Finally, while specific inhibitors of HIF-2α are available and are currently being tested in clinical trials[25–27], our findings demonstrating the importance of HIF-1α for ccRCC formation argue that the development of specific inhibitors of HIF-1α or of new specific dual HIF-1α/HIF-2α inhibitors would also be desirable and may have therapeutic benefit in ccRCC. Proof-of-principle that these approaches are likely to be tolerable and effective comes from previously reported therapeutic effects of Acriflavine, which inhibits the binding of both HIF-1α and HIF-2α to HIF-1β[79,80], in the Vhl$^{\Delta/\Delta}$Trp53$^{\Delta/\Delta}$Rb1$^{\Delta/\Delta}$ ccRCC model[16], as well as in xenograft and autochthonous mouse models of several different types of tumours[79,81–83].

## Methods

**Mice.** Previously described Ksp1.3-CreER$^{T2}$;*Vhl$^{fl/fl}$;Trp53$^{fl/fl}$;Rb1$^{fl/fl}$* mice[16] were intercrossed with previously described Ksp1.3-CreER$^{T2}$;*Vhl$^{fl/fl}$;Trp53$^{fl/fl}$;Hif1a$^{fl/fl}$* and Ksp1.3-CreER$^{T2}$;*Vhl$^{fl/fl}$;Trp53$^{fl/fl}$;Hif2a$^{fl/fl}$* mice[29] to generate the experimental Ksp1.3-CreER$^{T2}$;*Vhl$^{fl/fl}$;Trp53$^{fl/fl}$;Rb1$^{fl/fl}$*, Ksp1.3-CreER$^{T2}$;*Vhl$^{fl/fl}$;Trp53$^{fl/fl}$;Rb1$^{fl/fl}$ Hif1a$^{fl/fl}$* and Ksp1.3-CreER$^{T2}$;*Vhl$^{fl/fl}$;Trp53$^{fl/fl}$;Rb1$^{fl/fl}$Hif2a$^{fl/fl}$* mouse lines. Littermate mice that lacked the Cre transgene served as WT controls. Gene deletion in 6-week-old mice was achieved by feeding with food containing tamoxifen (400 parts per million) for 2 weeks. Mouse crosses and phenotyping were conducted under the breeding license of the Laboratory Animal Services Center, University of Zurich and tumour monitoring studies were conducted under license ZH116/16 of the Canton of Zurich. Investigators were not blinded to the genotype of the mice.

**µCT imaging.** Monitoring of tumour growth in mice was performed on a monthly basis using µCT as previously described[16]. Tumour size was assessed by measuring the maximum diameter of all three dimensions in the respective planes (x, y, and z-plane). The volumes were then calculated using the mathematical formula of an ellipsoid:

$$V = \frac{4}{3} \times \pi \times \text{radius}(x) \times \text{radius}(y) \times \text{radius}(z). \quad (1)$$

**Generation of MEFs and mouse ccRCC cell line.** Isolation of MEF lines[29] and preparation of primary renal epithelial cells[15] have been previously described. MEFs were infected with adenovirus expressing Cre recombinase and GFP (Ad-Cre-GFP; Vector Biolabs; #1700) or GFP only (Ad-CMV-GFP; Vector Biolabs; #1060). The mouse ccRCC cell line 2020 was isolated from a piece of tumour tissue from a Vhl$^{\Delta/\Delta}$Trp53$^{\Delta/\Delta}$Rb1$^{\Delta/\Delta}$ mouse, minced with a scalpel blade and digested for 70 min at 37 °C with 1 mg/ml collagenase II solution in 1× HBSS. The digestion was inactivated with 20 ml K-1 medium (Dulbecco's modified Eagle's medium) and Hams F12 mixed 1:1, 2 mM glutamine, 10 kU/ml penicillin, 10 mg/ml streptomycin, hormone mix (5 µg/ml insulin, 1.25 ng/ml prostaglandin E$_1$ (PGE$_1$), 34 pg/ml triiodothyronine (T3), 5 µg/ml transferrin, 1.73 ng/ml sodium selenite, 18 ng/ml of hydrocortisone, and 25 ng/ml epidermal growth factor) + 10% FCS. The cell solution was subsequentially filtered through a 70-µm cell strainer, pelleted, and plated in K-1 medium + 10% FCS in a humidified 5% (v/v) CO$_2$ and 20% O$_2$ incubator at 37 °C. Medium was changed 48 h after plating. Cells were split 1:5 when sub confluent.

**Retroviral and lentiviral infections.** Retroviral and lentiviral infections and cell selection were carried out as previously described[84]. Cells were infected with the retroviruses pBabe-PURO (Vector) or pBabe-PURO-VHL30 or the lentiviruses LKO.1 expressing non-silencing control shRNA (shRNA-ns), or shRNAs against *Hif1a* (TRCN0000232220, TRCN0000232222, or TRCN0000232223), respectively termed shRNA-*Hif1a* #220, shRNA-*Hif1a* #222, and shRNA-*Hif1a* #223.

**MEF and ccRCC cell proliferation assays.** 3T3 proliferation assays of MEFs[15] have been previously described. In total, 3000 mouse ccRCC 2020 cells per well were seeded in 96 well plates in six replicates and incubated for 6 days. Cells were cultivated in K-1 medium + 10% FCS, medium was changed after 3 days of incubation. Cell proliferation was measured using a sulforhodamine B (SRB) colorimetric assay[15]. After the indicated time points the cells were fixed with 10% (w/v) trichloroacetic acid and stained with 0.057% (w/v) SRB solution. Overall, 10 mM Tris base solution (pH 10.5) was used to solubilise SRB, followed by OD measurement at 510 nm in a microplate reader (Tecan Spark 10 M plate reader). For sphere-forming assays, cell suspensions were filtered through a 40-µm cell strainer and 1000 cells were seeded in six-well low attachment plates (Corning). The cells were cultivated in K-1 medium + 10% FCS. Every 3 days fresh medium was added to the wells. After 14 days, microscopy pictures of the formed spheres were captured with DKM 23U274 camera connected to Eclipse Ts2R-FL

microscope (Nikon) at ×20 magnification. Images were acquired with IC Capture 2.4 software and analysed using ImageJ software.

**Allograft assay**. Single-cell suspensions were prepared with Accutase (Gibco) and $5 \times 10^6$ cells were resuspended in 75 µl RPMI following transfer into a precooled 30G insulin syringe mixed with 75 µl Matrigel (Corning). Syringes with cell suspension were kept on ice to avoid hardening of the Matrigel. SCID-Beige mice (Charles River Laboratories) were anesthetised by inhalation of 3% isoflurane using oxygen as carrier gas. Mice were shaved and cells were injected subcutaneously in the flank. Tumour volumes were measured weekly with a calliper. Experiments were conducted under license G-17/165 of the Regierungspräsidium Freiburg.

**Conditioned medium T-cell proliferation assay**. In total, 10,000 mouse ccRCC 2020, human RPTEC (from Dr. Jiing-Kuan Yee), 786-O (ATCC) or A498 (ATCC) ccRCC cells were seeded in triplicates in a six-well plate with 2 ml RPMI + 10% FCS and kept incubated in a humidified 5% (v/v) $CO_2$ and 20% $O_2$ incubator at 37 °C for 2 days. Two days later, spleens from C57BL/6 mice were extracted, washed in PBS and mashed through a 100 µm cell strainer in MACS buffer (PBS 1x + 2% FCS + 2 mM EDTA). The mashed spleen was filtered again through a 100 µm cell strainer into a 50 ml conical tube and centrifuged for 10 min at 290 × g. The pellet was labelled manually with magnetic CD8a (Ly-2) MicroBeads (Miltenyi Biotech). Isolated $CD8a^+$ T cells were centrifuged, resuspended in proliferation medium (RPMI + 10% FCS supplemented with 25 µM β-Mercaptoethanol) and counted. $CD8a^+$ T cells were then stained with the CellTrace Violet Proliferation Dye (Thermo Fisher). Stained $CD8a^+$ T cells were stimulated with CD3/CD28 Dynabeads (Thermo Fisher) and activated with interleukin-2 (IL-2). The conditioned medium was distributed into fresh six-well plates and $2 \times 10^5$ of stained, stimulated, and activated $CD8a^+$ T cells were added. The mix of conditioned medium and T cells was incubated for 3 days in a humidified 5% (v/v) $CO_2$ and 20% $O_2$ incubator at 37 °C. After the incubation time the T cells were resuspended and centrifuged in a 2 ml reaction tube for 5 min at 1600 rpm and 4 °C. The dead cells within the pellet were stained with the Live/Dead Fixable Aqua Dead Cell Stain Kit (Thermo Fisher), washed with 200 µl MACS buffer and centrifuged for 5 min at 515 × g and 4 °C, 25 µl CD16/32 antibody (Fisher Scientific, 14016185, diluted 1:25 in MACS buffer) was added to the pellet to block Fc-mediated reactions. After 10 min of incubation at 4 °C in the dark, 25 µl of CD8a antibody (APC-conjugated, Biolegend, 100712, diluted 1:100 in MACS buffer) was added to the suspension and incubated for 30 min at 4 °C in the dark. Afterwards T cells were washed twice with MACS buffer and the pellet was resuspended in 100 µl MACS buffer. Via flowcytometry (BD LSRFortessa) dead/living cells were measured with a 405 nm Extinction Laser (AmCyan), T cells were measured with a 640 nm Extinction Laser (APC), and the Proliferation Dye was measured with a 405 nm Extinction Laser (Pacific Blue). Data were gated and analysed using FlowJo Software.

**Immunohistochemistry**. Immunohistochemical stainings were performed as previously described[10]. Primary antibodies against the following proteins or epitopes were used at the following dilutions and antigen retrieval conditions: B220 (1:3000, BD Biosciences, 553084, Tris/EDTA 20 min, 100 °C), CA9 (1:2000, Invitrogen, PA1-16592, citrate, 10 min, 110 °C), CD3 (1:250, Zytomed, RBK024, citrate 30 min, 95 °C), CD4 (1:1000, eBioscience, 14-9766, citrate, 30 min, 100 °C), CD8a (1:200, Invitrogen, 14-0808-82, citrate buffer, 15 min, 114 °C), CD10 (1:2000, Thermo Fisher Scientific, PA5-47075, citrate buffer, 10 min, 110 °C), CD68 (1:100, abcam ab125212, citrate buffer, 30 min, 95 °C), CD69 (1:1000, Bioss, bs-2499R, Tris/EDTA, 15 min, 114 °C), F4/80 (1:250, Linaris Biologische Produkte, T-2006, BOND Enzyme Pretreatment Kit (Leica AR9551), 10 min, 37 °C), HIF-1α (1:20,000, Novus Biotechnologies, NB-100-105, citrate buffer, 10 min, 110 °C, Catalysed Signal Amplification Kit (DakoCytomation)), HIF-2α (1:1000, abcam ab109616, Tris/EDTA 15 min, 114 °C), Ly-6G (1:800, BD, 551459), MHC II (1:500, Novus Biotechnologies, NBP1-43312, BOND Enzyme Pretreatment Kit (Leica AR9551), 10 min, 37 °C), PD-1 (1:100, R&D systems, AF1021, Tris/EDTA 20 min, 100 °C), perforin (1:100, Biorbyt, orb312827, Tris/EDTA, 15 min, 114 °C), phospho-Thr37/Thr46-4E-BP1 (1:800, Cell Signalling Technologies, 2855, citrate buffer, 10 min, 110 °C). The following anti-HIF-2α antibodies did not provide specific nuclear signals in immunohistochemical staining using either Citrate or Tris/EDTA antigen retrieval methods: abcam ab199, Aviva Systems Biology ARP32253, Biorbyt orb96817, Sigma MAB3472, GeneTex GTX30114. For analyses of immune cell markers sections were scanned using a Nanozoomer Scansystem (Hamamatsu Photonics). Automatic quantifications of B220, CD3, CD4, CD8a, and CD68 positive cells were carried out from duplicate stains (average values were determined) as previously described[85] using the VIS software suite (Visiopharm, Hoersholm, Denmark). Each tumour was outlined manually. Immune cell densities were calculated as cells per $mm^2$ based on surface area and immune cell quantification. The quantifications of cells stained with F4/80 was performed using a positive pixel count and presented as percentage positive pixel. PD-1, perforin, Ly-6G, and CD69 stains were quantified by manual annotation of positively stained cells.

**Real-time PCR of mRNA and genomic DNA and recombination-specific genomic DNA PCR**. RNA was isolated from powdered frozen samples using the NucleoSpin RNA kit (Machery Nagel), cDNA prepared using random hexamer primers and Ready-To-Go You-Prime First-Strand Beads (GE Healthcare). Real-time PCR was performed using the LightCycler 480 SYBR Green Master mix (Roche) using a LightCycler 480 (Roche). The following sets of primer pairs (sequences provided as 5′-3′) were used:

*18s* (fwd GTTCCGACCATAAACGATGCC, rev TGGTGGTGCCCTTCCGTCAAT)

*Hif1a* (fwd GGTTCCAGCAGACCCAGTTA, rev AGGCTCCTTGGATGAGCTTT)

*Hif2a* (fwd GAGGAAGGAGAAATCCCGTGA, rev CTGATGGCCAGGCGCATGATG)

*Vhl* (fwd CAGCTACCGAGGTCATCTTTG, rev CTGTCCATCGACATTGAGGGA)

Genomic DNA was isolated from powdered frozen samples using the GeneElute Mammalian Genomic DNA Miniprep kit (Sigma-Aldrich). Overall, 60 ng genomic DNA per reaction was subjected to real-time PCR using the LightCycler 480 SYBR Green Master mix (Roche) using a LightCycler 480 (Roche) and annealing temperature of 55 °C. The following sets of primer pairs (sequences provided as 5′-3′) were used to amplify the following floxed and non-floxed exons:

*Vhl* Exon 1 (floxed) (fwd ATAATGCCCCGGAAGGCAG, rev TGAGCCACAAAGGCAGCAC)

*Vhl* Exon 3 (not floxed) (fwd ACCCTGAAAGAGCGGTGCCTTC, rev CGCTGTATGTCCTTCCGCACAC)

*Hif1a* Exon 2 (floxed) (fwd CGGCGAAGCAAAGAGTCTGAAG, rev CGGCATCCAGAAGTTTTCTCACAC)

*Hif2* Exon 2 (floxed) (fwd GCTGAGGAAGGAGAAATCCCG, rev CTTATGTGTCCGAAGGAAGCTG)

*Hif2a* Exon 1 (not floxed) (fwd TGGCGTCTTACAACCTCCTCCC, TCCGAGAGTCCCGCTCAATCAG)

*Trp53* Exon 4 (floxed) (fwd TGAAGCCCTCCGAGTGTCAG, rev AGCCCAGGTGGAAGCCATAG)

*Trp53* Exon 11 (not floxed) (fwd AGAAGGGCCAGTCTACTTCCCG, rev AAAAGGCAGCAGAAGGGACCG)

*Rb1* Exon 19 (floxed) (fwd AATACAGAGACACAAGCAGCC, rev GAGCCACAACTTAACCTAGTCC)

The following sets of primers were used for PCR amplification of DNA products that are specific to Cre- recombined alleles of the *Hif1a*[86] and *Hif2a*[87] genes.

*Hif1a* (Fwd II GCAGTTAAGAGCACTAGTTG, Rev GGAGCTATCTCTCTAGACC,)

*Hif2a* (P1 CAGGCAGTATGCCTGGCTAATTCCAGTT, P3 GCTAACACTGTACTGTCTGAAAGAGTAGC)

**Western blotting**. Antibodies against the following proteins were used for western blotting: β-ACTIN (1:5,000, Sigma-Aldrich, A2228), HIF-1α (1:500, Novus Biologicals, NB-100-479), LAMIN-A/C (1:500, Santa Cruz, sc-376248), LDH-A (1:500, Santa Cruz Biotechnology, sc-27230), PDK1 (1:1,000, Assay Designs, KAP-PK112-0), VHL (1:1,000, Cell Signalling Technologies, #68547), VINCULIN (1:5,000, Abcam, ab130007).

**RNA-sequencing**. RNA was isolated from powdered frozen samples of WT kidney cortex controls from Cre negative mice in the $Vhl^{fl/fl}Trp53^{fl/fl}Rb1^{fl/fl}$ background and from tumours of the different genetic backgrounds using the NucleoSpin RNA kit (Machery Nagel). Paired-end RNA-sequencing was performed on an Illumina HISEQ4000 device by the core facility of the German Cancer Research Center (DKFZ) in Heidelberg with the Illumina TruSeq Stranded RNA library preparation kit. Previously published sequencing data of WT cortex and tumours from the $Vhl^{\Delta/\Delta}Trp53^{\Delta/\Delta}Rb1^{\Delta/\Delta}$ mouse model was also included for subsequent analyses[16]. Raw data fastq-files were pre-processed with trimmomatic[88] to assure sufficient read quality by removing adaptors and bases in the low-quality segment regions (end of the reads) with a base quality below 20. Before trimming the average number of reads was 48309915 ± 12246964 [26804116,70620322], after trimming the average number of reads was 45451780 ± 13975818 [20629675,70032834]. Hence, an average of 93.2% ± 10.5% of the raw reads survived the trimming step (Supplementary Fig. 6a). The overall quality of the bases and reads was good. After quality control and trimming the reads were 2-pass aligned using the STAR aligner[89] and the GRCm38 reference genome from Ensembl. In total, 85.1% ± 3.3% of the reads were uniquely mapped and considered (Supplementary Fig. 6b). The alignment step was followed by normalisation and differential expression analysis with the R/Bioconductor[90] package DESeq2[91]. The normalisation of the raw read counts was performed with DESeq2 by considering the library size. In addition, all genes with a low count across all samples, i.e., the row sum of a gene was below 5 in a gene by sample matrix, were removed from the dataset. After pre-processing and filtering 19,723 genes were further analysed and fitted with a negative binomial generalised linear model followed by Wald statistics to identify differentially expressed genes. Genes were considered significant with an adjusted p value < 0.001 (Benjamini–Hochberg). Raw RNA-sequencing data have been uploaded to GEO

with identifier GSE150983 [https://www.ncbi.nlm.nih.gov/geo/query/acc.cgi?acc=GSE150983].

**Gene-set enrichment analysis.** Enrichment of signalling pathways was performed as implemented in the R/Bioconductor package GAGE[92] with signalling pathways from Gene Ontology[93,94], ConsensusPathDB[95] and MSigDB[96]. The human gene identifiers from the MSigDB pathways were mapped on mouse homologues with the R/Bioconductor package GeneAnswers (R package version 2.28.0). Pathways were considered significant with an adjusted $p$ value < 0.05 (Benjamini–Hochberg).

**ssGSEA immune deconvolution analysis.** RNA-seq raw read sequences were aligned against mouse genome assembly mm10 by STAR 2-pass alignment[89]. QC metrics, for example general sequencing statistics, gene feature, and body coverage, were then calculated based on the alignment result through RSeQC. RNA-seq gene level count values were computed by using the R package GenomicAlignments[97] over aligned reads with UCSC KnownGene[98] in mm10 as the base gene model. The Union counting mode was used and only mapped paired reads after alignment quality filtering were considered. Gene level FPKM (fragments per kilobase million) and raw read count values were computed by the R package DESeq2[91]. Single-sample GSEA[99] was utilised for immune deconvolution analyses based on FPKM expression values to estimate the abundance of immune cell types[44], MHC class I antigen presenting machinery expression, T-cell infiltration score, immune infiltration score[43], and immune cytolytic score[100] as well as the eTME signatures[42] which was developed from leveraging RCC patient-derived xenograft RNA-sequencing data. In addition to the gene signature-based deconvolution approach, CIBERSORT[48] which is a regression-based method using Support Vector Machine algorithm was also employed using either the human gene panel or the mouse specific reference panel, ImmuCC[45].

**Mass spectrometry-based proteome profiling.** Tissue sections of WT kidney cortex controls from Cre negative mice in the $Vhl^{fl/fl}Trp53^{fl/fl}Rb1^{fl/fl}$ background and from tumours of the different genetic backgrounds were homogenised by sonication (Diagenode, 4102 Seraing, Belgium) in 100 mM HEPES, pH 8.0, 4% (w/v) SDS, 10 mM DTT, followed by heat incubation (95 °C, 10 min), centrifugation ($16,000 \times g$, 10 min), cysteine alkylation, and buffer exchange to 100 mM HEPES, pH 8.0 with subsequent trypsinization of 100 µg proteome based on the filter aided sample preparation protocol[101]. Overall, 24 samples (six mice from four genotypes) were distributed across three sets and differentially labelled by amine-reactive tandem-mass tags (TMT11plex, Thermo/Pierce, Rockford, lL, USA) including a pooled normalisation sample. A summary of the labelling scheme can be found online as part of the ProteomeXchange submission (see below for details). Each batch was fractionated by high pH reversed phase chromatography (XBridge C18, 3.5 µm, 150 mm × 4.5 mm column (Waters, MA, USA)). Both eluents A (water) and B (70% acetonitrile) contained 10 mM ammonium formate, adjusted to pH 10 with ammonium hydroxide. Flow rate was 0.3 ml/min. After washing with 16% B, samples were eluted by a linear gradient from 16–55% B in 40 min. Peptide elution was monitored by UV/VIS absorption at 214 nm. Overall, 16 fractions were collected and concatenated into eight final fractions (pool scheme was 1 + 9, 2 + 10, 3 + 11, 4 + 12, 5 + 13, 6 + 14, 7 + 15, and 8 + 16). For analysis by liquid chromatography–tandem-mass spectrometry (LC–MS/MS), fractions were separated by an EASY nano-LC system 1000 (Thermo Fisher Scientific, Waltham, MA, USA) and using an EASY-Spray™ C18 column (250 mm × 75 µm, 2 µm particles heated at 50 °C, Thermo Fisher Scientific, Waltham, MA, USA). Both eluents A (water) and B (acetonitrile) contained 0.1% formic acid. The gradient programme consisted of the following steps: linear 2–25% B increase in 60 min and 25–60% B in 30 min, providing a 90 min separation window at 300 nl/min flow rate. Peptides were analysed using an Oribtrap Q-Exactive Plus mass spectrometer (Thermo Fisher Scientific, Waltham, MA, USA) operating in data dependent acquisition mode. Survey scans were performed at 70,000 resolution, an AGC target of 3e6 and a maximum injection time of 50 ms followed by targeting the top ten precursor ions for fragmentation scans at 35,000 resolution with 1.2 m/z isolation windows, an NCE of 32 and a dynamic exclusion time of 40 s. For all MS2 scans, the intensity threshold was set to 1000, the AGC to 1e5, maximum injection time of 100 ms and the fixed first mass to 100 m/z. Data were analysed by MaxQuant v 1.6.013 with the following settings: tryptic specificity, up to two missed cleavages, TMT-modification of peptide N-termini and lysine side chains; cysteine carbamido-methylation, mouse reviewed sequences (downloaded from Uniprot on Aug 26th, 2019), 1% FDR for peptides and proteins, precursor intensity fraction = 0.5, one or more unique peptides for protein quantitation. MaxQuant output was further processed by MSStatsTMT[102] for normalisation, batch removal, and protein assembly. Differential protein abundance was assessed using linear models of microarray analysis. Data are available via PRIDE/ProteomeXchange with identifier PXD016630[103]. Of note, the dataset is associated with the ontology term "TMT6plex" since an ontology term for the isobaric TMT11plex has not yet been established.

**Reporting summary.** Further information on research design is available in the Nature Research Reporting Summary linked to this article.

## Data availability
Raw proteomics data are available via PRIDE/ProteomeXchange with identifier PXD016630. Raw RNA-sequencing data have been uploaded to GEO with identifier GSE150983. The source data underlying analyses in Figs. 3b–i, 4c–g, 5a–c, 6a, 7a and Supplementary Figs. S3, S7a–e, S8a, b, S9a–d, S10a, and S12a are provided as Supplementary Data 1–6. Full scan blots for Supplementary Figs. 2 and 5 are provided as Source Data file. All remaining relevant data are available in the article, Supplementary Information, or from the corresponding author upon reasonable request. Source data are provided with this paper.

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

## Acknowledgements

We are most grateful to the team of Stephan Wolf from the Genomics and Proteomics Core Facility, German Cancer Research Center/DKFZ, Heidelberg, Germany for their sequencing service, to Martin Biniossek for support with the proteomic analysis and to the members of the Frew laboratory for helpful discussions. This work was supported by grants from the Deutsche Forschungsgemeinschaft to I.J.F. (BIOSS Excellence Cluster and CRC 850), to R.Z. and M.B. (CRC 850), to O.S. (SCHI 871/11-1), from the Else-Kröner-Fresenius Stiftung and Berta-Ottenstein Programme for Clinician Scientists to R.H., and to M.B. and P.M. from the German Federal Ministry of Education and Research (BMBF) for MIRACUM within the Medical Informatics Funding Scheme (FKZ 01ZZ1801B). Open access funding provided by Projekt DEAL.

## Author contributions

R.H., S.H., S.S., D.S., M.A., A.P., F.U., P.S., D.H., C.C., and I.F., designed and conducted experiments, P.M., C.C., M.C.-C., R.H., O.S., F.K., and I.F. analysed RNA seq and proteomic data, C.S. contributed pathological analyses, A.J., B.L., N.G., T.C., J.D., R.Z., M.H., O.S., A.H., M.B., and I.F. provided technical assistance and scientific advice, I.F. designed the study and wrote the paper in consultation with all authors.

## Competing interests

The authors declare no potential conflicts of interest.
