## [Peer Review File · Nature Communications]

Reviewers' comments:

Reviewer #1 (Remarks to the Author): expertise in ccRCC (and HIF)

This manuscript, which represents a tour de force as it involves the generation of mice with 4 different conditional genes that are extensively characterized not only phenotypically, but also at the transcriptomic and proteomic levels, compellingly challenges the current paradigm that HIF1a functions as a TSG in ccRCC. While the results are of necessity contextual to the other mutations, VPR, the data is unequivocal. This work represents a major advance in the field and the authors deserve to be commended for the effort and thoroughness.

Major comments

The authors need to emphasize that their findings while rather provocative are contextual to the VPR genetic background and of course, mice.

It is unclear from SFig2, whether the recombination rates for the different alleles are the same. Error bars are quite large. May be good to expand on the number of tumors evaluated and provide n in the Fig legend.

RNAseq for H2 targeting is least convincing for effective recombination. IHC is also not available. Thus, comments about the modest role of H2 loss on tumor growth need to be tempered by these observations, or more compelling data about its effective recombination should be provided.

The authors report profound differences between the transcriptome they assign to H2 in their model and the previously defined H2 transcriptome in ccRCC using PDX models. These data are important and need to be expanded upon. If the H2 transcriptome they define is so different from what is observed in human ccRCC treated with a highly specific inhibitor, there may be concerns about how the particular genetic mutations in their model are shaping the transcriptome. Broader analyses looking at different thresholds should be performed.

In their transcriptomic analyses, the authors argue that targeting H2 results in broad activation of the immune system. Two comments. First, it seems that a bigger difference is suppression of at least some of these markers by loss of H1 (at least based on some of the heatmaps provided). Second, to assess the significance of these findings for humans, the authors are encouraged to contrast them with recent reports of the H2-dependent transcriptome in patient metastases (Courtney et al., CCR 2020).

It would be fitting to expand on the analyses of the impact of H2 loss on the tumor microenvironment, by comparison to the empirically defined ccRCC TME (Wang et al Cancer Discovery 2018).

Comparative analyses with human ccRCC with loss of 14q need to be interpreted cautiously as 14q involves many more genes than H1a. It is also unclear whether these analyses are restricted to tumors with homozygous deletions.

Minor comments

Authors should comment on the potential impact of intron read through of Vhl following LoxP recombination.

The authors state that CA9 is a H1 target gene, but the evidence provided is not compelling, and it is unclear that this is the case in humans. They are uniquely poised to address this and it would be good to examine this systemically across all the tumors they have generated.

Would recommend including S5A in the main report and changing the colors of the dots to

facilitate analysis. Authors should also comment on the fact that PC2 accounts only for 9% of the variability.

Reviewer #2 (Remarks to the Author): expertise in proteomic

The selection of proteomic protocols for sample preparation, LC-MS/MS analysis, data interrogation is very solid, but requires corrections / clarifications. The authors used state of the art sample preparation technique, popular TMT-labeling technique for quantitation, advanced 2D-LC peptide separation procedure, robust MS instrument. I found very convincing the comparison between quantitative proteomic analyses of human and mice ccRCC. At the same time the description is not sufficiently clear.

Minor corrections:

(page 22).

1) Please specify exact type of TMT labeling chemistry used: "TMT-10plex" or "TMT-6plex"

2) please specify the sample distribution through 10 channels: e.g 2(WT) – 2 – 2 – 2 (tumour) – 2 (pooled control samples) ?

3) Description of the project at Proteome Exchange PXD016630 lists TMT6-plex as labeling chemistry. This is unclear how 24 samples can fit three 6-plex datasets.

4) Thermo – Thermo Fisher Scientific.

5) "Waters" This is unclear why in one case in this section the state/country of origin is provided and in the others is not.

6) "XBridge C18 column, 150 mm x 4.5 mm column containing 3.5µm particles (Waters)" – "XBridge C18, 3.5µm, 150 mm x 4.5 mm column (Waters)".

7) Please provide gradient slope for high pH RP separation: e.g. "2% acetonitrile per minute increase" and number of fractions used for the analysis. It is very hard to judge on the quality of proteomic identification /quantitation without knowing overall analysis time / number of fractions used.

8) "...linear gradient of increasing buffer B (0.1% formic acid in acetonitrile, Fluka)..." – please provide composition of buffer A and clarify the description: e.g.

"Both eluents A (water) and B (acetonitrile) contained 0.1% formic acid. Gradient program consisted of following steps: linear 2-25% B increase in 60 minutes and 25-60% B in 30 minutes, providing 90 min separation window at 300 nL/min flow rate."

9) Which nano-LC system has been used?

10) The description of DDA parameters isn't complete. It usually includes resolution for both MS and MS/MS, maximum IT (injection times), target AGC level, collision energy for MS/MS. Please refer to some previous publications using QE Plus instrument.

11) "Q-Exactive Plus" – "Orbitrap Q-Exactive Plus"

12) Proteome Exchange submission PXD016630 does not provide sufficient details for the researchers who want to use authors data: it has 24 raw files and one output MaxQuant file – all in duplicates. Fraction #s for each of three experiments and associated with each run distribution of samples throughout the channels should be available for Proteome Exchange users.

Reviewer #3 (Remarks to the Author): expertise in ccRCC - VHL/HIF

This is an important paper contributing to the long-standing discussion on the importance of HIF 1 vs HIF2 in ccRCC initiation and progression. Although the field is leaning more towards HIF2 being an oncogene and HIF1 a tumor suppressor, there are data supporting HIF1 as an oncogene, and the discussion continues to be a hot topic. This paper utilizes a mouse model of ccRCC, which relies on the loss of VHL tumor suppressor along with p53 and pRb. The data acquired in that model are complemented with the data from shRNA knockdowns of HIF1. Overall the paper is interesting and well written. It experimentally addresses the discrepancies in experimental data

acquired with ccRCC cell lines +/- VHL and +/- HIFs in vitro and mouse models of ccRCC in vivo, including human cell line-based models and their own mouse model in the current study, in terms of the influence of HIF1 and HIF2 on tumor initiation and progression. The paper further investigates the gene expression signatures and proteomic alterations when HIF1a is deleted vs HIF2a is deleted from the VHL-/-p53-/-pRb-/- tumors. To be accepted for publication, the paper needs to be revised to better interpret the acquired results; more experimental support needs to be provided for the involvement of HIF-mediated immune infiltrates in ccRCC progression, as well as some additional comments below addressed. The reviewer highly encourages to submit the revised manuscript.

Major:

1. Although the study did a good job showing that HIF1a is important for tumor initiation, no data were provided to address the issue as to why HIF1a is lost during tumor progression in 30% of patients. It would be helpful to see the data when HIF1a deletion is introduced at later stages of tumor progression in this mouse model and the tumor growth assessed, ideally with analysis of immune microenvironment (see comments on claims below).
2. The immunologic part of the study is very interesting, but seems incomplete in terms of data acquisition and data interpretation.
 - a) It is well accepted that inflammation is a hallmark of cancer promoting tumorigenesis; at the same time infiltration of certain immune cells is harmful to the tumor and is exploited for immunotherapies. The study falls short at separating these two branches of immune microenvironments. It should be revised to make the message clear. For all of the immune cell types found in this study which are affected by HIF1 and/or HIF2 expression, the data interpretation is largely absent: which cell types promote tumor growth, which suppress, in terms of initiation vs progression, how are HIFs regulating the microenvironment?
 - b) The transcriptomic study operates with tumor material containing a mixture of tumor cells and the cells in the microenvironment, pooling these together. It is assumed that the immune signatures derive from the infiltrated immune cells, but HIFs and VHL/p53/Rb are manipulated in tumor cells. There is no mechanism provided for the seen phenotype, please speculate on potential mechanism or provide additional data.
 - c) Please discuss the human ccRCC tumor infiltration by tumor-associated immune cells for patients with HIF1-only, HIF2-only, and HIF1/HIF2 tumors if the data is available. Speculate on checkpoint inhibitors as therapeutics for HIF1-only, HIF2-only, and HIF1/HIF2 tumors based on your findings.
 - d) In the context of the findings, it is important to conduct tumor growth experiments with inhibitors or activators of specific branches of immune system in mice with deletion of HIF1 or HIF2 of HIF1WT/HIF2WT to clear the message of the paper about the role of HIF1 and HIF2 in immune response and tumorigenesis. Currently the message is confusing and lacks experimental support.
 - e) The data in Figure 3H&I are not consistent with Figure 4G. In figure 3 it seemed that HIF1 was promoting IFN production and HIF2 inhibiting it, in Figure4 VHL deletion was not providing a change, HIF1 and HIF2 were doing the same (inhibiting IFN response): please comment on these conflicting data to reconcile the discrepancies.
 - f) In figure 6 there does not seem to be an advantage of HIF2 expression in terms of immune cell infiltration (E)? Please interpret the findings.
3. Please comment in text of the paper on the mouse model of ccRCC used. It is very well known that VHL deletion is not sufficient for ccRCC development in mice. Thus additional genetic events are necessary for the mouse model to develop disease. It was stated in the paper that p53 loss alleviates senescence. The caveat is that p53 is generally not mutated in ccRCC.
4. Please discuss in paper the higher grades of the tumors when HIF1a or HIF2a were deleted compared to control (VHL-/-p53-/-pRb-/-).
5. It seems important to validate the model with HIF2a antibody by IHC or by western blot. The abcam anti-HIF2a antibody should work for this purpose (worked in published studies on mouse cell lysates). If it does not, then several HIF2a specific proteins should be used instead (from the list determined in this study).

Minor:

1. Please show the HIF1a expression in HIF2a deleted tumors and HIF2a expression in HIF1a deleted tumors to rule out the possibility of cross-regulation.
2. Page 8 1st sentence, please provide a citation for the rescue experiment.
3. "3T3 proliferation assays" are confusing since the assays are done here on MEFs. Please either justify the name of the assay or remove 3T3.
4. HIF2 transcriptional network on page 9 states PGK1, later in the paper it is referenced as HIF1 and HIF2 gene (Figure S4).
5. Please comment on chronic (knockout) vs acute (HIF2 inhibitor PT2399) way to inhibit HIF2 and the absence of overlap of HIF2 target genes, top of page 10.
6. Page 17, HIF1a inhibition seems to be not feasible as a therapeutic approach due to ubiquitous expression. Please describe another feasible approach based on your data to target consequences of HIF1a overexpression.
7. Sections in Figure 1D seem to be non-serial, if possible, provide serial sections.
8. Figure 2: H, I, why did the mice die? If they were sac'ed due to tumor burden, please state in the figure and text.

Reviewer #4 (Remarks to the Author): expertise in transcriptomic (cancer and TME)

Hofflin et al. performed genomic studies and functional experiments to assess the functional consequences of Hif1a or Hif2a deletion in ccRCC models. While I find this work is interesting, there are however a number of issues (e.g. data quality, data processing, superficial analysis, etc.) that limit the overall impact of the study.

Specific points:

Some of the main conclusions of this manuscript were drawn based on their RNA-seq data expression and pathway analysis. However, there is no detailed description in the Methods (RNA-sequencing paragraph) about the generation of their RNA-seq data, such as the kit and protocol used for sequencing library preparation, the platform used for data generation, and the detailed quality check metrics at sample level. The authors mentioned "reads trimming" in their methods for RNA-seq data processing, was it because of the poor quality reads in their data? The author should provide details on this, such as the base and read-level quality metrics of their RNA-seq data, how the reads trimming was performed? the proportion of reads trimmed per sample, the number of bases trimmed off? the author should evaluate whether the difference in sample quality, reads trimming, or other filtering process will impact their downstream expression and pathway analysis. Also important, the author should describe the details how the RNA-seq data is normalized, were there any batch effects by statistical assessment, if present, how the batch effects were corrected and how the data was normalized and filtered. Without these details, it is hard to make any adjustment about the quality of their RNA-seq data and the reliability of their conclusions.

For pathway GSEA analysis, the authors analyzed signaling pathways from Gene Ontology, ConsensusPathDB and MSigDB, it is known that some pathways from these different databases are functionally redundant (or most of the genes are overlapped), however, it is unclear from their methods, how these redundant pathways were processed and counted. In addition, the authors only used adjusted p-values to select the significant pathways, however, the normalized enrichment scores should also be considered. In their figures 3-4, multiple heatmaps are shown on different pathways. First, it is unclear how many pathways in total were studied? How these

pathways were selected (based on which criteria)? Are there additional pathways showed statistical significant? If so, what are those pathways? Are those pathways functionally linked to the pathways displayed here? Second, it doesn't seem like the authors displayed a full list of the genes included in each pathway, questions are: how these genes (listed in the figures) were selected? How about the rest of genes in each selected pathway?

The authors used Single-Sample GSEA for immune deconvolution analyses, it is unclear how the immune related signatures are selected and whether this approach is validated and statistically sound. It is recommended that the author should also try a couple of other validated immune deconvolution approaches such as CIBERSORT, MCP-Counter, TIMER, etc. to ensure that their discoveries are reproducible.

For LC-MS based proteomics data analysis, as it is known that not all the proteins are covered well, similarly, not the full length of every protein is covered. And the expression of many gene at mRNA and protein level are not consistent due to other possible regulatory mechanisms. The authors tried to use the LC-MS data to support their discovery from RNA-seq data. However, first, it is unclear that how many proteins were detected (and highly expressed) by LC-MS. Especially, the proteins involved in the pathways of interest. If a protein showed no expression from LC-MS, was it because the protein was low abundant, or because the digestion approach they used was not appropriate (e.g. the enzyme used for LC-MS results in numerous of either too long or too short peptides that were not easily captured) The correlation between protein and mRNA expression (for both low-abundant and highly abundant genes) should be evaluated. Figures 4F-G, author picked genes from multiple pathways and put them together in the heatmap, again, the problem is that how these genes were selected? An overview of GSEA results on all pathways should be displayed.

Figure 1B, although the authors stated statistical difference between VpR and VpRH1 and VpRH2, the values of VpRH1 and VpRH2 were widely distributed along the entire y axis, indicating a high degree of heterogeneity. What are the possible molecular determinants of such a heterogeneity? What are other factors likely influence the molecular consequences of HIF-1a or HIF-2a deletion?

Figure 3D, there are many different DNA repair pathways, which pathway(s) in particular showed significant differences?

Figure 3H, the author showed T-cell activation genes, however, some genes in this list, for example, Foxp3, Visr, are immune inhibitory. What are these genes here? Are they related to CD4 T cell or CD8 T cell activation? Similarly, in the panel I, the IFN- γ production gene list contains Cd14, Cd276, Foxp3, which raised a concern whether the author used the right gene signature for their analysis.

Figure 4F, the author showed "HIF-1a-dependent" genes in the heatmap, however, some of these genes showed similar expression levels between both WT-Cortex and VpRH1 (the bottom part of 4F), such as Apool, Mzb1, C1qbp, etc. Based on what evidence, that the authors determined that these genes are HIF-1a "dependent"?

Figure 5A, how was the immune infiltration score was calculated? Is this relative or absolute abundance? Again, it is strongly suggested that multiple immune deconvolution approaches should be used to ensure that the finding is reproducible. The authors used IHC staining to show the density of CD3, CD4 and CD8 positive T cells. but the number of markers used is very limited. It would be great to add several more markers showing T cell cytolytic activity and myeloid cell states. It is of note that the values of CD4+ cells and CD8+ cells in Figure 5C in the VpRH1 and VpRH2 varied greatly along the y axis. Some values are very low while others are much higher. How many areas are counted? Are the technical duplicates applied in this assay? The author should explain this variation.

Figures 6A-B, when the author identify HIF1A loss, how the "loss" was defined here? the mRNA expression should be integrated with copy number data. the author showed OS, how about DSS, DFS? Are they also significant? Is HIF1A loss co-existing with other factors in patients showed worse survival? the multivariate Cox model should be applied here to ensure that impact of HIF1A loss is still significant after adjusting for other confounding factors.

Supplementary Figure S2, the author showed coverage track of the RNA-seq data, however, the scale on y axis was not displayed and it is not clear whether the same scale was used for different tracks. It is of note that for the the Floxed exon 2 of *hif2a*, there is still signals and coverage on exon 2, are these Residual floxed exons? What are the % stromal cells? If the Residual floxed exons was indeed caused by DNA derived from non-Cre-expressing cells of the tumour stroma, appropriate normalization approach should be applied to correct the % stromal cells for a fair comparison.

Reviewer #1

This manuscript, which represents a tour de force as it involves the generation of mice with 4 different conditional genes that are extensively characterized not only phenotypically, but also at the transcriptomic and proteomic levels, compellingly challenges the current paradigm that HIF1 α functions as a TSG in ccRCC. While the results are of necessity contextual to the other mutations, VPR, the data is unequivocal. This work represents a major advance in the field and the authors deserve to be commended for the effort and thoroughness.

We greatly appreciate this very positive endorsement of our study as well as the careful consideration of our study by the reviewer. This included several important suggestions that we have been able to address in the revised manuscript. We believe that this has led to a great improvement in our study.

Summary of new data/figures:

Figure 1d: Addition of new HIF-2 α immunohistochemical stainings

Figure 3a: Moved PCA analyses from previous Suppl. Fig.

Figure 5: Entirely new figure with new analyses of MHC class I and II in mouse and human ccRCC

Figure 6a: New figure. Addition of two new methods of bioinformatic immune cell deconvolution.

Figure 6e-k: New quantifications of stainings of 7 additional immune cell markers

Figure 7c: Addition of two new methods of bioinformatic immune cell deconvolution and new analyses based on tumour segregation based on *HIF2A* mRNA expression

Suppl. Fig. 3: New figure. Analyses of *Hif1a*, *Hif2a* and CA9 mRNA expression.

Suppl. Fig. 6a-c: New figure. Description of RNA-seq read trimming, mapping and absence of batch effects.

Suppl. Fig. 7: New panels d and e. Comparison of human and mouse HIF-2 α -dependent genes.

Suppl. Fig. 8: Moved GSEA enrichment graph from previous Figure to panel a, added new mRNA analyses of T cell activation genes in panel b.

Suppl. Fig. 9a-d: New figure. Correlations between proteome and transcriptome.

Suppl. Fig. 11: New figure. Shows examples of tumours with high and low abundance of cells staining for the different immune cell markers. Shows a better description of the methods of automatic quantification.

Suppl. Fig. 12a-h: New figure. Series of analyses related to Figure 7.

Suppl. Data 3: Added 8 new tabs to the previous Excel sheet (previously Suppl. Table 3 which showed only the proteomic expression values) describing the results of two new methods of gene set enrichment analyses of our proteomic data.

Suppl. Data 4: Analyses of MHC class I and II mRNA correlations to *HIF2A* mRNA abundance in human ccRCC.

Suppl. Data 5: New table describing the gene sets used for immune cell bioinformatic deconvolution.

Suppl. Data 6: Clinical co-variants of *HIF1A* copy number loss in human ccRCC

Major comments

The authors need to emphasize that their findings while rather provocative are contextual to the VPR genetic background and of course, mice.

We completely agree with this comment and while we tried to discuss these limitations in the first version of our manuscript, this was not clearly stated enough. We have addressed this issue through new text in the Discussion section.

(p17) "While it cannot be excluded that there might be differences between mice and humans, and the findings may be contextual to the *Vhl/Trp53/Rb1* mutant background, this result seemingly....."

(p18) "Since our current and previous mouse genetic studies collectively show that the requirement for HIF-1 α and HIF-2 α is dependent on the underlying genotype of the tumour (*Vhl/Trp53* mutations versus *Vhl/Trp53/Rb1* mutations), it will be important to test the generality of our findings in other genetic mouse models of ccRCC, particularly as epigenetic modifications resulting from mutations in *Pbrm1* have been shown to alter HIF- α transcriptional outputs."

It is unclear from SFig2, whether the recombination rates for the different alleles are the same. Error bars are quite large. May be good to expand on the number of tumors evaluated and provide n in the Fig legend. We apologise for the oversight in not providing the number of samples analysed in this graph. This has now been added to the figure legend. We believe that the variability in the measurements most likely comes from different relative ratios of tumour cells to non-tumour cells within the sample, reflecting different stromal and immune cell compositions of each tumour biopsy. Given this fact, we concluded that the suggestion below related to IHC provided a better method of proving *Hif2* targeting than repeating these analyses (which by nature are subject to high variability) using more samples.

RNAseq for H2 targeting is least convincing for effective recombination. IHC is also not available. Thus, comments about the modest role of H2 loss on tumor growth need to be tempered by these observations, or more compelling data about its effective recombination should be provided.

We would like to emphasise that the degrees of recombination of *Trp53*, *Rb1* and *Hif2a* that are detected both by DNA PCR (Suppl. Fig. S2B) and by analysis of RNA seq traces (Suppl. Fig. S2B) in VpRH2 tumours are fairly equivalent to one another and it is not really the case that *Hif2a* is the odd one out in the analyses. However, we certainly take the point that additional confirmation of loss of HIF-2 α in the VpRH2 tumours would be desirable. We now tested six different commercially-available anti-HIF-2 α antibodies using two different antigen retrieval methods and found one condition that gave specific nuclear signals for HIF-2 α in IHC stainings. (We have included the details of the antibody that we found to work, as well as those that do not work in our hands in the Methods section, to hopefully save other groups in the future the trouble that we have had in detecting HIF-2 α). Examples of these stainings are shown in Fig. 1d. Nuclear HIF-2 α staining was present in tumours from *Vhl* Δ/Δ *Trp53* Δ/Δ *Rb1* Δ/Δ (35/35 tumours) and *Vhl* Δ/Δ *Trp53* Δ/Δ *Rb1* Δ/Δ *Hif1a* Δ/Δ (11/11 tumours) mice but absent in all tumours from *Vhl* Δ/Δ *Trp53* Δ/Δ *Rb1* Δ/Δ *Hif2a* Δ/Δ (0/54 tumours) mice. This result demonstrates that HIF-2 α is deleted in the tumours that emerge in *Vhl* Δ/Δ *Trp53* Δ/Δ *Rb1* Δ/Δ *Hif2a* Δ/Δ mice and validates our conclusion that HIF-2 α has relatively minor effects on tumour growth in this model.

The authors report profound differences between the transcriptome they assign to H2 in their model and the previously defined H2 transcriptome in ccRCC using PDX models. These data are important and need to be expanded upon. If the H2 transcriptome they define is so different from what is observed in human ccRCC treated with a highly specific inhibitor, there may be concerns about how the particular genetic mutations in their model are shaping the transcriptome. Broader analyses looking at different thresholds should be performed.

Second, to assess the significance of these findings for humans, the authors are encouraged to contrast them with recent reports of the H2-dependent transcriptome in patient metastases (Courtney et al., CCR 2020).

Thank you for this comment, we agree that this was an important issue to clarify and the suggestion was very helpful in this respect. We have now conducted new analyses based on the findings and method reported in Courtney et al CCR 2020 that defined HIF-2 specific target genes based on RNA seq analyses of the effects of the HIF-2 inhibitors PT2385 or PT2399 on ccRCC tumorgrafts or patient metastases. We used the set of 277 genes defined by Courtney et al CCR 2020 as being dependent on HIF-2 α specifically in human ccRCC tumour cells (note that this does not formally exclude that some of these genes might also be dependent on HIF-1 α) and analysed the expression levels of the mouse orthologues of these genes in our tumour RNA seq. This is described in the Results section on p10 as follows:

“To further investigate potential overlap with recently-defined HIF-2 α -dependent genes in human ccRCC, we used a set of 277 genes that were identified as being inhibited specifically in tumour cells in ccRCC tumorgrafts in mice treated with the HIF-2 α inhibitor PT2399^{1,2}. Analyses of the expression levels of the mouse orthologues of this set of HIF-2 α target genes revealed that many of these genes are highly upregulated in *Vhl* Δ/Δ *Trp53* Δ/Δ *Rb1* Δ/Δ tumours compared to WT cortex, but that the loss of either HIF-1 α or HIF-2 α did not broadly affect the up-regulation of these genes (Supplementary Fig. 7d). Nonetheless, 11 genes, marked in red in Supplementary Fig. 7d and shown in Supplementary Fig. 7e, were expressed at significantly lower levels (fold change <-1.7, P <0.05) in *Vhl* Δ/Δ *Trp53* Δ/Δ *Rb1* Δ/Δ *Hif2a* Δ/Δ tumours than in *Vhl* Δ/Δ *Trp53* Δ/Δ *Rb1* Δ/Δ tumours. The relatively small overlap between mouse and human HIF-2 α -dependent genes may be due to inherent differences between mice and humans, to the very different experimental settings of acute pharmacological inhibition versus tumour evolution in the genetic absence of *Hif2a*, or to specific features of this particular model of ccRCC. In this latter context, it is noteworthy that many human HIF-2 α -dependent ccRCC genes are related to the cell cycle and to DNA damage responses. These signatures are highly represented in the comparison *Vhl* Δ/Δ *Trp53* Δ/Δ *Rb1* Δ/Δ vs WT cortex (see GAGE signatures in Supplementary Data 2). We speculate that it is likely that these genes are not dependent on HIF-2 α in the mouse model due to the fact that the deletion of *Rb1* and *Trp53* already strongly affects these classes of genes.”

In their transcriptomic analyses, the authors argue that targeting H2 results in broad activation of the immune system. Two comments. First, it seems that a bigger difference is suppression of at least some of these markers by loss of H1 (at least based on some of the heatmaps provided).

It would be fitting to expand on the analyses of the impact of H2 loss on the tumor microenvironment, by comparison to the empirically defined ccRCC TME (Wang et al Cancer Discovery 2018).

Thank you for this suggestion which we have also acted upon. Since other reviewers also commented about the importance of using independent methods (different sets of gene signatures) of bioinformatic deconvolution of the immune microenvironment we have conducted an entirely new set of analyses using our original sets of signatures (Bindea&Others), using the signatures of the ImmuneCC approach (which is essentially CiberSort using mouse-specific matrices of genes) and using the eTME signatures that were defined by Wang et al Cancer Discovery 2018. A comparison of the genes used in the Bindea&Others and eTME signatures has now been provided in Supplementary Data 5. Both of these signatures were previously used to deconvolve the immune microenvironment of human ccRCC using a ssGSEA approach and we therefore utilised the mouse orthologues of these genes for our analyses. While the Bindea&Others and the eTME methods analyse the relative enrichment (z-scores) of gene sets that are specifically expressed in different subsets of cells, the ImmuneCC method uses a matrix-weighted score based on both high and low expressed genes in each immune subset to assess relative abundance of the population via a z-score. We have now summarised all the z-scores from these three analyses and their statistical significances in a new Figure 6A in which the terms have been grouped based on the immune subtype that they correspond to, to facilitate comparisons between the three methods. Our new sets of conclusions about the effects of HIF-1a and HIF-2a loss are based on the consensus of these signatures and are presented in the Results on p13 and p14. We have also added a new series of IHC stainings for different cell types/activation states (MHC-II, PD-1, CD69, Perforin, B220, CD68, F4/80, Ly-6G) to gain a better overview of the immune microenvironment in our tumour models (Fig. 5, Fig. 6). We also compare our results to other findings in ccRCC. This is reflected in extensive alterations and additions to the Results and Discussion sections.

Comparative analyses with human ccRCC with loss of 14q need to be interpreted cautiously as 14q involves many more genes than H1a. It is also unclear whether these analyses are restricted to tumors with homozygous deletions.

Thank you for this comment, we have now addressed this issue. “*HIF1A* loss” includes tumours with either homozygous deletions (only a very few tumours show this) or heterozygous deletions (the vast majority) as reflected in the Oncoprints based on GISTIC analyses in Figure 7A. We agree that there are potential confounding factors related to the co-deletion of other genes and we have conducted further analyses that are described by the following text in the Results section, p15.

“Previous studies have identified that loss of larger regions of chromosome 14q, including the *HIF1A* gene, correlates with poor prognosis¹⁹. We took advantage of the extensive clinical and whole exome sequencing data of the TCGA dataset to investigate whether co-variants of *HIF1A* loss may account for the observed survival differences. Tumours with *HIF1A* loss were statistically more likely to have higher grade and stage, and display lymph node positivity and metastases (Supplementary Data 6), consistent with this subgroup representing a more aggressive form of ccRCC. The only mutation that occurred more frequently in the *HIF1A* loss subgroup than the unaltered subgroup was *BAP1*, which was detected in 9% of all ccRCC tumours (Supplementary Fig. S13e,f). However, *BAP1* mutation status alone did not significantly affect survival (Supplementary Fig. S13g) in this cohort and removal of all *BAP1* mutant tumours from the cohort did not alter the correlation of *HIF1A* loss with poor prognosis (Supplementary Fig. S13h,i). The conclusion that *BAP1* mutation status is not a relevant co-variant that affects survival outcome in the *HIF1A* loss cohort was also demonstrated by COX univariate (HR 1.839, 95% CI 1.332-2.539, $P = 0.00021$) and multivariate proportional hazards analyses (HR 1.776, 95% CI 1.274-2.474, $P = 0.00069$). These findings suggest that loss of one allele of *HIF1A*, which is predicted to lead to diminished HIF-1 α abundance, may be selected for during the evolution of some ccRCC tumours and that this correlates with aggressive disease.”

Minor comments

Authors should comment on the potential impact of intron read through of *Vhl* following *LoxP* recombination.

We have addressed this issue with the addition of the following text in the Results on p7. “Since the intronic sequencing reads do not start at the same position in different tumour samples, it is difficult to assess the effect of this read-through on potential translation of the resulting mRNA transcript, however western blotting of primary cells derived from *Vhl*^{fl/fl} mice demonstrated that Cre-mediated recombination results in complete loss of the pVHL protein isoforms (¹- and Supplementary Fig. S5).”

The authors state that CA9 is a H1 target gene, but the evidence provided is not compelling, and it is unclear that this is the case in humans. They are uniquely poised to address this and it would be good to examine this systemically across all the tumors they have generated.

This systematic analysis is in fact exactly what had done, but we unfortunately did not clearly state this in the original manuscript. We did not specify the number of tumours of each genotype that we analysed (we stated this only for the H&E analyses) and that the images shown in Figure 1D are representative of. We have now added this information into the Results (p7) and in Fig. 1. We also show now in Supplementary Fig S3c the mRNA abundance of Car9 in the different tumour genotypes, which further emphasizes that in this model Car9 expression is induced by HIF-1 α and not by HIF-2 α .

Would recommend including S5A in the main report and changing the colors of the dots to facilitate analysis.

Thank you for this suggestion, we have moved the PCA plot from Suppl S5A to Figure 3A. We have also taken the opportunity to alter the colours throughout all figures to harmonise the colours for each genotype. We have now chosen colours that are more readily distinguishable, particularly for readers with red-green colour blindness.

Authors should also comment on the fact that PC2 accounts only for 9% of the variability.

By definition the PC2 axis accounts for less variability than the PC1 axis. In this case, the PC1 axis separates the tumours from normal kidney tissue and this accounts for 36% of the variability while the tumours (partly) separate from one another along the PC2 axis which accounts for 9% of the variability, meaning that the tumours are more similar to one another in terms of their overall transcriptomes than they are distinct from the wild type. This is perhaps not surprising given that all tumour genotypes have *Vhl*, *Trp53* and *Rb1* mutations. These PCA differences are directly reflected in the numbers of genes that we identify as being differentially expressed when comparing the tumours to normal versus the tumours to other tumour genotypes. We have added the following text in the Results (p9) to address this issue:

“Principal component analysis (Fig. 3a) also revealed clear separation of WT cortex from all tumour samples on the PC1 axis and this accounted for 36% of the overall variability. *Vhl* ^{Δ/Δ} *Trp53* ^{Δ/Δ} *Rb1* ^{Δ/Δ} tumours and *Vhl* ^{Δ/Δ} *Trp53* ^{Δ/Δ} *Rb1* ^{Δ/Δ} *Hif2a* ^{Δ/Δ} tumours tended to segregate from one another on the PC2 axis, which represented 9% of total variability, suggesting that they are the most distinct in terms of gene expression patterns, whereas *Vhl* ^{Δ/Δ} *Trp53* ^{Δ/Δ} *Rb1* ^{Δ/Δ} *Hif1a* ^{Δ/Δ} tumours were more widely distributed along the entire axis. These analyses are consistent with the deletion of *Vhl*, *Trp53* and *Rb1* in all three tumour genotypes inducing large transcriptional changes, with more limited and specific contributions of HIF-1a and HIF-2a to the regulation of specific sets of genes.”

Reviewer #2

The selection of proteomic protocols for sample preparation, LC-MS/MS analysis, data interrogation is very solid, but requires corrections / clarifications. The authors used state of the art sample preparation technique, popular TMT-labeling technique for quantitation, advanced 2D-LC peptide separation procedure, robust MS instrument. I found very convincing the comparison between quantitative proteomic analyses of human and mice ccRCC. At the same time the description is not sufficiently clear.

We are most grateful to the reviewer for their careful consideration of our study and for their helpful suggestions which have improved the accuracy and completeness of our reporting of studies.

Minor corrections:

(page 22).

1) Please specify exact type of TMT labeling chemistry used: "TMT-10plex" or "TMT-6plex"

This information has now been included.

2) please specify the sample distribution through 10 channels: e.g 2(WT) – 2 – 2 – 2 (tumour) – 2 (pooled control samples) ?

This information has now been included. Due to complexity of the labeling scheme (3 batches) we now included the statement in the manuscript: "A summary of the labeling scheme can be found online as part of the ProteomeXchange submission".

3) Description of the project at Proteome Exchange PXD016630 lists TMT6-plex as labeling chemistry. This is unclear how 24 samples can fit three 6-plex datasets.

TMT11-plex (which we have used) is an isobaric extension of TMT6-plex. Due to their isobaric natures, many online databases for protein modifications (such as unimod) do not explicitly list TMT11-plex. This has hindered us in correctly listing TMT11-plex during data submission. We have discussed this issue with the PRIDE team. Their response was that an ontology term for TMT11plex has not yet been established. For this reason, we have kept the TMT6plex annotation and better explain the issue on page 27.

4) Thermo – Thermo Fisher Scientific.

This has been corrected.

5) "Waters" This is unclear why in one case in this section the state/country of origin is provided and in the others is not.

This has been corrected.

6) "XBridge C18 column, 150 mm x 4.5 mm column containing 3.5µm particles (Waters)" – "XBridge C18, 3.5µm, 150 mm x 4.5 mm column (Waters)".

This has been corrected.

7) Please provide gradient slope for high pH RP separation: e.g. "2% acetonitrile per minute increase" and number of fractions used for the analysis. It is very hard to judge on the quality of proteomic identification /quantitation without knowing overall analysis time / number of fractions used.

This information has now been included.

8) "...linear gradient of increasing buffer B (0.1% formic acid in acetonitrile, Fluka)..." – please provide composition of buffer A and clarify the description: e.g. "Both eluents A (water) and B (acetonitrile) contained 0.1% formic acid. Gradient program consisted of following steps: linear 2-25% B increase in 60 minutes and 25-60% B in 30 minutes, providing 90 min separation window at 300 nL/min flow rate."

This information has now been included.

9) Which nano-LC system has been used?

This information has now been included. EASY nano-LC system 1000 (Thermo Fisher Scientific, Waltham, MA, USA).

10) The description of DDA parameters isn't complete. It usually includes resolution for both MS and MS/MS, maximum IT (injection times), target AGC level, collision energy for MS/MS. Please refer to some previous publications using QE Plus instrument.

This information has now been included.

11) "Q-Exactive Plus" – "Orbitrap Q-Exactive Plus"

This has been corrected.

12) Proteome Exchange submission PXD016630 does not provide sufficient details for the researchers who want to use authors data: it has 24 raw files and one output MaxQuant file – all in duplicates. Fraction #s for each of three experiments and associated with each run distribution of samples throughout the channels should be available for Proteome Exchange users.

Thank you for pointing this out. A summary of the labeling scheme and corresponding raw file names has been added to the ProteomeXchange submission. We believe that this file describing the samples using the nomenclature of the manuscript allows interested scientists to access all of the raw data.

Reviewer #3

This is an important paper contributing to the long-standing discussion on the importance of HIF 1 vs HIF2 in ccRCC initiation and progression. Although the field is leaning more towards HIF2 being an oncogene and HIF1 a tumor suppressor, there are data supporting HIF1 as an oncogene, and the discussion continues to be a hot topic. This paper utilizes a mouse model of ccRCC, which relies on the loss of VHL tumor suppressor along with p53 and pRb. The data acquired in that model are complemented with the data from shRNA knockdowns of HIF1. Overall the paper is interesting and well written. It experimentally addresses the discrepancies in experimental data acquired with ccRCC cell lines +/- VHL and +/- HIFs in vitro and mouse models of ccRCC in vivo, including human cell line-based models and their own mouse model in the current study, in terms of the influence of HIF1 and HIF2 on tumor initiation and progression. The paper further investigates the gene expression signatures and proteomic alterations when HIF1a is deleted vs HIF2a is deleted from the VHL-/-p53-/-pRb-/- tumors. To be accepted for publication, the paper needs to be revised to better interpret the acquired results; more experimental support needs to be provided for the involvement of HIF-mediated immune infiltrates in ccRCC progression, as well as some additional comments below addressed. The reviewer highly encourages to submit the revised manuscript.

We are most grateful to the reviewer for their positive comments about our study and for their careful consideration and the helpful suggestions. We have fully addressed all but two of these suggestions (which are either not possible for technical reasons or would require several years of further study) and believe that all of these alterations have greatly improved our study.

Summary of new data/figures:

Figure 1d: Addition of new HIF-2 α immunohistochemical stainings

Figure 3a: Moved PCA analyses from previous Suppl. Fig.

Figure 5: Entirely new figure with new analyses of MHC class I and II in mouse and human ccRCC

Figure 6a: New figure. Addition of two new methods of bioinformatic immune cell deconvolution.

Figure 6e-k: New quantifications of stainings of 7 additional immune cell markers

Figure 7c: Addition of two new methods of bioinformatic immune cell deconvolution and new analyses based on tumour segregation based on *HIF2A* mRNA expression

Suppl. Fig. 3: New figure. Analyses of *Hif1a*, *Hif2a* and CA9 mRNA expression.

Suppl. Fig. 6a-c: New figure. Description of RNA-seq read trimming, mapping and absence of batch effects.

Suppl. Fig. 7: New panels d and e. Comparison of human and mouse HIF-2 α -dependent genes.

Suppl. Fig. 8: Moved GSEA enrichment graph from previous Figure to panel a, added new mRNA analyses of T cell activation genes in panel b.

Suppl. Fig. 9a-d: New figure. Correlations between proteome and transcriptome.

Suppl. Fig. 11: New figure. Shows examples of tumours with high and low abundance of cells staining for the different immune cell markers. Shows a better description of the methods of automatic quantification.

Suppl. Fig. 12a-h: New figure. Series of analyses related to Figure 7.

Suppl. Data 3: Added 8 new tabs to the previous Excel sheet (previously Suppl. Table 3 which showed only the proteomic expression values) describing the results of two new methods of gene set enrichment analyses of our proteomic data.

Suppl. Data 4: Analyses of MHC class I and II mRNA correlations to *HIF2A* mRNA abundance in human ccRCC.

Suppl. Data 5: New table describing the gene sets used for immune cell bioinformatic deconvolution.

Suppl. Data 6: Clinical co-variants of *HIF1A* copy number loss in human ccRCC

Major:

1. Although the study did a good job showing that HIF1a is important for tumor initiation, no data were provided to address the issue as to why HIF1a is lost during tumor progression in 30% of patients. It would be helpful to see the data when HIF1a deletion is introduced at later stages of tumor progression in this mouse model and the tumor growth assessed, ideally with analysis of immune microenvironment (see comments on claims below).

We agree with the thought that our current study now raises the follow-up question of whether there is a difference in the role or requirement of HIF-1 α at early versus later stages of tumour formation. However, the suggestion to address this question experimentally is unfortunately not possible using this experimental model, or indeed any other currently existing lines of mice. Our mouse model relies on inducible Cre activation which necessarily leads to the simultaneous deletion of either 3 or 4 homozygously floxed genes. Thus, *Hif1a* deletion cannot be temporally separated from the deletion of the *Vhl*, *Trp53* and *Rb1* genes. The experiment suggested by the reviewer would require for example generating some sort of new FRT-flanked *Hif1a* mouse line, as well as some sort of a new *Ksp-FLP* that is inducible by another stimulus than

Tamoxifen and then interbreeding these new alleles into the *Ksp-CreERT2*, *Vhl*, *Trp53*, *Rb1* background and then getting all 6 alleles to the appropriate homozygous state with the correct transgenes, then conducting all of the several year-long tumour forming studies with all of the associated follow-up analyses. This is the work of about 5 years.

2. The immunologic part of the study is very interesting, but seems incomplete in terms of data acquisition and data interpretation.

We believe that our completely new series of analyses provide a lot more information in terms of the analyses of the mouse and human tumours. Since other reviewers also commented about the importance of using independent methods (different sets of gene signatures) of bioinformatic deconvolution of the immune microenvironment we have conducted an entirely new set of analyses using our original sets of signatures (Bindea&Others), using the signatures of the CIBERSORT approach (or ImmCC, which is essentially CIBERSORT using mouse-specific matrices of genes) and using the eTME signatures that were defined by Wang et al Cancer Discovery 2018. A comparison of the genes used in the Bindea&Others and eTME signatures has now been provided in Supplementary Data 5. Both of these signatures were previously used to deconvolve the immune microenvironment of human ccRCC using a ssGSEA approach and we therefore utilised the mouse orthologues of these genes for our analyses. While the Bindea&Others and the eTME methods analyse the relative enrichment (z-scores) of gene sets that are specifically expressed in different subsets of cells, the CIBERSORT/ImmCC method uses a matrix-weighted score based on both high and low expressed genes in each immune subset to assess relative abundance of the population via a z-score. We have now summarised all the z-scores from these three analyses and their statistical significances in a new Figure 6a and Figure 7n in which the terms have been grouped based on the immune subtype that they correspond to, to facilitate comparisons between the three methods. Our new sets of conclusions about the effects of HIF-1a and HIF-2a loss in mouse and human ccRCC are based on the consensus of these signatures and are presented in extensive alterations to the Results section. In addition to our previous analyses of CD3, CD4 and CD8, we have also added a new series of IHC stainings for different cell types/activation states (MHC-II, PD-1, CD69, Perforin, B220, CD68, F4/80, Ly-6G) to gain a better overview of the immune microenvironment in our tumour models (Fig. 5, Fig. 6). We also compare our results to other findings in ccRCC. This is reflected in extensive alterations and additions to the Results and Discussion sections.

a) It is well accepted that inflammation is a hallmark of cancer promoting tumorigenesis; at the same time infiltration of certain immune cells is harmful to the tumor and is exploited for immunotherapies. The study falls short at separating these two branches of immune microenvironments. It should be revised to make the message clear. For all of the immune cell types found in this study which are affected by HIF1 and/or HIF2 expression, the data interpretation is largely absent: which cell types promote tumor growth, which suppress, in terms of initiation vs progression, how are HIFs regulating the microenvironment?

Thank you for this helpful comment. We have now added a section in the Discussion section (p19) that attempts to briefly put our new findings and conclusions into the context of the known roles of the identified cells in ccRCC or in other tumour types.

“A major advantage of the study of autochthonous models of ccRCC is that the tumours develop in the presence of a functioning immune system. In general, the activities of cytotoxic innate and adaptive immune cells can act to inhibit or delay tumour formation and progression, but an inflamed tumour microenvironment can also promote tumour formation through a variety of mechanisms⁷². The immune microenvironment of human ccRCC appears to be unusual in some respects as higher levels of T cell infiltration have been shown to correlate with disease recurrence and worse survival^{43,73,74}. In almost all other tumour types the degree of T cell inflammation is a good prognostic factor¹⁴. ccRCC tumours are also frequently inflamed with immature myeloid lineage cells that at least partly reflect different states of differentiation along the path from myeloid progenitor to differentiated dendritic cells, macrophages and neutrophils. At least some of these myeloid lineage cells have been proposed to promote tumour formation by suppressing T cell activation⁷⁵⁻⁷⁷. Our bioinformatic immune deconvolution and immunohistochemical analyses of the immune microenvironment of mouse ccRCC tumours is broadly consistent with this latter observation, namely that the immune microenvironment is dominated by myeloid lineage cells with a modest degree of T cell inflammation and activation. An unexpected finding was that tumours with *Hif2a* deletion showed stronger mRNA signatures associated with tumour immune cell infiltration, antigen presentation and interferon activity, as well as higher densities of CD8 T cells and cells expressing the T cell activation markers CD69 and Perforin, compared to the other two tumour genotypes. Furthermore, MHC class I and II genes, as well as other genes involved in antigen processing and presentation, were upregulated and tumour cells in *Vhl*^{Δ/Δ}*Trp53*^{Δ/Δ}*Rb1*^{Δ/Δ}*Hif2a*^{Δ/Δ} tumours were more frequently immunohistochemically positive for MHC class II expression, suggestive of increased presentation of

intracellular and extracellular epitopes in this tumour genotype. Collectively, these observations reflect a generally higher degree of immune activity directed against *Hif2a*-deficient tumour cells and it is plausible that this immune control may at least partly contribute to the reduced numbers of tumours and delayed tumour onset seen in this genotype. Interestingly, we demonstrated not only in $Vhl^{\Delta/\Delta}Trp53^{\Delta/\Delta}Rb1^{\Delta/\Delta}Hif2a^{\Delta/\Delta}$ tumours but also in $Vhl^{\Delta/\Delta}Trp53^{\Delta/\Delta}Rb1^{\Delta/\Delta}Hif1a^{\Delta/\Delta}$ tumours that the density of intra-tumoural CD8⁺ T cells were approximately double the density in $Vhl^{\Delta/\Delta}Trp53^{\Delta/\Delta}Rb1^{\Delta/\Delta}$ tumours. The densities of CD8⁺ T cells observed in the mouse ccRCC tumours (approximately 40-80 cells/mm²) are within the range seen in most cases of human ccRCC (approximately 20-160 cells/mm²)⁴⁹, providing support for the relevance of our mouse model. Consistent with our findings of HIF-2 α acting as a suppressor of T cell inflammation in mouse ccRCC, it was shown that HIF-2 α expression levels in human ccRCCs anti-correlate with T cell abundance and markers of T cell activation⁴⁹. We speculate that this apparent suppression of anti-tumour immune responses by HIF-2 α might reflect a mechanism of positive selection that maintains HIF-2 α expression in ccRCCs. Moreover, we show that loss of one copy of *HIF1A* correlates with a worse survival outcome, higher mRNA signatures of T cell abundance and a broadly altered immune microenvironment in human ccRCCs. This observation is consistent with the fact that higher levels of T cell infiltration have been shown to correlate with disease recurrence and worse survival in ccRCC^{43,73,74}.”

b) The transcriptomic study operates with tumor material containing a mixture of tumor cells and the cells in the microenvironment, pooling these together. It is assumed that the immune signatures derive from the infiltrated immune cells, but HIFs and VHL/p53/Rb are manipulated in tumor cells. There is no mechanism provided for the seen phenotype, please speculate on potential mechanism or provide additional data.

We were unable to elucidate the underlying mechanisms, however in the last section of the Results and described in the original Suppl. Fig S8 (now Suppl. Fig S13) we investigated some first experimental hypotheses that might potentially account for these differences, but these unfortunately did not reveal any interesting leads. We had already speculated in the Discussion section about other relevant factors that might be important to investigate in future studies and have now expanded upon these speculations, incorporating the conclusions of our new, more expansive analyses of the tumour microenvironment, p20:

“Thus, genetic and likely transcriptional and translational mechanisms that alter the balance of HIF-1 α and HIF-2 α abundance and activities appear to affect T cell inflammation. The mechanisms that underlie the increased T cell infiltration and/or activity in the absence of HIF-1 α or HIF-2 α will require further study. Analyses of RNA sequencing revealed that the mRNA levels of many cytokine-encoding genes are upregulated in tumours compared to normal cortex, but none of these genes were differentially regulated by HIF-1 α or HIF-2 α . Our functional tests to investigate if T cell proliferation might be suppressed by ccRCC cells in general, for example through the secretion of immunosuppressive glycolytic metabolic products such as lactate or H⁺⁷⁸, did not reveal any crosstalk between mouse or human ccRCC cells and mouse CD8⁺ T cells under cell culture conditions. It is possible that these simplified assays failed to reproduce the metabolic conditions that are present *in vivo*. It is also likely that many other complex factors such as the presence and activation states of other immune microenvironmental cells (such as macrophages, MDSCs, dendritic cells, regulatory T cells), antigen presentation and presence or absence of co-activating or inhibitory ligands by ccRCC cells, as well as the composition of the extracellular matrix might all play a role in the trafficking and activation of T cells.”

c) Please discuss the human ccRCC tumor infiltration by tumor-associated immune cells for patients with HIF1-only, HIF2-only, and HIF1/HIF2 tumors if the data is available.

While there are no datasets of human ccRCC for which there are both IHC data of HIF-1 α and HIF-2 α status as well as RNA seq data available, we tried to address this question about potential differences in the tumour microenvironment as a consequence of altered HIF-1 α and/or HIF-2 α activity in our experiments in Fig. 7 (formerly Fig. 6) based on gene copy number. We provide an entirely new series of analyses of human ccRCC based on immune deconvolution in samples that are segregated based on their mRNA expression levels of *HIF2A*. This is described in the following sections:

Results, p15: “To investigate whether *HIF1A* loss correlates with altered inflammation, we first demonstrated that *HIF1A* loss tumours exhibit on average between 1.9- and 2.1-fold higher levels of mRNA of *CD3D*, *CD3E*, *CD8A* and *CD8B* and 1.4-fold higher levels of *CD4* than unaltered tumours (Fig. 7d,f,h,j,l), suggesting that this group of tumours has higher CD8⁺, and to a lesser extent CD4⁺, T cell infiltration. This observation is consistent with the mouse analyses in which *Hif1a*-deficient tumours on average display approximately double the number of CD8⁺ and CD4⁺ T cells. In contrast, *HIF2A* gain tumours show no differences in the expression of any of these T cell marker genes compared to unaltered tumours (Fig. 7e,g,i,k,m). To gain a more in-depth overview of the effects of *HIF* gene copy number or expression level alterations on the immune microenvironment, we performed immune deconvolution analyses of RNA-seq data, again using three independent methods of immune cell deconvolution; ssGSEA using the

Bindea&Others and eTME gene signatures and using the CIBERSORT method ⁴⁸. We compared *HIF1A* loss and *HIF2A* gain tumours to diploid human ccRCCs (Fig. 7n) and also took advantage of the wide distribution of mRNA expression levels of *HIF2A* to compare tumours in the top (Q4) and bottom (Q1) quartiles of *HIF2A* mRNA abundance. While *HIF2A* gain tumours exhibited very few alterations in immune scores, *HIF1A* loss tumours showed statistically significant increases or decreases in 57 of 83 immune signatures of a variety of lymphoid and myeloid lineage cells. Notable amongst these are consistently upregulated scores for T helper cells and for B cells, mirroring our immunohistochemical findings of the comparison of *Vhl*^{Δ/Δ}*Trp53*^{Δ/Δ}*Rb1*^{Δ/Δ} and *Vhl*^{Δ/Δ}*Trp53*^{Δ/Δ}*Rb1*^{Δ/Δ}*Hif1a*^{Δ/Δ} tumours. It should however be noted that the magnitude of the z-scores are generally low, suggesting that this group of ccRCCs on average exhibits numerous subtle differences in immune inflammation compared to tumours with normal *HIF1A* copy number. In contrast, high *HIF2A* mRNA expressing tumours displayed downregulation of scores for interferon-γ and for APM2 (measuring MHC class II antigen presentation), consistent with the upregulation of these features in mouse ccRCC tumours lacking HIF-2a. Somewhat paradoxically, high *HIF2A* expressing tumours also displayed general upregulation of CD8 T cell scores and some NK cell scores and downregulation of all three scores for regulatory T cells and for the immune checkpoint proteins PD1 and CTLA4. These scores might be predicted to reflect elevated CD8 T cell activity. However, these tumours also display elevated scores for monocytes, neutrophils and mast cells, which in some settings contribute to suppression of anti-tumour CD8 T cell responses. Immunosuppressive mast cells were shown to correlate with *HIF2A* mRNA abundance in human ccRCC ⁴⁹. Thus, the extent of T cell mediated anti-tumour immunity is likely to be determined by the balance of the abundance and activities of several different immune cell types in a manner that is partly influenced by *HIF2A* expression. Finally, it is also noteworthy that *HIF1A* copy loss and *HIF2A* mRNA high tumours showed opposite effects on signatures for pericytes, endothelial cells and angiogenesis, implying that both HIF-1a and HIF-2a may act as positive factors that promote blood vessel formation in ccRCC tumours.”

Discussion, p20: “Indeed, immune deconvolution analyses revealed that the group of tumours that exhibit loss of one copy of *HIF1A* or that show high levels of mRNA expression of *HIF2A* show many differences in signatures of different types of cells in the immune microenvironment. Since our own studies comparing multiple bioinformatic immune cell deconvolution methods showed that different methods and different signatures can lead to different results for the same immune cell type, as well as the fact that we observed both consistencies and inconsistencies between bioinformatic predictions and direct immune cell enumeration using immunohistochemistry, it will be important to treat these bioinformatic observations as hypothesis-generating starting points that will need to be orthogonally tested by staining for specific immune cell markers in larger cohorts of human ccRCC tumours.”

c) Speculate on checkpoint inhibitors as therapeutics for HIF1-only, HIF2-only, and HIF1/HIF2 tumors based on your findings.

We agree that this is an interesting point to explore, but are also reluctant to engage in too much speculation that is not based on strong evidence about the likely effects on checkpoint therapy responses. We also believe that there are generally still far too many unknown factors that contribute to therapeutic responses in ccRCC to be able to make any meaningful predictions. Our findings may however prompt further studies about this issue and we have included a sentence to this effect in the Discussion stating that further studies will be necessary to figure out if the status of HIF-1α or HIF-2α have a bearing on the response to immune modulatory therapies.

“Given the clinical importance of immune checkpoint-based therapies for ccRCC and the fact that not all patients respond to these regimes, we believe that further investigation of the relationship between HIF-1α and HIF-2α status and the immune microenvironment is of potential therapeutic relevance. A further corollary of our findings is that inhibition of HIF-1α and HIF-2α transcription factor activities in ccRCC cells could be investigated therapeutically to inhibit tumour cell proliferation and simultaneously to attempt to increase T cell infiltration and activation. The potential direct effects of pharmacological inhibition of HIF-α factors on different immune cells would also have to be considered in this strategy.”

d) In the context of the findings, it is important to conduct tumor growth experiments with inhibitors or activators of specific branches of immune system in mice with deletion of HIF1 or HIF2 of HIF1WT/HIF2WT to clear the message of the paper about the role of HIF1 and HIF2 in immune response and tumorigenesis. Currently the message is confusing and lacks experimental support.

This suggestion is interesting and we agree that for ccRCC (and for many other tumours) that the precise roles of different immune cells in tumour formation and progression and therapeutic responses is unclear and will be important to clarify, particularly with a view to improving immune checkpoint therapy approaches. However, this rather open-ended experimental suggestion would also represent many years of

work and goes well beyond the scope of the current study. We are in fact conducting various therapy studies and these are incredibly labour intensive and turn into massive studies when using only the VpR genotype and just a few therapeutic agents. One major challenge of therapy experiments with our autochthonous mouse model is that the tumours arise with a long latency and incomplete penetrance, meaning that very large initial cohorts are required, every single mouse has to be imaged on a monthly basis using ultrasound or CT scanning to identify on a mouse-by mouse basis when tumours begin to form and then initiate therapy for each animal, switching to weekly monitoring using MRI. These experiments literally take years. The suggestion to do this in three different genetic backgrounds, one of which (VpRH1) almost completely rescues the tumour phenotype, using a series of different manipulations of different branches of the immune system is unfortunately not feasible. The mixed genetic background (which was necessitated by the backgrounds of the different Cre and floxed alleles that were available) also prevent syngeneic xenograft experiments which would potentially have simplified the task. We are currently working on new engineered model systems to try to facilitate these types of experiments in the future.

e) The data in Figure 3H&I are not consistent with Figure 4G. In figure 3 it seemed that HIF1 was promoting IFN production and HIF2 inhibiting it, in Figure4 VHL deletion was not providing a change, HIF1 and HIF2 were doing the same (inhibiting IFN response): please comment on these conflicting data to reconcile the discrepancies.

This interpretation is not quite consistent with our data. We agree that it appears in Fig 3i (now Supplementary Fig. 8b) that there is less expression of genes in the IFN- γ production GSEA term in the VpR1 tumours compared to the VpR tumours, however, the GAGE analyses (Supplementary Data 2 and Supplementary Fig 8a) revealed that there are no statistically significant differences between these terms. On the other hand, VpRH2 tumours appear to have higher expression of these genes compared to VpR tumours and the GAGE analyses revealed that the gene set was even more strongly enriched in VpRH2 compared to VpRH1. These same conclusions are also supported by the ssGSEA immune deconvolution analyses shown originally in Figure 5B, now Figure 6a.

The data in Figure 4g is based on proteomic analyses, where the coverage of proteins is much less than the coverage of mRNAs. However, as shown in original Suppl Fig S6A, which compares WT Cortex to VpR tumours, proteomics gene set enrichment analyses also identified IFN- γ signaling as an upregulated term. Figure 4f shows all of the proteins that were identified as being differentially abundant between VpRH1 and VpR tumours and IFN- γ was not identified as a GAGE term. In contrast, IFN- γ was one of the GSEA terms identified amongst the proteins that were statistically more abundant in VpRH2 compared to VpR tumours. Thus, the proteomic data are completely consistent with two different methods of analysis of the transcriptomic data in this respect. Prompted by our new analyses of the tumour immune microenvironment, we include in the Results and Discussion sections new descriptions and interpretations of the observed IFN signatures.

f) In figure 6 there does not seem to be an advantage of HIF2 expression in terms of immune cell infiltration (E)? Please interpret the findings.

As described for point 2, we have completely re-run and expanded the immune deconvolution analyses and adapted our conclusions.

3. Please comment in text of the paper on the mouse model of ccRCC used. It is very well known that VHL deletion is not sufficient for ccRCC development in mice. Thus additional genetic events are necessary for the mouse model to develop disease. It was stated in the paper that p53 loss alleviates senescence. The caveat is that p53 is generally not mutated in ccRCC.

Thank you, this is an important point to include to guide the readers in their understanding of the mouse model. We have now included a sentence in the introduction (p5) summarising the aspects of the disease that this model reflects (the main points of our previous publication describing the generation of this model). "This mouse model at least partly reflects the complex patterns of chromosomal copy number gains and losses of cell cycle regulatory genes in human ccRCC and reproduces many aspects of the evolution of human ccRCC by first developing cystic and solid precursor lesions that progress to tumours over the course of 5-12 months following gene deletion in adult mice¹⁶. »

4. Please discuss in paper the higher grades of the tumors when HIF1a or HIF2a were deleted compared to control (VHL-/-p53-/-pRb-/-).

Thank you for this suggestion, we have now included a section in the Results (p7) with some ideas about possible HIF-dependent biological alterations that might potentially underlie these observations.

“Since tumour grade up to grade 3 is classified mostly based on nucleolus size, these data hint that loss of HIF-1a or HIF-2a may modify processes such as transcription of ribosomal DNA genes that affect the nucleolus²⁸. Potentially relevant mechanisms that have been previously linked to HIF-a activities and that might contribute to nucleolar alterations include metabolic generation of ATP and deoxynucleotides to fuel transcription, epigenetic regulatory mechanisms and DNA repair²⁸.”

5. It seems important to validate the model with HIF2a antibody by IHC or by western blot. The abcam anti-HIF2a antibody should work for this purpose (worked in published studies on mouse cell lysates). If it does not, then several HIF2a specific proteins should be used instead (from the list determined in this study). We agree that this was a very important point to clarify and thank you for the antibody suggestion. We now tested six different commercially-available anti-HIF-2 α antibodies using two different antigen retrieval methods and found one condition (one of the two abcam antibodies that we tried) that gave specific nuclear signals for HIF-2 α in IHC stainings. (We have included the details of the antibody that we found to work, as well as those that do not work in our hands in the Methods section, to hopefully save other groups in the future the trouble that we have had in detecting HIF-2 α). Examples of these stainings are shown in Fig. 1d. Nuclear HIF-2 α staining was present in tumours from $Vhl^{\Delta/\Delta}Trp53^{\Delta/\Delta}Rb1^{\Delta/\Delta}$ (35/35 tumours) and $Vhl^{\Delta/\Delta}Trp53^{\Delta/\Delta}Rb1^{\Delta/\Delta}Hif1a^{\Delta/\Delta}$ (11/11 tumours) mice but absent in all tumours from $Vhl^{\Delta/\Delta}Trp53^{\Delta/\Delta}Rb1^{\Delta/\Delta}Hif2a^{\Delta/\Delta}$ (0/54 tumours) mice. This result demonstrates that HIF-2 α is deleted in the tumours that emerge in $Vhl^{\Delta/\Delta}Trp53^{\Delta/\Delta}Rb1^{\Delta/\Delta}Hif2a^{\Delta/\Delta}$ mice and validates our conclusion that HIF-2 α has relatively minor effects on tumour growth in this model.

Minor:

1. Please show the HIF1a expression in HIF2a deleted tumors and HIF2a expression in HIF1a deleted tumors to rule out the possibility of cross-regulation.

Thank you for this question. We show that there are no effects at the protein level in Fig. 1d and we have now added analyses in a new Supplementary Fig. 3, described in the Results, p6:

“We additionally analysed RNA sequencing data (see experiments described below) which showed that $Vhl^{\Delta/\Delta}Trp53^{\Delta/\Delta}Rb1^{\Delta/\Delta}$ tumours displayed lower mRNA levels of *Hif1a* and *Hif2a* than WT cortex but that there was no compensatory upregulation of *Hif2a* in $Vhl^{\Delta/\Delta}Trp53^{\Delta/\Delta}Rb1^{\Delta/\Delta}Hif1a^{\Delta/\Delta}$ tumours, nor of *Hif1a* in $Vhl^{\Delta/\Delta}Trp53^{\Delta/\Delta}Rb1^{\Delta/\Delta}Hif2a^{\Delta/\Delta}$ tumours (Supplementary Fig. S3a,b).”

2. Page 8 1st sentence, please provide a citation for the rescue experiment.

This sentence was in fact referring to our experiments described in the previous sections and not to another study. We have changed the sentence to make the meaning clearer.

3. “3T3 proliferation assays” are confusing since the assays are done here on MEFs. Please either justify the name of the assay or remove 3T3.

The 3T3 assay refers to the standard experimental protocol (seeding at 3×10^5 cells per 6 cm plate and splitting cells every 3 days and then repeating) used to measure proliferation of MEFs. However, we agree that this terminology could be a cause of confusion and therefore refer simply to proliferation assays (the methodology is contained in the Methods section).

4. HIF2 transcriptional network on page 9 states PGK1, later in the paper it is referenced as HIF1 and HIF2 gene (Figure S4).

Thank you for this comment. We presume that the reviewer is referring to the heatmaps in original Figure S5 (not S4). This observation highlights the general limitations of GSEA analyses, in that the gene sets that are placed in the databases are generated based on specific knowledge or experimental conditions and should not be viewed as a universal truth that applies to all conditions. While *PGK1* is listed in the HIF2 GSEA term, our statistical analyses of our datasets identified *Pgk1* expression as being sensitive to either *Hif1a* or *Hif2a* deletion. We agree that this may potentially be a cause of confusion and have now removed the reference to specific genes in the HIF2 GSEA dataset and as described in the answer to point 5 below, we have instead added a much more detailed analysis of HIF-2 α target genes.

5. Please comment on chronic (knockout) vs acute (HIF2 inhibitor PT2399) way to inhibit HIF2 and the absence of overlap of HIF2 target genes, top of page 10.

Thank you for this comment, we agree that this was an important issue to clarify. We have now conducted new analyses based on the findings and method reported in Courtney et al CCR 2020 that defined HIF-2 specific target genes based on RNA seq analyses of the effects of the HIF-2 inhibitors PT2385 or PT2399 on ccRCC tumorgrafts or patient metastases. We used the set of 277 genes defined by Courtney et al CCR 2020 as being dependent on HIF-2 α specifically in human ccRCC tumour cells (note that this does not

formally exclude that some of these genes might also be dependent on HIF-1 α) and analysed the expression levels of the mouse orthologues of these genes in our tumour RNA seq. This is described in the Results section on p10 as follows:

“To further investigate potential overlap with recently-defined HIF-2 α -dependent genes in human ccRCC, we used a set of 277 genes that were identified as being inhibited specifically in tumour cells in ccRCC tumourgrafts in mice treated with the HIF-2 α inhibitor PT2399^{1,2}. Analyses of the expression levels of the mouse orthologues of this set of HIF-2 α target genes revealed that many of these genes are highly upregulated in *Vhl* ^{Δ/Δ} *Trp53* ^{Δ/Δ} *Rb1* ^{Δ/Δ} tumours compared to WT cortex, but that the loss of either HIF-1 α or HIF-2 α did not broadly affect the up-regulation of these genes (Supplementary Fig. 7d). Nonetheless, 11 genes, marked in red in Supplementary Fig. 7d and shown in Supplementary Fig. 7e, were expressed at significantly lower levels (fold change <-1.7, *P*<0.05) in *Vhl* ^{Δ/Δ} *Trp53* ^{Δ/Δ} *Rb1* ^{Δ/Δ} *Hif2a* ^{Δ/Δ} tumours than in *Vhl* ^{Δ/Δ} *Trp53* ^{Δ/Δ} *Rb1* ^{Δ/Δ} tumours. The relatively small overlap between mouse and human HIF-2 α -dependent genes may be due to inherent differences between mice and humans, to the very different experimental settings of acute pharmacological inhibition versus tumour evolution in the genetic absence of *Hif2a*, or to specific features of this particular model of ccRCC. In this latter context, it is noteworthy that many human HIF-2 α -dependent ccRCC genes are related to the cell cycle and to DNA damage responses. These signatures are highly represented in the comparison *Vhl* ^{Δ/Δ} *Trp53* ^{Δ/Δ} *Rb1* ^{Δ/Δ} vs WT cortex (see GAGE signatures in Supplementary Data 2). We speculate that it is likely that these genes are not dependent on HIF-2 α in the mouse model due to the fact that the deletion of *Rb1* and *Trp53* already strongly affects these classes of genes.”

6. Page 17, HIF1a inhibition seems to be not feasible as a therapeutic approach due to ubiquitous expression. Please describe another feasible approach based on your data to target consequences of HIF1a overexpression.

We do not agree with the suggestion that HIF-1 α is not a feasible target for therapy based simply on the fact that it is more widely expressed in cells and tissues than HIF-2 α . There are numerous examples of successful cancer drugs that target fundamental cellular effectors that are important in almost all cells (e.g. MEK inhibitors, proteasome inhibitors, mTORC1 inhibitors). We have in fact shown proof of principle that combined inhibition of HIF-1 α and HIF-2 α using acriflavine has therapeutic effects in this mouse model of ccRCC (Harlander Nature Medicine, 2017) and others have used acriflavine in the settings of other cancer models. We nonetheless thank you for raising this question as it has prompted us to directly address this important point in the discussion section and reference these promising studies.

Discussion, p21: “Finally, while specific inhibitors of HIF-2 α are available and are currently being tested in clinical trials²⁵⁻²⁷, our findings demonstrating the importance of HIF-1 α for ccRCC formation argue that the development of specific inhibitors of HIF-1 α or of new specific dual HIF-1 α /HIF-2 α inhibitors would also be desirable and may have therapeutic benefit in ccRCC. Proof-of-principle that these approaches are likely to be tolerable and effective comes from previously reported therapeutic effects of Acriflavine, which inhibits the binding of both HIF-1 α and HIF-2 α to HIF-1 β ^{79,80}, in the *Vhl* ^{Δ/Δ} *Trp53* ^{Δ/Δ} *Rb1* ^{Δ/Δ} ccRCC model¹⁶, as well as in xenograft and autochthonous mouse models of several different types of tumours^{79,81-83}.”

7. Sections in Figure 1D seem to be non-serial, if possible, provide serial sections.

The images that we show are representative of the staining patterns that were observed in the different genotypes. Providing images from serial sections is unfortunately not possible as we have used about 30-50 sections from each kidney to test all sorts of different antibodies over a long period of time in the laboratory (many of the stainings are shown in the manuscript but many are not). Repeating every set of stainings on precious tumour material to obtain images from serial sections of many tumours is not feasible. On this point, we however have realised that we omitted to state the number of tumours for which the depicted stainings are representative and have now included this information in the new Fig 1d.

8. Figure 2: H, I, why did the mice die? If they were sac'ed due to tumor burden, please state in the figure and text.

Thank you for this question. Yes, they were sacrificed due to tumour burden. This is important information to include. We have now put this into the text as suggested.

Reviewer #4 (Remarks to the Author): expertise in transcriptomic (cancer and TME)

Hofflin et al. performed genomic studies and functional experiments to assess the functional consequences of Hif1a or Hif2a deletion in ccRCC models. While I find this work is interesting, there are however a number of issues (e.g. data quality, data processing, superficial analysis, etc.) that limit the overall impact of the study.

Thank you for your careful consideration of our study and for your helpful comments and suggestions. We have now undertaken a new series of analyses described below that we believe have greatly improved our study.

Summary of new data/figures:

Figure 1d: Addition of new HIF-2 α immunohistochemical stainings

Figure 3a: Moved PCA analyses from previous Suppl. Fig.

Figure 5: Entirely new figure with new analyses of MHC class I and II in mouse and human ccRCC

Figure 6a: New figure. Addition of two new methods of bioinformatic immune cell deconvolution.

Figure 6e-k: New quantifications of stainings of 7 additional immune cell markers

Figure 7c: Addition of two new methods of bioinformatic immune cell deconvolution and new analyses based on tumour segregation based on *HIF2A* mRNA expression

Suppl. Fig. 3: New figure. Analyses of *Hif1a*, *Hif2a* and CA9 mRNA expression.

Suppl. Fig. 6a-c: New figure. Description of RNA-seq read trimming, mapping and absence of batch effects.

Suppl. Fig. 7: New panels d and e. Comparison of human and mouse HIF-2 α -dependent genes.

Suppl. Fig. 8: Moved GSEA enrichment graph from previous Figure to panel a, added new mRNA analyses of T cell activation genes in panel b.

Suppl. Fig. 9a-d: New figure. Correlations between proteome and transcriptome.

Suppl. Fig. 11: New figure. Shows examples of tumours with high and low abundance of cells staining for the different immune cell markers. Shows a better description of the methods of automatic quantification.

Suppl. Fig. 12a-h: New figure. Series of analyses related to Figure 7.

Suppl. Data 3: Added 8 new tabs to the previous Excel sheet (previously Suppl. Table 3 which showed only the proteomic expression values) describing the results of two new methods of gene set enrichment analyses of our proteomic data.

Suppl. Data 4: Analyses of MHC class I and II mRNA correlations to *HIF2A* mRNA abundance in human ccRCC.

Suppl. Data 5: New table describing the gene sets used for immune cell bioinformatic deconvolution.

Suppl. Data 6: Clinical co-variants of *HIF1A* copy number loss in human ccRCC

Specific points:

Some of the main conclusions of this manuscript were drawn based on their RNA-seq data expression and pathway analysis. However, there is no detailed description in the Methods (RNA-sequencing paragraph) about the generation of their RNA-seq data, such as the kit and protocol used for sequencing library preparation, the platform used for data generation, and the detailed quality check metrics at sample level.

We have now added this information to the Methods section.

“RNA was isolated from powdered frozen samples of wild-type kidney cortex controls from Cre negative mice in the *Vhl^{fl/fl}Trp53^{fl/fl}Rb1^{fl/fl}* background and from tumours of the different genetic backgrounds using the NucleoSpin RNA kit (Machery Nagel). Paired-end RNA-sequencing was performed on an Illumina HiSeq4000 device by the core facility of the German Cancer Research Center (DKFZ) in Heidelberg with the Illumina TruSeq Stranded RNA library preparation kit.”

The authors mentioned “reads trimming” in their methods for RNA-seq data processing, was it because the poor quality reads in their data? The author should provide details on this, such as the base and read-level quality metrics of their RNA-seq data, how the reads trimming was performed? the proportion of reads trimmed per sample, the number of bases trimmed off? the author should evaluate whether the difference in sample quality, reads trimming, or other filtering process will impact their downstream expression and pathway analysis.

We have now included this data about read trimming and alignment in the Methods and in Supplementary Fig. S6a,b.

“Raw data fastq-files were pre-processed with trimmomatic⁸⁸ to assure sufficient read quality by removing adapters and bases in the low quality segment regions (end of the reads) with a base quality below 20. Before trimming the average number of reads was 48309915 ± 12246964 [26804116,70620322], after trimming the average number of reads was 45451780 ± 13975818 [20629675,70032834]. Hence, an

average of $93.2\% \pm 10.5\%$ of the raw reads survived the trimming step (Supplementary Fig. S6a). The overall quality of the bases and reads was good. After quality control and trimming the reads were 2-pass aligned using the STAR aligner⁸⁹ and the GRCm38 reference genome from Ensembl. $85.1\% \pm 3.3\%$ of the reads were uniquely mapped and considered (Supplementary Fig. S6b)."

also important, the author should describe the details how the RNA-seq data is normalized, were there any batch effects by statistical assessment, if present, how the batch effects were corrected and how the data was normalized and filtered. Without these details, it is hard to make any adjustment about the quality of their RNA-seq data and the reliability of their conclusions.

We describe the normalisation in the Methods "The alignment step was followed by normalization and differential expression analysis with the R/Bioconductor⁹⁰ package DESeq2⁹¹. The normalisation of the raw read counts was performed with DESeq2 by considering the library size. Additionally, all genes with a low count across all samples, i.e. the row sum of a gene was below 5 in a gene by sample matrix, were removed from the dataset. After pre-processing and filtering 19,723 genes were further analysed and fitted with a negative binomial generalised linear model followed by Wald statistics to identify differentially expressed genes. Genes were considered significant with an adjusted p-value < 0.001 (Benjamini-Hochberg).

We now show an analysis of potential batch effects (which were not observed) in Supplementary Fig. S6c. Results, p9: "Transcriptomic profile principal component analysis and unsupervised hierarchical clustering by sample Euclidean distance matrix (Supplementary Fig. S6c) suggested minimal batch effect amongst different sequencing runs."

For pathway GSEA analysis, the authors analyzed signaling pathways from Gene Ontology, ConsensusPathDB and MSigDB, it is known that some pathways from these different databases are functionally redundant (or most of the genes are overlapped), however, it is unclear from their methods, how these redundant pathways were processed and counted.

We chose to analyse several different databases of gene sets in order to obtain as wide a view as possible of potential biological processes that are reflected in the transcriptomics data. This included not only the consensus databases but also specific databases in MSigDB that are compiled from experimental interventions and from knowledge of transcription factor binding sites. It is true that there is some redundancy amongst these terms related to overlapping sets of genes. Keeping this in mind, and also the fact that GAGE analyses represent starting points to gain biological insight that should be validated by other methods (as we have done in the context of the further analyses of the TME), in order to provide the most information about our analyses to the scientific community we chose to present all statistically significantly enriched terms in Supplementary Data 2. Please also see comments below related to the question about pathway selection.

In addition, the authors only used adjusted p-values to select the significant pathways, however, the normalized enrichment scores should also be considered.

Our analyses were conducted using GAGE (Generally Applicable Gene-Set Analysis) which uses the normalised expression values of the genes, in comparison to GSEA which uses "only" a ranked list of genes (that could be either log₂FC or some kind of statistics) and involves the generation of normalised enrichment scores. For GAGE analyses the adjusted p-values (FDR-corrected p-values) are the statistically meaningful metrics. We only considered terms that showed adjusted p values of less than 0.05. These are the terms that are listed in Supplementary Data 2.

In their figures 3-4, multiple heatmaps are shown on different pathways. First, it is unclear how many pathways in total were studied?

The full information about all of the pathways is provided in Supplementary Data 2.

How these pathways were selected (based on which criteria)?

It is not feasible to follow up on all interesting observations that emerge from transcriptomic profiling data. In addition to looking for similar biological terms that occurred recurrently in the GAGE analyses, we focused on biological pathways that we believe are particularly relevant for cancer and ccRCC. We provide the full list of the GAGE terms in Supplementary Data 2 so that these analyses are available to the entire research community and in the belief that this information may potentially fuel studies by other groups that are more specialised in the study of additional aspects of this disease.

Are there additional pathways showed statistical significant? If so, what are those pathways?

Are those pathways functionally linked to the pathways displayed here?

See the answer to the question above and Supplementary Data 2 for a full list of all pathways.

Second, it doesn't seem like the authors displayed a full list of the genes included in each pathway, questions are: how these genes (listed in the figures) were selected? How about the rest of genes in each selected pathway?

The differentially regulated genes were selected for the purposes of visualisation, that is why we provided full data in Supplementary Data 2.

The authors used Single-Sample GSEA for immune deconvolution analyses, it is unclear how the immune related signatures are selected and whether this approach is validated and statistically sound. It is recommended that the author should also try a couple of other validated immune deconvolution approaches such as CIBERSORT, MCP-Counter, TIMER, etc. to ensure that their discoveries are reproducible.

[Thank you for this important suggestion. We believe that our completely new series of analyses provide a lot more information in this respect. Since other reviewers also commented about the importance of using independent methods \(different sets of gene signatures\) of bioinformatic deconvolution of the immune microenvironment we have conducted an entirely new set of analyses using our original sets of signatures \(Bindea&Others\), using the signatures of the CIBERSORT approach \(or ImmuCC, which is essentially CIBERSORT using mouse-specific matrices of genes\) and using the eTME signatures that were defined by Wang et al Cancer Discovery 2018. A comparison of the genes used in the Bindea&Others and eTME signatures has now been provided in Supplementary Data 5. Both of these signatures were previously used to deconvolve the immune microenvironment of human ccRCC using a ssGSEA approach and we therefore utilised the mouse orthologues of these genes for our analyses. While the Bindea&Others and the eTME methods analyse the relative enrichment \(z-scores\) of gene sets that are specifically expressed in different subsets of cells, the CIBERSORT/ImmuCC method uses a matrix-weighted score based on both high and low expressed genes in each immune subset to assess relative abundance of the population via a z-score. We have now summarised all the z-scores from these three analyses and their statistical significances in a new Figure 6a and Figure 7n in which the terms have been grouped based on the immune subtype that they correspond to, to facilitate comparisons between the three methods. Our new sets of conclusions about the effects of HIF-1a and HIF-2a loss in mouse and human ccRCC are based on the consensus of these signatures and are presented in extensive alterations to the Results section. In addition to our previous analyses of CD3, CD4 and CD8, we have also added a new series of IHC stainings for different cell types/activation states \(MHC-II, PD-1, CD69, Perforin, B220, CD68, F4/80, Ly-6G\) to gain a better overview of the immune microenvironment in our tumour models \(Fig. 5, Fig. 6\). We also compare our results to other findings in ccRCC. This is reflected in extensive alterations and additions to the Results and Discussion sections.](#)

For LC-MS based proteomics data analysis, as it is known that not all the proteins are covered well, similarly, not the full length of every protein is covered. And the expression of many gene at mRNA and protein level are not consistent due to other possible regulatory mechanisms. The authors tried to use the LC-MS data to support their discovery from RNA-seq data.

It is correct that the sensitivity of proteomics is less than that of RNA-seq and it is also correct that there are many post-transcriptional mechanisms that govern protein abundance independently of mRNA abundance. We therefore used proteomics as a separate unbiased approach to investigate potential biological differences between the tumour genotypes. This is described as follows in the text, with the new additions in bold.

"In order to further explore whether the biological alterations predicted by transcriptomic analyses are also reflected at the protein expression level, as well as to attempt to capture differences in the proteomes of the tumours that might not be reflected in their transcriptomes, **we used** exploratory quantitative proteomic analyses of 6 samples of WT cortex and 6 tumours each from \$Vhl\Delta/\Delta Trp53\Delta/\Delta Rb1\Delta/\Delta\$, \$Vhl\Delta/\Delta Trp53\Delta/\Delta Rb1\Delta/\Delta Hif1a\Delta/\Delta\$ and \$Vhl\Delta/\Delta Trp53\Delta/\Delta Rb1\Delta/\Delta Hif2a\Delta/\Delta\$ mice **as an independent discovery tool.**"

However, first, it is unclear that how many proteins were detected (and highly expressed) by LC-MS.

We had in fact provided this information in the text of the Results section and had also included this information in Supplementary Data 3. "These analyses allowed the quantification of 4257 proteins that were present in at least 4 of 6 samples of each genotype (Supplementary Data 3)."

Especially, the proteins involved in the pathways of interest.

We would like to emphasise that we have used proteomics as an independent discovery tool, regardless of pre-defined pathways, and not as a tool to validate RNA seq data. This point may have been unclearly expressed in the first version of the manuscript and has been altered as described above. We have kept the RNA-Seq and proteomics data separate for statistical analyses and related approaches such as GSEA.

If a protein showed no expression from LC-MS, was it because the protein was low abundant, or because the digestion approach they used was not appropriate (e.g. the enzyme used for LC-MS results in numerous of either too long or too short peptides that were not easily captured).

As rightfully pointed out by the reviewer, numerous factors may contribute to missing LC-MS/MS identifications for a given protein. These include its abundance on the protein level, presence of suitable digestion sites (for a suitable peptide length) for the digestion enzyme (here: trypsin), ionization and fragmentation characteristics of the peptides, presence (or absence) of post-translational modifications. This is an intrinsic limitation of explorative proteomics and its investigation remains beyond the scope of the present study. We therefore only further analysed proteins that were identified in at least 4 of the 6 samples of each genotype.

The correlation between protein and mRNA expression (for both low-abundant and highly abundant genes) showed be evaluated.

Thank you for this suggestion, which also relates to the comments above about the similarities and differences between the proteome and transcriptome. In a new Supplementary Fig 9), for each sample type we have now compared the protein and mRNA abundances of all detected proteins and their mRNAs, as well as compared the abundance of all differentially regulated proteins and their mRNAs between the different comparisons. All of these analyses show statistically significant correlations, with higher coefficients of correlation being observed for the differentially expressed proteins and their mRNAs. These analyses are described in the Results, p11:

“As is commonly observed in comparisons of proteome and transcriptome data, the overall correlations of protein abundance and mRNA abundance were low (Supplementary Fig. 9a). However, there were strong correlations between fold changes in mRNA abundance and fold changes in protein abundances when analysing only those proteins that showed differential expression between genotypes (Supplementary Fig. 9b-d).”

Figures 4F-G, author picked genes from multiple pathways and put them together in the heatmap, again, the problem is that how these genes were selected? An overview of GSEA results on all pathways should be displayed.

Thank you for this suggestion. These were all of the proteins that were statistically significantly differentially expressed between the genotypes, so there was no bias in the selection. We then analysed these genes by GSEA analysis to identify the pathways they are implicated in. However, we have now expanded our analyses and presentation of the results, described on p11 and p12.

“To characterise biological pathways that are altered in tumour compared to normal tissue, we conducted two complementary analyses; ROAST (rotation gene set testing) analysis³⁸ was used to assess gene set enrichment based on the expression levels of all measured proteins and gene set enrichment analysis using the online platform of MSigDB (<https://www.gsea-msigdb.org/gsea/msigdb/index.jsp>) was performed using only the lists of statistically differentially upregulated proteins. These analyses revealed many overlaps with one another as well as with GAGE gene set terms that emerged from the analyses of the transcriptome, including glycolysis, hypoxia, DNA repair, mTORC1 signalling, E2F and MYC targets and IFN γ response (Supplementary Data 3 and Supplementary Fig. S10a).

To compare the effect of the absence of HIF-1 α or HIF-2 α on the proteome, we first conducted principal components analysis (Supplementary Fig. S10b), which revealed that all tumour samples clustered separately from the WT cortex samples, but that the tumour samples of all of the different genotypes largely overlapped with one another, suggesting a relatively high degree of similarity in the overall protein expression patterns of tumours from the different genetic backgrounds. ROAST analyses as well as gene set enrichment analyses of the lists of proteins that are differentially expressed between Vhl Δ/Δ Trp53 Δ/Δ Rb1 Δ/Δ Hif1a Δ/Δ and Vhl Δ/Δ Trp53 Δ/Δ Rb1 Δ/Δ tumours (Fig. 4d) and between Vhl Δ/Δ Trp53 Δ/Δ Rb1 Δ/Δ Hif2a Δ/Δ and Vhl Δ/Δ Trp53 Δ/Δ Rb1 Δ/Δ tumours (Fig. 4d,e and Supplementary Data 3) both also highlighted numerous similarities to the transcriptomic analyses. HIF-1 α deficiency reduced expression of glycolytic enzymes and increased expression of proteins associated with oxidative phosphorylation and respiratory electron transport (Fig. 4f), while HIF-2 α deficiency reduced the expression of MYC targets and resulted in increased expression of genes associated with immune responses, interferon signaling, cytokine signalling and antigen presentation (Fig. 4g). In conclusion, the analyses of the proteomes strongly align

with the analyses of the transcriptomes, providing independent validation for the predicted biological differences between the tumour genotypes.”

Figure 1B, although the authors stated statistical difference between VpR and VpRH1 and VpRH2, the values of VpRH1 and VpRH2 were widely distributed along the entire y axis, indicating a high degree of heterogeneity. What are the possible molecular determinants of such a heterogeneity? What are other factors likely influence the molecular consequences of HIF-1a or HIF-2a deletion?

There may be many determinants of this heterogeneity. We note that there are morphological differences and differences in the observed densities of immune cells (seen in IHC stainings) and in bioinformatic deconvolutions between tumours within a given genotype that might partly account for the observed molecular heterogeneity. However, we are reluctant to speculate about this further in the manuscript as it would amount purely to speculation. We believe that rather than discussing potential tumour to tumour differences, it is more informative to focus on the molecular features of the tumours on average across the same genotype and their differences to the other genotypes, as we have done in the manuscript.

Figure 3D, there are many different DNA repair pathways, which pathway(s) in particular showed significant differences?

We have included this information in the Results section on p10: “Genes that were expressed at low levels in $Vhl^{\Delta/\Delta}Trp53^{\Delta/\Delta}Rb1^{\Delta/\Delta}Hif2a^{\Delta/\Delta}$ tumours compared to the other two tumour genotypes included those involved in different DNA repair processes (e.g. *Fancf* – DNA interstrand cross link repair, *Rad52* – homologous recombination repair, *Ogg1* – oxidative stress induced base excision repair, *Ercc2* – transcription coupled nucleotide excision repair) (Fig. 3e),...

Figure 3H, the author showed T-cell activation genes, however, some genes in this list, for example, *Foxp3*, *Visr*, are immune inhibitory. What are these genes here? Are they related to CD4 T cell or CD8 T cell activation? Similarly, in the panel I, the IFN- γ production gene list contains *Cd14*, *Cd276*, *Foxp3*, which raised a concern whether the author used the right gene signature for their analysis.

This comment of the reviewer highlights the general limitations of GSEA analyses, in that the gene sets that are placed in the databases are generated based on curated knowledge or are derived from specific experimental conditions and are therefore intrinsically limited or biased by the gene lists that are deposited. We have now tried to highlight this issue at two points of the manuscript to relativise the conclusions.

Results p11: “We conclude that these analyses suggest that there is a complex inflammatory response in $Vhl^{\Delta/\Delta}Trp53^{\Delta/\Delta}Rb1^{\Delta/\Delta}$ tumours that is further modified by HIF-2 α deficiency. These phenotypes were further investigated in experiments described in the following sections.”

Discussion, p20: Since our own studies comparing multiple bioinformatic immune cell deconvolution methods showed that different methods and different signatures can lead to different results for the same immune cell type, as well as the fact that we observed both consistencies and inconsistencies between bioinformatic predictions and direct immune cell enumeration using immunohistochemistry, it will be important to treat these bioinformatic observations as hypothesis-generating starting points that will need to be orthogonally tested by staining for specific immune cell markers in larger cohorts of human ccRCC tumours.”

Figure 4F, the author showed “HIF-1a-dependent” genes in the heatmap, however, some of these genes showed similar expression levels between both WT-Cortex and VpRH1 (the bottom part of 4F), such as *Apool*, *Mzb1*, *C1qbp*, etc. Based on what evidence, that the authors determined that these genes are HIF-1a “dependent”?

This heatmap shows all proteins that passed fold change and statistical significance cut-offs, either being downregulated or upregulated in VpRH1 compared to VpR. We use “dependent” to refer to proteins whose expression is affected up or down in the comparison between the tumour genotypes.

Figure 5A, how was the immune infiltration score was calculated? Is this relative or absolute abundance?

The immune scores came from using published gene signatures and were deconvolved by ssGSEA. For each feature, the z score is derived for each sample using the mean and standard deviation based on all of the samples.

Again, it is strongly suggested that multiple immune deconvolution approaches should be used to ensure that the finding is reproducible.

This has now been done, please see the answer to a previous question listed above.

The authors used IHC staining to show the density of CD3, CD4 and CD8 positive T cells. but the number of markers used is very limited. It would be great to add several more markers showing T cell cytolytic activity and myeloid cell states.

Thank you for this question. We have now added a series of new markers, described in the Results on p13: “To further characterise the immune microenvironments of the three different tumour genotypes, we next conducted immunohistochemical stainings for a series of markers of different types of immune cells to permit analyses of a larger set of tumours of each genotype (n = 14-26 tumours). We stained sections of whole tumour-bearing kidneys with antibodies against CD3 to label T cells, CD4 to label helper T cells, CD8 to label effector T cells, CD69 as an early activation marker of T cells and NK cells, Perforin to label activated cytotoxic T cells and NK cells, PD-1 to label antigen-exposed activated or exhausted T-cells, B220 to label B cells, CD68 to label monocytes and macrophages, F4/80 to label differentiated macrophages and Ly-6G to label granulocytes and neutrophils. These markers revealed considerable inter-tumoural heterogeneity in terms of the density of infiltrating cells, even within the same kidney (Supplementary Fig. S11a-j). We quantified the densities of positively stained cells either by manual counting, via automated detection and quantification algorithms, or we calculated the average relative staining intensity for F4/80 where it was not possible to identify individual cells in the network of macrophages, within the tumours as well as in unaffected regions of kidney tissue (normal) within the same mouse (Supplementary Fig. S11k,l). Consistent with HIF-2 α deficient tumours showing the highest GAGE mRNA signatures of T cell inflammation, Vhl $\Delta\Delta$ Trp53 $\Delta\Delta$ Rb1 $\Delta\Delta$ Hif2a $\Delta\Delta$ tumours displayed increased densities of CD3 (Fig. 6b), CD4 (Fig. 6c) and CD8 (Fig. 6d) positive T cells compared to normal tissue, whereas only CD8 positive T cell densities were significantly increased in Vhl $\Delta\Delta$ Trp53 $\Delta\Delta$ Rb1 $\Delta\Delta$ and Vhl $\Delta\Delta$ Trp53 $\Delta\Delta$ Rb1 $\Delta\Delta$ Hif1a $\Delta\Delta$ tumours compared to the respective normal tissues. Notably, both Vhl $\Delta\Delta$ Trp53 $\Delta\Delta$ Rb1 $\Delta\Delta$ Hif1a $\Delta\Delta$ and Vhl $\Delta\Delta$ Trp53 $\Delta\Delta$ Rb1 $\Delta\Delta$ Hif2a $\Delta\Delta$ tumours exhibited higher densities of CD8 positive T cells than Vhl $\Delta\Delta$ Trp53 $\Delta\Delta$ Rb1 $\Delta\Delta$ tumours, in line with the Bindea&Others ssGSEA CD8 T cell signature results. Vhl $\Delta\Delta$ Trp53 $\Delta\Delta$ Rb1 $\Delta\Delta$ Hif1a $\Delta\Delta$ but not Vhl $\Delta\Delta$ Trp53 $\Delta\Delta$ Rb1 $\Delta\Delta$ Hif2a $\Delta\Delta$ tumours showed increased densities of CD4 positive cells compared to Vhl $\Delta\Delta$ Trp53 $\Delta\Delta$ Rb1 $\Delta\Delta$ tumours. This result is not reflected by any of the bioinformatic immune deconvolution methods. Analyses of the T cell activation markers CD69 (Fig. 6e) and Perforin (Fig. 6f) revealed that all tumours showed increased T cell activation compared to normal tissue, and that Vhl $\Delta\Delta$ Trp53 $\Delta\Delta$ Rb1 $\Delta\Delta$ Hif2a $\Delta\Delta$ tumours showed higher levels of T cell activation than Vhl $\Delta\Delta$ Trp53 $\Delta\Delta$ Rb1 $\Delta\Delta$ or Vhl $\Delta\Delta$ Trp53 $\Delta\Delta$ Rb1 $\Delta\Delta$ Hif1a $\Delta\Delta$ tumours, consistent with our conclusions from the GAGE analyses. There was no statistically significant enrichment of PD-1 positive cells, a marker of exhausted T cells, in any of the tumour genotypes (Fig. 6g). B220 staining revealed increased B cell density in all tumour genotypes compared to normal tissue, and higher densities in Vhl $\Delta\Delta$ Trp53 $\Delta\Delta$ Rb1 $\Delta\Delta$ Hif1a $\Delta\Delta$ tumours than in Vhl $\Delta\Delta$ Trp53 $\Delta\Delta$ Rb1 $\Delta\Delta$ or Vhl $\Delta\Delta$ Trp53 $\Delta\Delta$ Rb1 $\Delta\Delta$ Hif2a $\Delta\Delta$ tumours (Fig. 6h). Interestingly, these observations do not reflect the results of the bioinformatic immune cell deconvolutions for B cells. In contrast to the relatively low numbers of tumour-infiltrating lymphocytes, myeloid lineage cells were much more abundant within tumours. CD68 positive monocytes/macrophages (Fig. 6i), F4/80 positive macrophages (Fig. 6j) and Ly-6G-labelled granulocytes/neutrophils (Fig. 6k) were highly abundant in tumours compared to normal tissue but there were no differences in abundance of these cells between tumour genotypes.”

It is of note that the values of CD4+ cells and CD8+ cells in Figure 5C in the VpRH1 and VpRH2 varied greatly along the y axis. Some values are very low while others are much higher. How many areas are counted? Are the technical duplicates applied in this assay? The author should explain this variation.

Thank you for this question. Cell densities were determined from analyses of the entire tumour (or an equivalently sized region of non-tumour kidney tissue) and not from analyses of multiple representative fields. We added a new Suppl. Fig S11 to address the issue of inter-tumoural heterogeneity. We have clarified the methodological aspects of the quantifications in the Results and in the Methods as below.

Results: “We quantified the densities of positively stained cells either by manual counting, via automated detection and quantification algorithms, or we calculated the average relative staining intensity for F4/80 where it was not possible to identify individual cells in the network of macrophages, within the tumours as well as in unaffected regions of kidney tissue (normal) within the same mouse (Supplementary Fig. S11k,l).”

Methods “For analyses of immune cell markers sections were scanned using a Nanozoomer Scansystem (Hamamatsu Photonics). Automatic quantifications of B220, CD3, CD4, CD8a positive cells were carried out from duplicate stains (average values were determined) as previously described⁸⁵ using the VIS software suite (Visiopharm, Hoersholm, Denmark). Each tumor was outlined manually. Immune cell densities were calculated as cells per mm² based on surface area and immune cell quantification. The

quantifications of cells stained with F4/80 was performed using a positive pixel count and presented as “percentage positive pixel” (%PP). PD-1, Perforin, Ly-6G and CD69 stains were quantified by manual annotation of positively stained cells.”

Figures 6A-B, when the author identify HIF1A loss, how the “loss” was defined here?

This was based on GISTIC analyses. “*HIF1A* loss” includes tumours with either homozygous deletions (only a very few tumours show this) or heterozygous deletions (the vast majority) as reflected in the Oncoprints in Figure 7A. We have now included this information in the Figure legend.

The mRNA expression should be integrated with copy number data.

We have added an analysis of the relationship between copy number and mRNA abundance for *HIF1A* and *HIF2A*. Results p14:

“To investigate whether genetic alterations in *HIF1A* or *HIF2A* might influence immune cell infiltration in human ccRCC, we analysed data from the TCGA KIRC study (Firehose-legacy data set)⁹ using cBioPortal^{37,38}. ccRCC tumours frequently lose one copy of *HIF1A* and less frequently gain one copy of *HIF2A* (Fig. 7a). Loss of one copy of either gene correlated with lower mRNA levels but gain of a copy did not correlate with increased mRNA abundance (Supplementary Fig. S12a,b).”

We have furthermore carried out a new series of immune deconvolution analyses segregating tumours based on high and low expression of *HIF2A*.

Results p12: “Analyses of TCGA mRNA expression data revealed that *HIF2A*, but not *HIF1A*, is more highly abundant in ccRCC in comparison to normal kidney and that *HIF2A* shows a wide distribution of expression levels between different tumours (Fig. 5e,f). This upregulation and wide expression level distribution is not observed in chromophobe RCC or papillary RCC (Fig. 5e,f). The wide expression distribution provided a good basis to investigate potential correlations between *HIF2A* mRNA abundance and the abundance of mRNAs encoding MHC genes.”

Results p15: “To gain a more in-depth overview of the effects of *HIF* gene copy number or expression level alterations on the immune microenvironment, we performed immune deconvolution analyses of RNA-seq data, again using three independent methods of immune cell deconvolution; ssGSEA using the Bindea&Others and eTME gene signatures and using the CIBERSORT method⁴⁸. We compared *HIF1A* loss and *HIF2A* gain tumours to diploid human ccRCCs (Fig. 7n) and also took advantage of the wide distribution of mRNA expression levels of *HIF2A* to compare tumours in the top (Q4) and bottom (Q1) quartiles of *HIF2A* mRNA abundance. While *HIF2A* gain tumours exhibited very few alterations in immune scores, *HIF1A* loss tumours showed statistically significant increases or decreases in 57 of 83 immune signatures of a variety of lymphoid and myeloid lineage cells. Notable amongst these are consistently upregulated scores for T helper cells and for B cells, mirroring our immunohistochemical findings of the comparison of $Vhl^{\Delta/\Delta}Trp53^{\Delta/\Delta}Rb1^{\Delta/\Delta}$ and $Vhl^{\Delta/\Delta}Trp53^{\Delta/\Delta}Rb1^{\Delta/\Delta}Hif1a^{\Delta/\Delta}$ tumours. It should however be noted that the magnitude of the z-scores are generally low, suggesting that this group of ccRCCs on average exhibits numerous subtle differences in immune inflammation compared to tumours with normal *HIF1A* copy number. In contrast, high *HIF2A* mRNA expressing tumours displayed downregulation of scores for interferon- γ and for APM2 (measuring MHC class II antigen presentation), consistent with the upregulation of these features in mouse ccRCC tumours lacking HIF-2 α . Somewhat paradoxically, high *HIF2A* expressing tumours also displayed general upregulation of CD8 T cell scores and some NK cell scores and downregulation of all three scores for regulatory T cells and for the immune checkpoint proteins PD1 and CTLA4. These scores might be predicted to reflect elevated CD8 T cell activity. However, these tumours also display elevated scores for monocytes, neutrophils and mast cells, which in some settings contribute to suppression of anti-tumour CD8 T cell responses. Immunosuppressive mast cells were shown to correlate with *HIF2A* mRNA abundance in human ccRCC⁴⁹. Thus, the extent of T cell mediated anti-tumour immunity is likely to be determined by the balance of the abundance and activities of several different immune cell types in a manner that is partly influenced by *HIF2A* expression. Finally, it is also noteworthy that *HIF1A* copy loss and *HIF2A* mRNA high tumours showed opposite effects on signatures for pericytes, endothelial cells and angiogenesis, implying that both HIF-1 α and HIF-2 α may act as positive factors that promote blood vessel formation in ccRCC tumours.”

The author showed OS, how about DSS, DFS? Are they also significant?

DFS information for this data set is available but DFS is also significant and we include this in a new Supplementary Fig. 12. We would like to point out that the relationship between *HIF1A*/14q loss has been previously described in the literature and this is not the main point of the analysis that we present here.

Results p14: “ccRCC tumours that exhibit mono- or bi-allelic loss of *HIF1A* (collectively *HIF1A* loss) show worse overall survival (Fig. 7b) and progression-free survival (Supplementary Fig. S12c) than unaffected tumours, whereas there are no overall or progression-free survival differences between tumours with a copy number gain of *HIF2A* and unaffected tumours (Fig. 7c and Supplementary Fig S12d).”

Is HIF1A loss co-existing with other factors in patients showed worse survival? the multivariate Cox model should be applied here to ensure that impact of HIF1A loss is still significant after adjusting for other confounding factors.

We now show carried out these analyses, Results p15:

“Previous studies have identified that loss of larger regions of chromosome 14q, including the *HIF1A* gene, correlates with poor prognosis¹⁹. We took advantage of the extensive clinical and whole exome sequencing data of the TCGA dataset to investigate whether co-variants of *HIF1A* loss may account for the observed survival differences. Tumours with *HIF1A* loss were statistically more likely to have higher grade and stage, and display lymph node positivity and metastases (Supplementary Data 6), consistent with this subgroup representing a more aggressive form of ccRCC. The only mutation that occurred more frequently in the *HIF1A* loss subgroup than the unaltered subgroup was *BAP1*, which was detected in 9% of all ccRCC tumours (Supplementary Fig. S13e,f). However, *BAP1* mutation status alone did not significantly affect survival (Supplementary Fig. S13g) in this cohort and removal of all *BAP1* mutant tumours from the cohort did not alter the correlation of *HIF1A* loss with poor prognosis (Supplementary Fig. S13h,i). The conclusion that *BAP1* mutation status is not a relevant co-variant that affects survival outcome in the *HIF1A* loss cohort was also demonstrated by COX univariate (HR 1.839, 95% CI 1.332-2.539, $P = 0.00021$) and multivariate proportional hazards analyses (HR 1.776, 95% CI 1.274-2.474, $P = 0.00069$). These findings suggest that loss of one allele of *HIF1A*, which is predicted to lead to diminished HIF-1 α abundance, may be selected for during the evolution of some ccRCC tumours and that this correlates with aggressive disease.”

Supplementary Figure S2, the author showed coverage track of the RNA-seq data, however, the scale on y axis was not displayed and it is not clear whether the same scale was used for different tracks.

Thank you for pointing out this issue. The scales are normalised for each track to facilitate a visual comparison of the relative coverage of each exon in each sample. We have now included this information in the figure legend.

It is of note that for the the Floxed exon 2 of *hif2a*, there is still signals and coverage on exon 2, are these Residual floxed exons? What are the % stromal cells? If the Residual floxed exons was indeed caused by DNA derived from non-Cre-expressing cells of the tumour stroma, appropriate normalization approach should be applied to correct the % stromal cells for a fair comparison.

As we noted in the Results section of the manuscript, we believe that the residual floxed exons of all of the genes seen in the PCR and RNA seq analyses most likely come from different relative ratios of tumour cells to non-tumour cells within the different samples, reflecting different stromal and immune cell compositions of each tumour biopsy (which we observe both in our immune deconvolution bioinformatic analyses and in our IHC analyses). While we take the point about the fact that stromal contamination is an issue, this is a limitation of these analysis techniques of whole tumour DNA and RNA. We are not aware of any (simple) method that would allow us to quantitatively assess and adjust for non-tumour stromal contribution in these RNA/DNA samples and therefore are unable to implement the normalisation suggestion of the reviewer.

However, we also take the point that it is very important to show that *Hif2a* is really deleted in tumour cells. We now tested six different commercially-available anti-HIF-2 α antibodies using two different antigen retrieval methods and found one condition that gave specific nuclear signals for HIF-2 α in IHC stainings. Examples of these stainings are shown in Fig. 1d. Nuclear HIF-2 α staining was present in tumours from $Vhl^{\Delta/\Delta}Trp53^{\Delta/\Delta}Rb1^{\Delta/\Delta}$ (35/35 tumours) and $Vhl^{\Delta/\Delta}Trp53^{\Delta/\Delta}Rb1^{\Delta/\Delta}Hif1a^{\Delta/\Delta}$ (11/11 tumours) mice but absent in all tumours from $Vhl^{\Delta/\Delta}Trp53^{\Delta/\Delta}Rb1^{\Delta/\Delta}Hif2a^{\Delta/\Delta}$ (0/54 tumours) mice. This result demonstrates that HIF-2 α is deleted in the tumours that emerge in $Vhl^{\Delta/\Delta}Trp53^{\Delta/\Delta}Rb1^{\Delta/\Delta}Hif2a^{\Delta/\Delta}$ mice and validates our conclusion that HIF-2 α has relatively minor effects on tumour growth in this model.

REVIEWERS' COMMENTS:

Reviewer #1 (Remarks to the Author):

The authors have done a remarkable job addressing my comments and included over a dozen new data panels. The authors significantly expanded also on the implications of HIF2a loss on the TME and in particular on the immune response. This raises the question whether reduced tumorigenesis associated with HIF2a loss in the mouse may be related to an immune response.

Jim Brugarolas

Reviewer #2 (Remarks to the Author):

The authors took into account all comments regarding details of proteomic analysis. I do not have further corrections / suggestions for this section. I recommend accepting this manuscript.

Reviewer #3 (Remarks to the Author):

Most of the comments were addressed adequately, but very little was addressed experimentally. Importantly, the authors conducted IHC for HIF2 to validate their model, this is much appreciated. Also they did additional analyses for the immune part of their paper, which greatly improved the manuscript. But two important comments were not addressed experimentally. The authors claim that it will take them several years to address these if they use genetic manipulations in mice, which is fair, but there are several other ways to address the comments without doing genetic manipulations in mice. For example, it is possible to set up cell lines from established tumors which are VHL-HIF1+HIF2+, knock down HIF1 with shRNA or CRISPR, and inject the cells subcutaneously to the mice of the same genetic background or to immunodeficient animals (if the mixed background was used in original model). For the second comment, it is possible to deplete specific immune cell types with the antibodies, and assess tumor forming ability in the mice with VHL-/-H1-H2+, VHL-H1+H2-, and VHL-H1+H2+ tumors.

Here are the comments for the reference:

1) "Although the study did a good job showing that HIF1a is important for tumor initiation, no data were provided to address the issue as to why HIF1a is lost during tumor progression in 30% of patients. It would be helpful to see the data when HIF1a deletion is introduced at later stages of tumor progression in this mouse model and the tumor growth assessed, ideally with analysis of immune microenvironment (see comments on claims below)."

2) "In the context of the findings, it is important to conduct tumor growth experiments with inhibitors or activators of specific branches of immune system in mice with deletion of HIF1 or HIF2 of HIF1WT/HIF2WT to clear the message of the paper about the role of HIF1 and HIF2 in immune response and tumorigenesis. Currently the message is confusing and lacks experimental support."

Reviewer #4 (Remarks to the Author):

The authors have appropriately responded and satisfactorily addressed my comments.

Response to the comments of Reviewer 3:

Most of the comments were addressed adequately, but very little was addressed experimentally. Importantly, the authors conducted IHC for HIF2 to validate their model, this is much appreciated. Also they did additional analyses for the immune part of their paper, which greatly improved the manuscript.

Thank you, we agree that the suggestions of all of the reviewers were very helpful in improving our study.

But two important comments were not addressed experimentally.

1) "Although the study did a good job showing that HIF1a is important for tumor initiation, no data were provided to address the issue as to why HIF1a is lost during tumor progression in 30% of patients. It would be helpful to see the data when HIF1a deletion is introduced at later stages of tumor progression in this mouse model and the tumor growth assessed, ideally with analysis of immune microenvironment (see comments on claims below)." The authors claim that it will take them several years to address these if they use genetic manipulations in mice, which is fair, but there are several other ways to address the comments without doing genetic manipulations in mice. For example, it is possible to set up cell lines from established tumors which are VHL-HIF1+HIF2+, knock down HIF1 with shRNA or CRISPR, and inject the cells subcutaneously to the mice of the same genetic background or to immunodeficient animals (if the mixed background was used in original model).

We agree that this question about the role of HIF-1a at the start of tumour formation versus at later timepoints is an interesting and important one. The experiment suggested here by the reviewer is in fact the exact experiment that we presented in the analyses in Figure 2h,i. These studies showed that *Hif1a* knockdown in an established ccRCC cell line derived from a VpR tumour did not enhance or inhibit tumour growth. We now address this issue about the different potential roles of HIF-1 α at different timepoints of tumour formation with a modification of the discussion section. The new text reads:

"Our present study also highlights a potential general caveat to the interpretation of the results of studies of human ccRCC cell lines, or patient-derived tumour lines, in cell culture and in xenograft assays. In contrast to the clear requirement for *Hif1a* in tumour formation in the autochthonous setting, we find that HIF-1 α rather exhibits putative tumour suppressor activities in cell culture-based assays, including inducing the early loss of proliferative capacity of MEFs following *Vhl* deletion, inhibiting the proliferation of immortalised *Vhl/Trp53* null MEFs and inhibiting anchorage independent growth of a *Vhl/Trp53/Rb1* mutant mouse ccRCC cell line. Furthermore, HIF-1 α knockdown did not affect the growth of allograft tumours generated with this ccRCC cell line. This latter experiment argues against the idea that removal of HIF-1 α activity in an established mouse ccRCC tumour, to mimic the loss of HIF-1 α function that arises as a later event in tumour progression in a subset of human ccRCCs, has a potent effect on tumour aggressiveness. Collectively, our studies show that in the same genetic ccRCC tumour model, the cell culture and allograft assays do not reflect the *in vivo* requirement in the autochthonous setting. Previous studies which have interpreted the oncogenic and tumour suppressive roles of HIF- α factors in ccRCC based on studies of cultured ccRCC cells and xenograft models may therefore not necessarily be reflective of the true *in vivo* functions of HIF-1 α and HIF-2 α at different stages of tumour development in the physiological context of the kidney."

Regarding the study of the immune environment of these allografted tumours, we do not believe that this would in any way be informative because these tumours have to be grown in immune deficient Scid-beige mice due to the highly mixed genetic background of the VpR mouse line. In general, the study of inflammatory responses in allograft settings is less informative than the study of autochthonous tumour models because the sub-cutaneous nature of the tumour does not reflect the physiological environment of the primary tumour and because these studies do not reproduce the co-evolution of tumour cells and immune cells that is present in the autochthonous setting.

2) "In the context of the findings, it is important to conduct tumor growth experiments with inhibitors or activators of specific branches of immune system in mice with deletion of HIF1 or HIF2 or HIF1WT/HIF2WT to clear the message of the paper about the role of HIF1 and HIF2 in immune response and tumorigenesis. Currently the message is confusing and lacks experimental support." For the second comment, it is possible to deplete specific immune cell types with the antibodies, and assess tumor forming ability in the mice with VHL-/-H1-H2+, VHL-H1+H2-, and VHL-H1+H2+ tumors. While we agree that immune cell subset depletions can be useful in certain short term experimental settings, similar studies in the VpR autochthonous tumour model are not feasible due to the very long time frame of tumour evolution (5-12 months). Immune depletion experiments are in general limited in duration and efficiency by compensations in cellular homeostasis of the immune system and due to

the development by the host animal of neutralising antibodies to the immune depleting antibodies, which over time act to block the intended immune cell depletion.